# Improved Algorithms for Stochastic Linear Bandits Using Tail Bounds for Martingale Mixtures

**Hamish Flynn[1,2], David Reeb[1], Melih Kandemir[3], Jan Peters[2,4]**
[1]Bosch Center for Artificial Intelligence, Renningen, Germany
[2]Technische Universität Darmstadt, Germany
[3]University of Southern Denmark, Odense, Denmark
[4]Deutsches Forschungszentrum für Künstliche Intelligenz (DFKI), Germany
`hamish@robot-learning.de`, `david.reeb@de.bosch.com`,
`kandemir@imada.sdu.dk`, `jan.peters@tu-darmstadt.de`

## Abstract

We present improved algorithms with worst-case regret guarantees for the stochastic linear bandit problem. The widely used "optimism in the face of uncertainty" principle reduces a stochastic bandit problem to the construction of a confidence sequence for the unknown reward function. The performance of the resulting bandit algorithm depends on the size of the confidence sequence, with smaller confidence sets yielding better empirical performance and stronger regret guarantees. In this work, we use a novel tail bound for adaptive martingale mixtures to construct confidence sequences which are suitable for stochastic bandits. These confidence sequences allow for efficient action selection via convex programming. We prove that a linear bandit algorithm based on our confidence sequences is guaranteed to achieve competitive worst-case regret. We show that our confidence sequences are tighter than competitors, both empirically and theoretically. Finally, we demonstrate that our tighter confidence sequences give improved performance in several hyperparameter tuning tasks.

## 1 Introduction

The stochastic linear bandit problem is a sequential decision-making problem where, in each round $t$, a learner chooses an action $a_t$ and then receives a stochastic reward $r_t$ for its choice of action. The expected value of each reward is a linear function $\phi(a_t)^\top \boldsymbol{\theta}^*$ of a known feature vector $\phi(a_t)$ associated with the corresponding action, while $\boldsymbol{\theta}^*$ is unknown. The linear bandit problem has attracted a great deal of attention because it is expressive enough to be a faithful model of many real-world decision-making problems, such as news recommendation (Li et al., 2010) and dynamic pricing (Cohen et al., 2020), yet it is simple enough to make theoretical analysis tractable.

A popular way to design algorithms for linear bandits is to follow the principle of optimism in the face of uncertainty. This principle states that we should choose actions as if the expected reward function is as nice as plausibly possible. For linear bandits, the principle can be instantiated with a confidence sequence $\Theta_0, \Theta_1, \dots$ for the parameter vector $\boldsymbol{\theta}^* \in \Theta$ of the expected reward function. A confidence sequence is a sequence of subsets of the full parameter space $\Theta$, which is built iteratively as data becomes available and is constructed such that with high probability over the observed data, $\boldsymbol{\theta}^*$ is contained in each confidence set $\Theta_t$. One can then run an upper confidence bound (UCB) algorithm, which at round $t$ chooses an action $a_{t+1}$ by maximising the UCB $\max_{\boldsymbol{\theta} \in \Theta_t} \left\{ \phi(a)^\top \boldsymbol{\theta} \right\}$ with respect to $a$.

The popularity of UCB algorithms stems from the fact that they come with worst-case regret guarantees and often perform well in practice. However, the performance of a UCB algorithm is intimately

37th Conference on Neural Information Processing Systems (NeurIPS 2023).

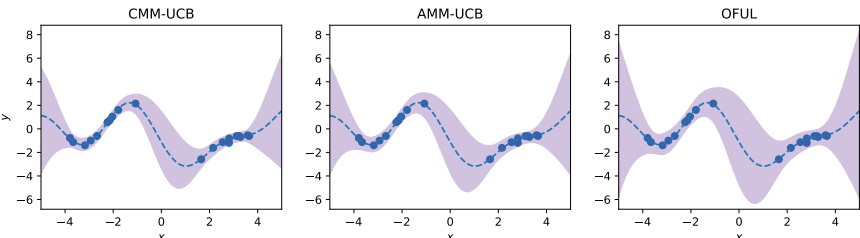

Figure 1: **Tighter upper and lower confidence bounds via tail bounds for martingale mixtures.** The upper and lower confidence bounds of CMM-UCB (left), AMM-UCB (middle), and OFUL (Abbasi-Yadkori et al., 2011) (right) for a test function linear in random Fourier features. The bounds from CMM-UCB and AMM-UCB are visibly closer to the true function (dashed line) than those of OFUL. The CMM-UCB confidence bounds are slightly tighter than the ones of AMM-UCB.

tied to the size of the confidence sets it uses. The smaller the subsets in the confidence sequence, the better the regret bound and, perhaps more importantly, the better the algorithm performs in practice.

In this work, we develop a general-purpose tail bound for martingale mixtures, which can be used to construct confidence sequences. When we specialise our general results to the linear bandit problem, the maximisation problem to compute the UCB is a convex program. We maximise the UCB over actions via gradient-based methods, and investigate two procedures for computing the UCB along with its gradient: (a) *Convex Martingale Mixture UCB (CMM-UCB):* We employ a convex solver for the UCB maximisation and calculate its gradients via differentiable convex optimisation (Agrawal et al., 2019); (b) *Analytic Martingale Mixture UCB (AMM-UCB):* We exploit weak Lagrangian duality to obtain an analytic upper bound on the UCB, whose gradient can be computed in closed-form or via standard automatic differentiation procedures.

Fig. 1 highlights a key observation: both of our UCBs are tighter than those used in the state-of-the-art OFUL algorithm (Abbasi-Yadkori et al., 2011) for stochastic linear bandits. We verify this claim empirically in Sec. 8 and prove it in App. C.2. We evaluate CMM-UCB, AMM-UCB, OFUL and various other linear bandit algorithms in several hyperparameter tuning problems (Sec. 8). We find that our tighter UCBs result linear bandit algorithms with better performance.

## 2   Related Work

Algorithms with regret guarantees have been developed for several variants of the stochastic linear bandit problem. Dani et al. (2008), Abbasi-Yadkori et al. (2009) and Rusmevichientong & Tsitsiklis (2010) proposed algorithms for a linear bandit problem where the action set is a fixed, possibly infinite subset of a finite-dimensional vector space. Auer (2002) and Chu et al. (2011) proposed algorithms for linear bandit problems where the action set has finite cardinality, but may change over time. Abbasi-Yadkori et al. (2011) proposed the OFUL algorithm for linear bandit problems with a changing and possibly infinite action set, which is essentially the problem that we investigate. We consider stochastic linear bandit problems where the reward function is a composition of a possibly non-linear feature map and a linear function. This can be seen as a restricted version of the stochastic kernelised bandit problem, where the kernel feature map is finite-dimensional. Srinivas et al. (2010); Valko et al. (2013); Chowdhury & Gopalan (2017); Camilleri et al. (2021); Salgia et al. (2021) and Li & Scarlett (2022) proposed algorithms with regret guarantees for various kernelised bandit problems.

In the bandit literature, confidence sets and confidence bounds constructed from online (e.g. non-i.i.d.) observation points for unknown linear functions have been proposed by Dani et al. (2008); Rusmevichientong & Tsitsiklis (2010) and Abbasi-Yadkori et al. (2011). Online confidence sets/bounds for unknown functions in separable Hilbert spaces and reproducing kernel Hilbert spaces (RKHSs) have been proposed by Srinivas et al. (2010); Abbasi-Yadkori (2012); Kirschner & Krause (2018) and Durand et al. (2018). Russo & Van Roy (2013) derived online confidence sets for unknown functions belonging to arbitrary function classes.

We use the term "mixture of martingales", or martingale mixture, to refer to a martingale of the form $\mathbb{E}_{v \sim P}[M_t(v)]$, where $(M_t(v)|t \in \mathbb{N})$ is a collection of martingales indexed by the variable $v \in \mathcal{V}$. Martingale mixtures can be traced back to (Darling & Robbins, 1968; Robbins, 1970), and have been used to construct confidence sequences since at least the work of Lai (1976). Proofs of tail bounds for martingale mixtures typically use the method of mixtures, which was first used by Robbins & Siegmund (1970) and was later popularised by de la Peña et al. (2004, 2009). Methods for martingale mixtures have seen renewed interest in the sequential testing literature (Howard et al., 2020; Kaufmann & Koolen, 2021). Examples of confidence sequences for bandits that use martingale mixtures include the works of Abbasi-Yadkori et al. (2011); Abbasi-Yadkori (2012); Kirschner & Krause (2018); Durand et al. (2018); Neiswanger & Ramdas (2021). Unlike in these examples, we construct confidence sequences based on *adaptive* martingale mixtures $(\mathbb{E}_{v \sim P_t}[M_t(v)]|t \in \mathbb{N})$, where the mixture distribution $P_t$ can be refined as more data is acquired at each time $t$.

## 3 Problem Statement and Background

We consider a problem in which a learner plays a game over a sequence of $T$ rounds, where $T$ may not be known in advance. In each round $t$, the learner observes an action set $\mathcal{A}_t$ and must choose an action $a_t \in \mathcal{A}_t$. The learner then receives a reward $r_t = \phi(a_t)^\top \boldsymbol{\theta}^* + \epsilon_t$. The feature map $\phi : \mathcal{A} \to \mathbb{R}^d$ is a known function that maps actions to $d$-dimensional feature vectors, where $\mathcal{A} = \bigcup_t \mathcal{A}_t$. $\boldsymbol{\theta}^* \in \mathbb{R}^d$ is an unknown parameter with Euclidean norm bounded by some known $B > 0$, i.e. $\|\boldsymbol{\theta}^*\|_2 \leq B$. $\epsilon_1, \epsilon_2, \ldots, \epsilon_T$ are conditionally zero-mean $\sigma$-sub-Gaussian noise variables. These assumptions on $\boldsymbol{\theta}^*$ and $\epsilon_1, \epsilon_2, \ldots, \epsilon_T$ are standard in the linear bandit literature, see e.g. (Abbasi-Yadkori et al., 2011). While our regret analysis applies to any action sets, our algorithms focus on the case where the action sets $\mathcal{A}_t$ are continuous subsets of $\mathbb{R}^{d_\mathcal{A}}$.

The goal of the learner is to choose a sequence of actions that maximises the total expected reward, which is equal to $\sum_{t=1}^T \phi(a_t)^\top \boldsymbol{\theta}^*$ after $T$ rounds. We use cumulative regret, which is the difference between the total expected reward of the learner and the optimal strategy, to evaluate the learner. For a single round, we define the regret as $\Delta(a_t) = \phi(a_t^*)^\top \boldsymbol{\theta}^* - \phi(a_t)^\top \boldsymbol{\theta}^*$, where $a_t^* = \operatorname{argmax}_{a \in \mathcal{A}_t} \{\phi(a)^\top \boldsymbol{\theta}^*\}$. After $T$ rounds, the cumulative regret is $\Delta_{1:T} = \sum_{t=1}^T \Delta(a_t)$.

**Confidence Sequences.** For any level $\delta \in (0, 1]$, a $(1 - \delta)$-confidence sequence for the parameter vector $\boldsymbol{\theta}^*$ is a sequence $\Theta_1, \Theta_2, \ldots$ of subsets of $\mathbb{R}^d$, such that each $\Theta_t$ can be calculated using the data $a_1, r_1, \ldots, a_t, r_t$ and the sequence satisfies

$$\mathbb{P}_{\substack{a_1, a_2, \ldots, \\ r_1, r_2, \ldots,}} [\forall t \geq 1 : \boldsymbol{\theta}^* \in \Theta_t] \geq 1 - \delta.$$

A confidence sequence $\Theta_1, \Theta_2, \ldots$ is thus a sequence of data-dependent confidence sets such that with high probability over the random actions and rewards, $\boldsymbol{\theta}^* \in \Theta_t$ holds for all $t \geq 1$ simultaneously. We remark that the confidence sets $\Theta_t$ in this paper are random closed sets in the sense of Def. 1.1.1 of (Molchanov, 2005), which implies that the event $\boldsymbol{\theta} \in \Theta_t$ is actually measurable for all $\boldsymbol{\theta} \in \mathbb{R}^d$.

## 4 UCB Algorithms for Linear Bandits

We describe here how to transform confidence sets for $\boldsymbol{\theta}^*$ into a UCB algorithm for the linear bandit problem. Such algorithms have appeared under various names, such as LinRel (Auer, 2002), LinUCB (Li et al., 2010) and OFUL (Abbasi-Yadkori et al., 2011). We refer to this meta algorithm as LinUCB, and give its pseudo-code in Algorithm 1. When run with our confidence sets, we call this algorithm CMM-UCB resp. AMM-UCB (see Sec. 6).

In each round $t$, the first step is to construct a confidence set $\Theta_t$ from the previous observations $\{(a_k, r_k)\}_{k=1}^t$. If $\boldsymbol{\theta}^* \in \Theta_t$ with high probability, then for any action $a$,

$$\mathrm{UCB}_{\Theta_t}(a) := \max_{\theta \in \Theta_t} \{\phi(a)^\top \boldsymbol{\theta}\} \tag{1}$$

is an upper confidence bound (UCB) on $\phi(a)^\top \boldsymbol{\theta}^*$. Taking $\min_{\theta \in \Theta_t}$ in (1) yields the lower confidence bound $\mathrm{LCB}_{\Theta_t}(a)$. Once a confidence set $\Theta_t$ has been constructed and the next action set $\mathcal{A}_{t+1}$ has been observed, the LinUCB algorithm chooses the action

$$a_{t+1} = \operatorname*{argmax}_{a \in \mathcal{A}_{t+1}} \{\mathrm{UCB}_{\Theta_t}(a)\}, \tag{2}$$

---

**Algorithm 1:** LinUCB

---
**for** $t = 0, 1, 2, \ldots$ **do**
  Construct a confidence set $\Theta_t$ from $\{(a_k, r_k)\}_{k=1}^t$
  Observe next action set $\mathcal{A}_{t+1}$
  Play next action $a_{t+1} = \operatorname{argmax}_{a \in \mathcal{A}_{t+1}} \{\mathrm{UCB}_{\Theta_t}(a)\}$
  Observe next reward $r_{t+1}$
**end**

---

which maximises the UCB. The remaining challenge lies in the construction of the confidence sets.

## 5 Confidence Sequences from Martingale Mixtures

In this section, we develop a general-purpose tail bound for adaptive martingale mixtures. We then specialise our general result to the stochastic linear bandit setting, described in Sec. 3, and construct confidence sequences for the parameter $\boldsymbol{\theta}^*$.

### 5.1 General-Purpose Tail Bound for Adaptive Martingale Mixtures

We consider a general setting in which we are given a filtration $(\mathcal{H}_t | t \in \mathbb{N})$, a sequence of adapted random functions $(Z_t : \mathbb{R} \to \mathbb{R} | t \in \mathbb{N})$, and a sequence of predictable random variables $(\lambda_t | t \in \mathbb{N})$. For $f_t \in \mathbb{R}$, we define the conditional cumulant generating function $\psi_t(f_t, \lambda_t)$ as

$$\psi_t(f_t, \lambda_t) := \ln \left( \mathbb{E} \left[ \exp(\lambda_t Z_t(f_t)) | \mathcal{H}_{t-1} \right] \right),$$

where the expectation $\mathbb{E}$ is over $Z_t(f_t)$ and $\lambda_t$ (although $\lambda_t$ is non-random when conditioned on $\mathcal{H}_{t-1}$). We use the shorthand $\boldsymbol{f}_t := (f_1, f_2, \ldots, f_t)$ and $\boldsymbol{\lambda}_t := (\lambda_1, \lambda_2, \ldots, \lambda_t)$. Let

$$M_t(\boldsymbol{f}_t, \boldsymbol{\lambda}_t) = \exp \left( \sum_{k=1}^t \lambda_k Z_k(f_k) - \psi_k(f_k, \lambda_k) \right). \tag{3}$$

$M_t(\boldsymbol{f}_t, \boldsymbol{\lambda}_t)$ is a slight generalisation of the martingale used in App. B.1 of (Russo & Van Roy, 2013). One can show that for any sequence $(f_t | t \in \mathbb{N})$, $(M_t(\boldsymbol{f}_t, \boldsymbol{\lambda}_t) | t \in \mathbb{N})$ is a martingale and $\mathbb{E}[M_t(\boldsymbol{f}_t, \boldsymbol{\lambda}_t)] = 1$ (App. A.1). We will now construct an *adaptive* martingale mixture.

We call a data-dependent sequence of probability distributions $(P_t | t \in \mathbb{N})$ an *adaptive sequence of mixture distributions* if: (a) $P_t$ is a distribution over $\boldsymbol{f}_t \in \mathbb{R}^t$; (b) $P_t$ is $\mathcal{H}_{t-1}$-measurable; (c) the distributions are consistent in the sense that their marginals coincide, i.e. $\int P_t(\boldsymbol{f}_t) df_t = P_{t-1}(\boldsymbol{f}_{t-1})$ for all $t$. These conditions on the sequence of distributions ensure that the martingale mixture $(\mathbb{E}_{\boldsymbol{f}_t \sim P_t}[M_t(\boldsymbol{f}_t, \boldsymbol{\lambda}_t)] | t \in \mathbb{N})$ is in fact a martingale. In App. A.1, we verify this and show that $\mathbb{E}[\mathbb{E}_{\boldsymbol{f}_t \sim P_t}[M_t(\boldsymbol{f}_t, \boldsymbol{\lambda}_t)]] = 1$. From here, we can use Ville's inequality for non-negative supermartingales (Ville, 1939) to obtain our general-purpose tail bound.

**Theorem 5.1** (Tail Bound for Adaptive Martingale Mixtures). *For any $\delta \in (0, 1)$, any sequence of predictable random variables $(\lambda_t | t \in \mathbb{N})$, and any adaptive sequence of mixture distributions $(P_t | t \in \mathbb{N})$, the following holds with probability at least $1 - \delta$:*

$$\ln \left( \mathbb{E}_{\boldsymbol{f}_t \sim P_t} [M_t(\boldsymbol{f}_t, \boldsymbol{\lambda}_t)] \right) \leq \ln(1/\delta) \quad \text{for all } t \geq 1.$$

We provide a proof of this result in App. A.2. Note that if each $\psi_k(f_k, \lambda_k)$ in (3) is replaced by an upper bound on $\psi_k(f_k, \lambda_k)$, the statement of the theorem still holds.

Thm. 5.1 is closely related to the general-purpose anytime PAC-Bayes bound in Thm. 3.1 of (Chugg et al., 2023). The main difference is that our inequality holds for adaptive sequences of mixture distributions or priors. PAC-Bayes bounds with somewhat similar adaptive sequences of mixture distributions/priors have recently been proposed by Haddouche & Guedj (2022, 2023).

## 5.2 Confidence Sequences for Stochastic Linear Bandits

We now specialise Thm. 5.1 to the stochastic linear bandit setting. For the filtration $(\mathcal{H}_t | t \in \mathbb{N})$, we set $\mathcal{H}_t$ to be the $\sigma$-algebra generated by $(a_1, r_1, \ldots, a_t, r_t, a_{t+1})$. For reasons that will become clear, we choose $Z_t(f_t) = (f_t - \phi(a_t)^\top \boldsymbol{\theta}^*)\epsilon_t$. Since $Z(f_t)$ is linear in the noise variable $\epsilon_t$, $\psi_t(f_t, \lambda_t)$ can be upper bounded using the sub-Gaussian property of $\epsilon_t$. We have

$$\psi_t(f_t, \lambda_t) = \ln\left(\mathbb{E}\left[\exp\left(\lambda_t(f_t - \phi(a_t)^\top \boldsymbol{\theta}^*)\epsilon_t\right) | \mathcal{H}_{t-1}\right]\right) \leq \lambda_t^2 \sigma^2 (f_t - \phi(a_t)^\top \boldsymbol{\theta}^*)^2/2. \quad (4)$$

With this upper bound on $\psi_t(f_t, \lambda_t)$, Thm. 5.1 implies that, with probability at least $1 - \delta$

$$\mathbb{E}_{\boldsymbol{f}_t \sim P_t}\left[\exp\left\{\sum_{k=1}^t \lambda_k(f_k - \phi(a_k)^\top \boldsymbol{\theta}^*)(r_k - \phi(a_k)^\top \boldsymbol{\theta}^*) - \frac{\sigma^2}{2}\lambda_k^2(f_k - \phi(a_k)^\top \boldsymbol{\theta}^*)^2\right\}\right] \leq \frac{1}{\delta}.$$

Since $Z_f(f_t)$ is linear in $f_t$, this integral has a closed-form solution whenever the mixture distribution is a Gaussian $P_t = \mathcal{N}(\boldsymbol{\mu}_t, \boldsymbol{T}_t)$. Although there is a closed-form solution for any predictable sequence $(\lambda_t | t \in \mathbb{N})$ (see App. B.1), we choose $\lambda_t \equiv 1/\sigma^2$, which yields a relatively simple convex quadratic constraint for $\boldsymbol{\theta}^*$. Collecting the feature vectors in $\Phi_t := [\phi(a_1), \ldots, \phi(a_t)]^\top \in \mathbb{R}^{t \times d}$ and writing the reward vector $\boldsymbol{r}_t := [r_1, \ldots, r_t]^\top$, we arrive at (see App. B.2)

$$\|\Phi_t \boldsymbol{\theta}^* - \boldsymbol{r}_t\|_2^2 \leq (\boldsymbol{\mu}_t - \boldsymbol{r}_t)^\top \left(\mathbb{1} + \frac{\boldsymbol{T}_t}{\sigma^2}\right)^{-1}(\boldsymbol{\mu}_t - \boldsymbol{r}_t) + \sigma^2 \ln\det\left(\mathbb{1} + \frac{\boldsymbol{T}_t}{\sigma^2}\right) + 2\sigma^2 \ln\frac{1}{\delta}. \quad (5)$$

This inequality has an attractive interpretation. At each step $t$ of the bandit process, the (unknown) ground-truth reward vector $\Phi_t^\top \boldsymbol{\theta}^*$ lies within a sphere around the observed reward vector $\boldsymbol{r}_t$, with squared radius equal to the RHS of (5). One can think of the mean vector $\boldsymbol{\mu}_t$ as a prediction of the reward vector $\boldsymbol{r}_t$, given the previous data $a_1, r_1, \ldots, a_{t-1}, r_{t-1}, a_t$. The covariance matrix $\boldsymbol{T}_t$ can be thought of as the uncertainty associated with the prediction $\boldsymbol{\mu}_t$. If $\boldsymbol{\mu}_t$ is a good prediction of $\boldsymbol{r}_t$, then the quadratic "prediction error" term in (5) will be close to 0, and we can afford to choose $\boldsymbol{T}_t$ close to zero to minimise the log determinant term. In this situation, (5) can give a much tighter constraint than the naive bound $\sim t\sigma^2$, especially when $\sigma$ is a pessimistic upper bound on the true sub-Gaussian parameter. This naive bound follows from the observation that $\|\Phi_t \boldsymbol{\theta}^* - \boldsymbol{r}_t\|_2 = \|\boldsymbol{\epsilon}_t\|_2$, where $\boldsymbol{\epsilon}_t = (\epsilon_1, \ldots, \epsilon_t)$. Combining the constraint in (5) with our assumption $\|\boldsymbol{\theta}^*\|_2 \leq B$ yields our confidence sequence for $\boldsymbol{\theta}^*$.

**Corollary 5.2** (Martingale Mixture Confidence Sequence). *For any adaptive sequence of mixture distributions $P_t = \mathcal{N}(\boldsymbol{\mu}_t, \boldsymbol{T}_t)$, it holds with probability at least $1 - \delta$ that for all $t \geq 1$ simultaneously, $\boldsymbol{\theta}^*$ lies in the set*

$$\Theta_t = \left\{\boldsymbol{\theta} \in \mathbb{R}^d \,\middle|\, \|\Phi_t \boldsymbol{\theta} - \boldsymbol{r}_t\|_2 \leq R_{\text{MM},t} \quad \text{and} \quad \|\boldsymbol{\theta}\|_2 \leq B\right\}, \quad (6)$$

*where we define $R_{\text{MM},t}^2$ as the right-hand-side of Eq. (5).*

The boundaries of the constraints in (6) are both ellipsoids in $\mathbb{R}^d$, which means that each $\Theta_t$ is the intersection of (the interiors of) two ellipsoids.

## 6 Martingale Mixture UCB Algorithms

In this section, we describe our CMM-UCB and AMM-UCB algorithms, which are two different implementations of LinUCB (Algorithm 1) with our confidence sequence from Corollary 5.2.

### 6.1 UCB Computation and Optimisation

To run the LinUCB action selection rule with our confidence sequence, we need to be able to maximise $\text{UCB}_{\Theta_t}(a)$ with respect to $a$. The value of the UCB at the action $a$ is the solution of

$$\text{UCB}_{\Theta_t}(a) = \max_{\boldsymbol{\theta} \in \mathbb{R}^d} \phi(a)^\top \boldsymbol{\theta} \quad \text{s.t.} \quad \|\Phi_t \boldsymbol{\theta} - \boldsymbol{r}_t\|_2 \leq R_{\text{MM},t} \quad \text{and} \quad \|\boldsymbol{\theta}\|_2 \leq B. \quad (7)$$

This is a convex optimisation problem, which can be efficiently solved via convex programming. If the action sets have finite cardinality, $\text{UCB}_{\Theta_t}(a)$ can be maximised by solving (7) for each $a \in \mathcal{A}_t$ and then comparing the solutions. If the action sets are continuous subsets of $\mathbb{R}^{d_{\mathcal{A}}}$, then exact maximisation of $\text{UCB}_{\Theta_t}(a)$ is (in general) infeasible. For example, when the feature map $\phi$ is linear in $a$, $\text{UCB}_{\Theta_t}(\cdot)$ is the maximum over a set of linear functions, which is a convex function of $a$ (see Eq. (3.7) in Sec. 3.2.3 of Boyd & Vandenberghe (2004)). Since maximisation of a convex function is (in general) NP-hard, exact maximisation of $\text{UCB}_{\Theta_t}(a)$ is also (in general) NP-hard. For this reason, when the action sets are continuous subsets of $\mathbb{R}^{d_{\mathcal{A}}}$ (and $\phi$ is differentiable), we approximately maximise $\text{UCB}_{\Theta_t}(a)$ via gradient-based local search.

## 6.2 Convex Martingale Mixture UCB

Our Convex Martingale Mixture UCB (CMM-UCB) algorithm is based on computing (7) using numerical convex (conic) solvers from the CVXPY library (Diamond & Boyd, 2016; Agrawal et al., 2018). Note that (7) is already stated in a conic form, which is favourable for conic solvers (Boyd & Vandenberghe, 2004). Solving (7) numerically gives the tightest UCBs that can be obtained from our confidence sequence. To compute the gradient of $\text{UCB}_{\Theta_t}(a)$ with respect to the action $a$, we use recently developed methods for differentiating conic programs at their optimum (Agrawal et al., 2019), which are implemented in the cvxpylayers library.

## 6.3 Analytic Martingale Mixture UCB

Our Analytic Martingale Mixture UCB (AMM-UCB) algorithm uses an analytic upper bound on the solution of (7). The resulting analytic confidence bounds are looser than the numerical confidence bounds used by CMM-UCB, but are cheaper to evaluate and maximise. Theorem 6.1 states our upper bound on the solution of (7).

**Theorem 6.1** (Analytic UCB). *For all $\alpha > 0$, we have*

$$\text{UCB}_{\Theta_t}(a) = \max_{\boldsymbol{\theta} \in \Theta_t} \left\{ \phi(a)^\top \boldsymbol{\theta} \right\} \leq \phi(a)^\top \widehat{\boldsymbol{\theta}}_{\alpha,t} + R_{\text{AMM},t} \sqrt{\phi(a)^\top \left( \Phi_t^\top \Phi_t + \alpha \mathbb{1} \right)^{-1} \phi(a)}, \quad (8)$$

$$\textit{where} \qquad \widehat{\boldsymbol{\theta}}_{\alpha,t} = \left( \Phi_t^\top \Phi_t + \alpha \mathbb{1} \right)^{-1} \Phi_t^\top \boldsymbol{r}_t,$$

$$R_{\text{AMM},t}^2 = R_{\text{MM},t}^2 + \alpha B^2 - \boldsymbol{r}_t^\top \boldsymbol{r}_t + \boldsymbol{r}_t^\top \Phi_t \left( \Phi_t^\top \Phi_t + \alpha \mathbb{1} \right)^{-1} \Phi_t^\top \boldsymbol{r}_t.$$

In App. C.1, we derive this analytic UCB by partial optimisation of the Lagrangian dual function. Using strong duality, one can show that the analytic UCB minimised with respect to $\alpha$ is equal to $\text{UCB}_{\Theta_t}(a)$. Due to the closed-form expression of the analytic UCB in (8), its gradient with respect to $a$ can be computed using standard automatic differentiation packages.

## 6.4 Choosing the Mixture Distributions

Both of our algorithms require us to choose mixture distributions $P_t = \mathcal{N}(\boldsymbol{\mu}_t, \boldsymbol{T}_t)$. The mixture distributions play a role similar to the priors used in the PAC-Bayes (Shawe-Taylor & Williamson, 1997; McAllester, 1998; Guedj, 2019; Alquier, 2021) and luckiness (Grünwald, 2007, 2023) frameworks. Our confidence sequences and regret bounds are valid for any sequence of adaptive mixture distributions, but (as seen in Eq. (5)) if better/worse mixture distributions are chosen, then our confidence sequences will get smaller/bigger and our regret guarantees will get tighter/looser.

Here, we describe some sensible choices for the mixture distributions. In order for a sequence of Gaussian mixture distributions $(\mathcal{N}(\boldsymbol{\mu}_t, \boldsymbol{T}_t) | t \in \mathbb{N})$ to be a *sequence of adaptive mixture distributions* (as defined in Section 5.1), we require: (a) $\boldsymbol{\mu}_t$ and $\boldsymbol{T}_t$ can only depend on $a_1, \ldots, a_t$ and $r_1, \ldots, r_{t-1}$; (b) the first $t-1$ elements of $\boldsymbol{\mu}_t$ must be equal to $\boldsymbol{\mu}_{t-1}$; (c) the upper left $t-1 \times t-1$ block of $\boldsymbol{T}_t$ must be $\boldsymbol{T}_{t-1}$; (d) $\boldsymbol{T}_t$ must be positive (semi-)definite. These conditions are all satisfied if we use a mean vector $\boldsymbol{\mu}_t$ and covariance matrix $\boldsymbol{T}_t$ of the form

$$\boldsymbol{\mu}_t = [m(a_1), m(a_2), \ldots, m(a_t)]^\top, \qquad \boldsymbol{T}_t = \begin{bmatrix} k(a_1, a_1) & k(a_1, a_2) & \cdots & k(a_1, a_t) \\ k(a_2, a_1) & k(a_2, a_2) & \cdots & k(a_2, a_t) \\ \vdots & \vdots & \ddots & \vdots \\ k(a_t, a_1) & k(a_t, a_2) & \cdots & k(a_t, a_t) \end{bmatrix}, \quad (9)$$

where $m : \mathcal{A} \to \mathbb{R}$ is a mean function and $k : \mathcal{A} \times \mathcal{A} \to \mathbb{R}$ is a positive-definite kernel function. For the linear bandit problem, it is natural to use a linear mean function $m(a) = \phi(a)^\top \boldsymbol{\theta}_0$ and a linear kernel function $k(a, a') = \phi(a)^\top \boldsymbol{\Sigma}_0 \phi(a')$ (where $\boldsymbol{\Sigma}_0$ is symmetric and positive-definite), since the resulting mixture distribution assigns non-zero probability only to those vectors of function values $\boldsymbol{f}_t$ that could have been generated by a linear reward function. By direct computation, the Gaussian mixture distribution with this $m$ and $k$, and with $\boldsymbol{\mu}_t$ and $\boldsymbol{T}_t$ as in (9), is $P_t = \mathcal{N}(\Phi_t \boldsymbol{\theta}_0, \Phi_t \boldsymbol{\Sigma}_0 \Phi_t^\top)$. When $\boldsymbol{\theta}_0 = \boldsymbol{0}$ and $\boldsymbol{\Sigma}_0 = \mathbb{1}$, we recover what we call the *standard mixture distributions* $P_t = \mathcal{N}(\boldsymbol{0}, \Phi_t \Phi_t^\top)$.

Choosing $\boldsymbol{T}_t = \Phi_t \boldsymbol{\Sigma}_0 \Phi_t^\top$ has the additional benefit that it allows for cheaper computation of the radius $R_{\mathrm{MM},t}$. In App. F, we show that one (only) has to compute the inverse and determinant of a $d \times d$ matrix instead of the inverse and determinant of the $t \times t$ matrix $\mathbb{1} + \boldsymbol{T}_t/\sigma^2$. Finally, we remark that the requirement that $(\mathcal{N}(\boldsymbol{\mu}_t, \boldsymbol{T}_t)|t \in \mathbb{N})$ is an adaptive sequence of mixture distributions allows for "more adaptive" choices of $\boldsymbol{\mu}_t$ and $\boldsymbol{T}_t$. In App E.2, we describe and investigate a method for updating $\boldsymbol{\mu}_t$ and $\boldsymbol{T}_t$ at each round $t$ based on previously observed actions *and* rewards.

# 7 Regret Bounds

In this section, we establish cumulative regret bounds for our CMM-UCB and AMM-UCB algorithms. First, we state a data-dependent regret bound that illustrates how the radius of the analytic UCB from Sec. 6.3 influences the regret of both algorithms. Then, we prove a data-independent regret bound which illustrates the worst-case growth rate of the cumulative regret, with explicit dependence on the feature vector dimension $d$ and the number of rounds $T$. We begin by stating the assumptions (which are standard) under which our regret bounds hold.

**Assumption 7.1** (Sub-Gaussian noise)**.** Let $\mathcal{H}_t$ denote (the $\sigma$-algebra generated by) $(a_1, r_1, \ldots, a_t, r_t, a_{t+1})$. Each noise variable $\epsilon_t$ is conditionally zero-mean and $\sigma$-sub-Gaussian, which means

$$\mathbb{E}\left[\epsilon_t | \mathcal{H}_{t-1}\right] = 0, \quad \text{and} \quad \forall \lambda \in \mathbb{R}, \ \mathbb{E}\left[\exp(\lambda \epsilon_t) | \mathcal{H}_{t-1}\right] \leq \exp(\lambda^2 \sigma^2 / 2).$$

**Assumption 7.2** (Bounded parameter vector)**.** For some $B > 0$, $\|\boldsymbol{\theta}^*\|_2 \leq B$.

**Assumption 7.3** (Bounded feature vectors)**.** For some $L > 0$, $\|\phi(a)\|_2 \leq L$ for all $a \in \mathcal{A}$.

**Assumption 7.4** (Bounded expected reward)**.** For some $C > 0$, $\phi(a)^\top \boldsymbol{\theta}^* \in [-C, C]$ for all $a \in \mathcal{A}$.

We remark that to run our algorithms and evaluate the data-dependent regret bound in Thm. 7.5, we only need to know (upper bounds on) the sub-Gaussian parameter $\sigma$ and the norm bound $B$. Assumption 7.2 and Assumption 7.3 together imply that Assumption 7.4 must hold with $C \leq LB$. We nevertheless state it as a separate assumption because this leaves open the possibility that a better (than $LB$) value for $C$ is known.

## 7.1 Data-Dependent Regret Bounds

Several authors (Dani et al., 2008; Abbasi-Yadkori et al., 2011; Russo & Van Roy, 2013) have shown that the cumulative regret of a UCB algorithm can be upper bounded by the sum of the widths of the confidence sets or confidence bounds that it uses. The width of a confidence set $\Theta_t$ at the action $a$ is the difference between the corresponding UCB and the LCB at $a$ (i.e., $\max_{\theta \in \Theta_t} \{\phi(a)^\top \boldsymbol{\theta}\} - \min_{\theta \in \Theta_t} \{\phi(a)^\top \boldsymbol{\theta}\}$). In App. D.1, we show that if $a_1, a_2, \ldots, a_T$ are the actions selected by our CMM-UCB algorithm, then

$$\sum_{t=1}^{T} \Delta(a_t) \leq \sum_{t=1}^{T} \max_{\theta \in \Theta_{t-1}} \{\phi(a_t)^\top \boldsymbol{\theta}\} - \min_{\theta \in \Theta_{t-1}} \{\phi(a_t)^\top \boldsymbol{\theta}\}. \tag{10}$$

This gives a data-dependent cumulative regret bound for CMM-UCB. AMM-UCB has a similar data-dependent cumulative regret bound. In App. D.1, we show that if $a_1, a_2, \ldots, a_T$ are the actions selected by our AMM-UCB algorithm, then

$$\sum_{t=1}^{T} \Delta(a_t) \leq \sum_{t=1}^{T} \mathrm{AUCB}_{\Theta_{t-1}}(a_t) - \mathrm{ALCB}_{\Theta_{t-1}}(a_t), \tag{11}$$

where $\text{AUCB}_{\Theta_t}(a)$ is the right-hand-side of (8) and $\text{ALCB}_{\Theta_t}(a)$ is the equivalent analytic LCB. Since, the analytic UCB/LCB is an upper/lower bound on the numerical UCB/LCB, the bound in Equation (11) also holds for the actions selected by CMM-UCB. By substituting in the expressions for the analytic UCB/LCBs, we obtain the following data-dependent cumulative regret bound for CMM-UCB and AMM-UCB.

**Theorem 7.5.** *Suppose that assumptions 7.1-7.2 hold. For any adaptive sequence of mixture distributions $P_t = \mathcal{N}(\boldsymbol{\mu}_t, \boldsymbol{T}_t)$, any $\delta \in (0,1)$, any $\alpha > 0$ and all $T \geq 1$, with probability at least $1 - \delta$, the cumulative regret of both CMM-UCB and AMM-UCB is bounded by*

$$\Delta_{1:T} \leq \sum_{t=1}^{T} 2R_{\text{AMM},t-1} \sqrt{\phi(a_t)^\top \left(\Phi_{t-1}^\top \Phi_{t-1} + \alpha \mathbb{1}\right)^{-1} \phi(a_t)}.$$

A proof is given in App. D.1. This regret bound tells us that if we choose an adaptive sequence of mixture distributions $P_t = \mathcal{N}(\boldsymbol{\mu}_t, \boldsymbol{T}_t)$, such that the radii $R_{\text{AMM},t}$ are small, then we can expect to have small cumulative regret.

### 7.2 Data-Independent Regret Bounds

We now state a data-independent cumulative regret bound for the special case when the adaptive sequence of mixture distributions is $P_t = \mathcal{N}(\mathbf{0}, c\Phi_t\Phi_t^\top)$, and $\alpha = \sigma^2/c$. These mixture distributions are scaled versions of the standard mixture distributions described in Sec. 6.4.

**Theorem 7.6.** *Suppose that assumptions 7.1-7.4 hold. If for any $c > 0$, the sequence of mixture distributions is $P_t = \mathcal{N}(\mathbf{0}, c\Phi_t\Phi_t^\top)$, then for all $T \geq 1$, with probability at least $1 - \delta$, the cumulative regret of both CMM-UCB and AMM-UCB (with $\alpha = \sigma^2/c$) is bounded by*

$$\Delta_{1:T} \leq \frac{2}{\sqrt{\ln 2}} \max \left\{ C, \sigma\sqrt{d \ln\left(1 + \frac{cL^2 T}{\sigma^2 d}\right)} + \frac{B^2}{c} + 2\ln\frac{1}{\delta} \right\} \sqrt{dT \ln\left(1 + \frac{cL^2 T}{\sigma^2 d}\right)}$$

$$\leq \mathcal{O}(d\sqrt{T}\ln(T)).$$

*Proof sketch.* Choosing $P_t = \mathcal{N}(\mathbf{0}, c\Phi_t\Phi_t^\top)$ and $\alpha = \sigma^2/c$ means that the two quadratic terms in $R_{\text{AMM},t}^2$ cancel out. We then find a data-independent upper bound for the log det term. Following Abbasi-Yadkori et al. (2011), the sum of the norms $\sqrt{\phi(a_t)^\top(\Phi_{t-1}^\top \Phi_{t-1} + \alpha \mathbb{1})^{-1}\phi(a_t)}$ is upper bounded using an elliptical potential lemma. The result is the data-independent regret bound in the statement of the theorem. □

In App. D.2, we give a proof of this special case. In addition, we also treat a more general case when the sequence of mixture distributions is $P_t = \mathcal{N}(\Phi_t\boldsymbol{\theta}_0, \sigma_0^2\Phi_t\Phi_t^\top)$ and $\alpha$ is any positive number. Using Eq. (5) with $\boldsymbol{\mu}_t = \Phi_t\boldsymbol{\theta}_0$ and $\boldsymbol{T}_t = \sigma_0^2\Phi_t\Phi_t^\top$, one can interpret $\boldsymbol{\theta}_0$ as a guess for $\boldsymbol{\theta}^*$ and $\sigma_0$ as a guess for the distance between the reward vector $\boldsymbol{r}_t$ and the prediction $\Phi_t\boldsymbol{\theta}_0$.

Focusing on the dependence on $d$ and $T$, this regret bound (and the more general one in App. D.2) is at most $\mathcal{O}(d\sqrt{T}\ln(T))$, which matches OFUL and is minimax optimal up to the $\ln(T)$ factor. If (upper bounds on) $\sigma^2$, $B$, $L$ and $C$ are known, then we can evaluate this cumulative regret bound before running the algorithm.

## 8 Experiments

In all our experiments, we set $\delta = 0.01$. When using our analytic UCBs (Thm. 6.1), we always choose $\alpha = \sigma^2$. Unless stated otherwise, we use the standard mixture distributions $P_t = \mathcal{N}(\mathbf{0}, \Phi_t\Phi_t^\top)$.

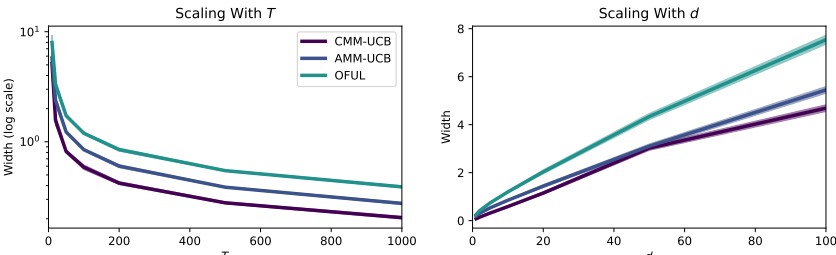

Figure 2: Average confidence bound width for different data set sizes $T$ and feature dimensions $d$.

## 8.1 Upper and Lower Confidence Bounds

**Compared Methods.** We evaluate the following upper/lower confidence bounds: (a) *CMM-UCB:* our numerical UCBs/LCBs from Sect. 6.2; (b) *AMM-UCB:* our analytic UCBs/LCBs from Thm. 6.1; (c) *OFUL:* the UCBs/LCBs used by the OFUL algorithm (Abbasi-Yadkori et al., 2011); (d) *Bayes:* a Bayesian credible interval constructed from the Bayesian posterior for linear regression with a Gaussian prior and likelihood (see App. E.1 for details).

**Experimental Setup.** We conduct experiments on randomly generated linear functions of the form $f(\boldsymbol{x}) = \phi(\boldsymbol{x})^\top \boldsymbol{\theta}^*$, with inputs $\boldsymbol{x} \in \mathbb{R}^{d_\mathcal{X}}$ and $\boldsymbol{\theta}^* \in \mathbb{R}^d$, the latter drawn from a standard Gaussian distribution and if necessary scaled down to $\|\boldsymbol{\theta}^*\|_2 \leq 10 =: B$. For the feature map $\phi : \mathbb{R}^{d_\mathcal{X}} \to \mathbb{R}^d$, we use Random Fourier Features (cf. Algorithm 1 of (Rahimi & Recht, 2007)). We investigate the properties of upper and lower confidence bounds constructed from random data sets $\{(\boldsymbol{x}_t, y_t)\}_{t=1}^T$, where $y_t = \phi(\boldsymbol{x}_t)^\top \boldsymbol{\theta}^* + \epsilon_t$, $\epsilon_t \sim \mathcal{N}(0, \sigma^2)$ and $\sigma = 0.1$.

**UCB/LCB Tightness.** Fig. 1 shows the data $\{(\boldsymbol{x}_t, y_t)\}_{t=1}^T$ and the UCBs/LCBs of CMM-UCB, AMM-UCB and OFUL for a randomly generated linear function with $d_\mathcal{X} = 1$ and $d = 20$. In this example, the confidence bounds of CMM-UCB are slightly tighter than those of AMM-UCB, which are considerably tighter than those of OFUL. Next, we investigate the tightness of the confidence bounds for functions with higher dimensional inputs ($d_\mathcal{X} = 10$), a range of data set sizes ($T \in \{1, 2, 5, 10, 20, 50, 100, 200, 500, 1000\}$) and a range feature vector dimensions ($d \in \{1, 2, 5, 10, 20, 50, 100\}$). For each $T$ and $d$, we sample a random feature map $\phi$ and weight vector $\boldsymbol{\theta}$ of appropriate size. Then, we sample random training data $\{(\boldsymbol{x}_t, y_t)\}_{t=1}^T$ and random test points $\{\boldsymbol{x}_t'\}_{t=1}^{100}$, where $\boldsymbol{x}_k$ and $\boldsymbol{x}_t'$ are drawn uniformly from the $d_\mathcal{X}$-dimensional unit hypercube. Finally, we use the training data to construct confidence bounds with each method and calculate the average width (UCB minus LCB) at the test points. Fig. 2 shows the average width of the CMM-UCB, AMM-UCB and OFUL confidence bounds with: $d = 10$ and varying $T$ (left), and $T = 100$ and varying $d$ (right). We observe the same pattern at every $d$ and $T$: CMM-UCB produces the tightest confidence bounds followed by AMM-UCB and then OFUL.

**Effect of the Mixture Distributions.** Fig. 4 in App. E.1 shows the confidence bounds of CMM-UCB and a Bayesian credible interval for different choices of the mixture distributions/prior $P_t$. For a well-specified prior, either uninformative or informative, the Bayesian credible interval is slightly tighter than the confidence bounds of CMM-UCB. For a misspecified prior, the confidence bounds of CMM-UCB become looser whereas the Bayesian credible interval becomes wrong (not containing the ground-truth function). Here, misspecification refers solely to Bayesian prior misspecification. In Figs. 5 and 6 in App. E.2 we show that adaptive choices of the mixture distributions $P_t$ can lead to smaller confidence bounds and smaller cumulative regret in linear bandit problems.

## 8.2 Linear Bandits

**Compared Methods.** We compare: (a) *CMM-UCB:* cf. Sec. 6.2; (b) *AMM-UCB:* cf. Sec. 6.3; (c) *OFUL:* the OFUL algorithm (Abbasi-Yadkori et al., 2011), with regularisation parameter $\lambda = \alpha = \sigma^2$; (d) *IDS:* the frequentist Information Directed Sampling (IDS) algorithm (Kirschner & Krause, 2018), specifically the DIDS-F version; (e) *Freq-TS:* Thompson Sampling with posterior covariance inflation (Agrawal & Goyal, 2013), which we call Frequentist Thompson Sampling.

Table 1: Average test accuracy and maximum test accuracy of our UCB algorithms and OFUL, IDS and Freq-TS in the SVM hyperparameter tuning problems after $T = 500$ rounds (100 repetitions).

| | Raisin | | Maternal | | Banknotes | |
|---|---|---|---|---|---|---|
| | Mean Acc | Max Acc | Mean Acc | Max Acc | Mean Acc | Max Acc |
| CMM-UCB (Ours) | **0.818** $\pm$ 0.018 | **0.893** $\pm$ 0.019 | **0.744** $\pm$ 0.020 | **0.829** $\pm$ 0.023 | **0.954** $\pm$ 0.005 | **1.000** $\pm$ 0.000 |
| AMM-UCB (Ours) | 0.800 $\pm$ 0.017 | 0.892 $\pm$ 0.020 | 0.736 $\pm$ 0.020 | **0.829** $\pm$ 0.023 | 0.948 $\pm$ 0.005 | **1.000** $\pm$ 0.000 |
| OFUL | 0.764 $\pm$ 0.019 | 0.891 $\pm$ 0.019 | 0.722 $\pm$ 0.019 | 0.827 $\pm$ 0.022 | 0.929 $\pm$ 0.006 | **1.000** $\pm$ 0.000 |
| IDS | 0.706 $\pm$ 0.048 | 0.891 $\pm$ 0.020 | 0.714 $\pm$ 0.019 | 0.827 $\pm$ 0.024 | 0.926 $\pm$ 0.007 | **1.000** $\pm$ 0.000 |
| Freq-TS | 0.527 $\pm$ 0.022 | 0.884 $\pm$ 0.019 | 0.616 $\pm$ 0.018 | 0.823 $\pm$ 0.022 | 0.808 $\pm$ 0.012 | **1.000** $\pm$ 0.000 |

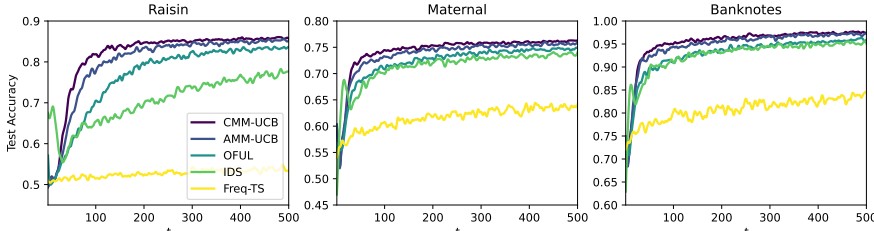

Figure 3: The smoothed per-round expected reward (test accuracy) of our UCB algorithms compared with OFUL, IDS and Freq-TS in the SVM hyperparameter tuning experiments on three datasets. Shown is the mean reward over 100 runs of each experiment, after Gaussian kernel smoothing.

**Experimental Setup.** We investigate whether our tighter upper confidence bounds translate to better UCB algorithms. We use each linear bandit algorithm to optimise the hyperparameters of a kernel Support Vector Machine (SVM) for three classification data sets from the UCI Machine Learning Repository (Dua & Graff, 2017): Raisin (Cinar et al., 2020), Maternal (Ahmed et al., 2020), and Banknotes. The expected reward function $f^*(a)$ is the average test set accuracy of a kernel SVM trained using an ARD RBF kernel with hyperparameters $a = (C, \boldsymbol{\gamma})$, with $C$ the regularisation and $\boldsymbol{\gamma}$ the length-scales. The observed reward $r_t$ is the validation set accuracy at $a_t$. The feature map $\phi$ is a neural network layer with 20 outputs and random weights. We choose $\sigma = 0.05$ for the sub-Gaussian parameter and $B = 10$, i.e. we assume that $f^*(a) \approx \phi(a)^\top \boldsymbol{\theta}^*$ for some $\|\boldsymbol{\theta}^*\|_2 \leq 10$.

**Results.** Fig. 3 compares the average test accuracy (expected reward) obtained by each bandit algorithm for each data set and at each round $t = 1, \ldots, 500$. Our CMM-UCB and AMM-UCB methods outperform all other methods. From the reward curves of CMM-UCB, AMM-UCB and OFUL, we observe that CMM-UCB outperforms AMM-UCB, which outperforms OFUL. Therefore, we can conclude that our tighter confidence bounds (compared to OFUL's) lead to UCB algorithms with improved performance.

# 9   Conclusion

In this paper, we developed a novel tail bound for adaptive martingale mixtures and showed that it can be used to construct tighter confidence sequences for linear bandits. We proved that our CMM-UCB and AMM-UCB algorithms match the worst-case cumulative regret of OFUL, and we found that our tighter confidence sequences allowed CMM-UCB and AMM-UCB to achieve greater average and maximum reward in several hyperparameter tuning problems.

A limitation of our algorithms is that they assume a linear expected reward function, which may not always be a realistic assumption for real-world bandit problems. Our general-purpose tail bound in Thm. 5.1 already allows one to derive confidence sequences for non-linear reward functions by simply choosing $Z_t(f_t) = (f_t - f^*(a_t))\epsilon_t$, where $f^*$ is the non-linear reward function. In the case where $f^*$ lies in a reproducing kernel Hilbert space, the corresponding UCB is still the solution of a convex program. The main challenge in this setting is that the regret bound must now depend on quantities like the effective dimension or the maximum information gain of the kernel, since the dimension $d$ of the feature vectors is $d = \infty$ for interesting kernels.

We believe that further investigation into the degree to which adaptive mixture distributions can lead to improved performance and regret bounds (see App. E.2) is another exciting topic for future work.

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

# A  Proof of the General-Purpose Tail Bound for Adaptive Martingale Mixtures

## A.1  Verifying Martingale Properties

First, we recall the definition of $(M_t(\boldsymbol{f}_t, \boldsymbol{\lambda}_t)|t \in \mathbb{N})$ in Eq. (3). We are given a filtration $(\mathcal{H}_t|t \in \mathbb{N})$, a sequence of adapted random functions $(Z_t : \mathbb{R} \to \mathbb{R}|t \in \mathbb{N})$, and a sequence of predictable random variables $(\lambda_t|t \in \mathbb{N})$.

A filtration is an increasing sequence of $\sigma$-algebras $\mathcal{H}_0 \subseteq \mathcal{H}_1 \subseteq \mathcal{H}_2 \cdots$. Each $\sigma$-algebra $\mathcal{H}_t$ represents the information available at time $t$. $(Z_t : \mathbb{R} \to \mathbb{R}|t \in \mathbb{N})$ being a sequence of *adapted* (to the filtration $(\mathcal{H}_t|t \in \mathbb{N})$) functions means that, when conditioned on $\mathcal{H}_t$, $Z_t$ is no longer random. $(\lambda_t|t \in \mathbb{N})$ being a sequence of *predictable* random variables means that, when conditioned on $\mathcal{H}_{t-1}$, $\lambda_t$ is no longer random.

For a sequence of real numbers $(f_t : t \in \mathbb{N})$, we define

$$M_t(\boldsymbol{f}_t, \boldsymbol{\lambda}_t) = \exp\left(\sum_{k=1}^t \lambda_k Z_k(f_k) - \psi_k(f_k, \lambda_k)\right),$$

where $\psi_t(f_t, \lambda_t)$ is the conditional cumulant generating function

$$\psi_t(f_t, \lambda_t) := \ln\left(\mathbb{E}\left[\exp(\lambda_t Z_t(f_t))|\mathcal{H}_{t-1}\right]\right).$$

**Lemma A.1.** *For any sequence of real numbers $(f_t|t \in \mathbb{N})$, $(M_t(\boldsymbol{f}_t, \boldsymbol{\lambda}_t)|t \in \mathbb{N})$ is a martingale and $\mathbb{E}[M_t(\boldsymbol{f}_t, \boldsymbol{\lambda}_t)] = 1$ for all $t \in \mathbb{N}$.*

*Proof.* For $t = 1$, we have

$$
\begin{aligned}
\mathbb{E}[M_1(\boldsymbol{f}_1, \boldsymbol{\lambda}_1)|\mathcal{H}_0] &= \mathbb{E}\left[\exp(\lambda_1 Z_1(f_1) - \psi_1(f_1, \lambda_1))|\mathcal{H}_0\right] \\
&= \mathbb{E}\left[\exp(\lambda_1 Z_1(f_1))|\mathcal{H}_0\right] / \exp(\psi_1(f_1, \lambda_1)) \\
&= \exp(\psi_1(f_1, \lambda_1)) / \exp(\psi_1(f_1, \lambda_1)) \\
&= 1.
\end{aligned}
$$

Using the tower rule of conditional expectation, we also have

$$\mathbb{E}[M_1(\boldsymbol{f}_1, \boldsymbol{\lambda}_1)] = \mathbb{E}[\mathbb{E}[M_1(\boldsymbol{f}_1, \boldsymbol{\lambda}_1)|\mathcal{H}_0]] = 1.$$

Now, we verify the martingale property. For any $t \geq 2$, we have

$$
\begin{aligned}
\mathbb{E}\left[M_t(\boldsymbol{f}_t, \boldsymbol{\lambda}_t)|\mathcal{H}_{t-1}\right] &= \mathbb{E}\left[\exp\left(\sum_{k=1}^t \lambda_k Z_k(f_k) - \psi_k(f_k, \lambda_k)\right)\bigg|\mathcal{H}_{t-1}\right] \\
&= \exp\left(\sum_{k=1}^{t-1} \lambda_k Z_k(f_k) - \psi_k(f_k, \lambda_k)\right) \mathbb{E}\left[\exp\left(\lambda_t Z_t(f_t) - \psi_t(f_t, \lambda_t)\right)|\mathcal{H}_{t-1}\right] \\
&= \exp\left(\sum_{k=1}^{t-1} \lambda_k Z_k(f_k) - \psi_k(f_k, \lambda_k)\right) \\
&= M_{t-1}(\boldsymbol{f}_{t-1}, \boldsymbol{\lambda}_{t-1}).
\end{aligned}
$$

Using the tower rule again, we have for any $t \geq 2$

$$\mathbb{E}[M_t(\boldsymbol{f}_t, \boldsymbol{\lambda}_t)] = \mathbb{E}[\mathbb{E}[M_t(\boldsymbol{f}_t, \boldsymbol{\lambda}_t)|\mathcal{H}_{t-1}]] = \mathbb{E}[M_{t-1}(\boldsymbol{f}_{t-1}, \boldsymbol{\lambda}_{t-1})].$$

Therefore, we have

$$\mathbb{E}[M_t(\boldsymbol{f}_t, \boldsymbol{\lambda}_t)] = \mathbb{E}[M_{t-1}(\boldsymbol{f}_{t-1}, \boldsymbol{\lambda}_{t-1})] = \cdots = \mathbb{E}[M_1(\boldsymbol{f}_1, \boldsymbol{\lambda}_1)] = 1.$$

$\square$

**Lemma A.2.** *For any adaptive sequence of mixture distributions* $(P_t | t \in \mathbb{N})$, $(\mathbb{E}_{\boldsymbol{f}_t \sim P_t}[M_t(\boldsymbol{f}_t, \boldsymbol{\lambda}_t)] | t \in \mathbb{N})$ *is a martingale and* $\mathbb{E}[\mathbb{E}_{\boldsymbol{f}_t \sim P_t}[M_t(\boldsymbol{f}_t, \boldsymbol{\lambda}_t)]] = 1$ *for all* $t \in \mathbb{N}$.

*Proof.* For any $t \geq 1$, since $M_t(\boldsymbol{f}_t, \boldsymbol{\lambda}_t)$ is non-negative and $P_t$ is $\mathcal{H}_{t-1}$-measurable, Tonelli's theorem implies

$$\mathbb{E}\left[\mathbb{E}_{\boldsymbol{f}_t \sim P_t}[M_t(\boldsymbol{f}_t, \boldsymbol{\lambda}_t)] | \mathcal{H}_{t-1}\right] = \mathbb{E}_{\boldsymbol{f}_t \sim P_t}\left[\mathbb{E}[M_t(\boldsymbol{f}_t, \boldsymbol{\lambda}_t) | \mathcal{H}_{t-1}]\right].$$

The requirement that the distributions $P_1, P_2, \ldots$ have coinciding marginals, i.e. $\int P_t(\boldsymbol{f}_t) \mathrm{d}f_t = P_{t-1}(\boldsymbol{f}_{t-1})$, means that for all $t \geq 2$

$$\mathbb{E}_{\boldsymbol{f}_t \sim P_t}[M_{t-1}(\boldsymbol{f}_{t-1}, \boldsymbol{\lambda}_{t-1})] = \mathbb{E}_{\boldsymbol{f}_{t-1} \sim P_{t-1}}[M_{t-1}(\boldsymbol{f}_{t-1}, \boldsymbol{\lambda}_{t-1})].$$

Using these two results, and the fact that $(M_t(\boldsymbol{f}_t, \boldsymbol{\lambda}_t) | t \in \mathbb{N})$ is a martingale for any sequence $(f_t | t \in \mathbb{N})$, we now verify that the martingale mixture $(\mathbb{E}_{\boldsymbol{f}_t \sim P_t}[M_t(\boldsymbol{f}_t, \boldsymbol{\lambda}_t)] | t \in \mathbb{N})$ is a martingale with expected value 1. For $t = 1$, we have

$$
\begin{aligned}
\mathbb{E}\left[\mathbb{E}_{\boldsymbol{f}_1 \sim P_1}[M_1(\boldsymbol{f}_1, \boldsymbol{\lambda}_1)] | \mathcal{H}_0\right] &= \mathbb{E}_{\boldsymbol{f}_1 \sim P_1}\left[\mathbb{E}[M_1(\boldsymbol{f}_1, \boldsymbol{\lambda}_1) | \mathcal{H}_0]\right] \\
&= \mathbb{E}_{\boldsymbol{f}_1 \sim P_1}[1] \\
&= 1.
\end{aligned}
$$

Using the tower rule as before, this also means that $\mathbb{E}\left[\mathbb{E}_{\boldsymbol{f}_1 \sim P_1}[M_1(\boldsymbol{f}_1, \boldsymbol{\lambda}_1)]\right] = 1$. For any $t \geq 2$, we have

$$
\begin{aligned}
\mathbb{E}\left[\mathbb{E}_{\boldsymbol{f}_t \sim P_t}[M_t(\boldsymbol{f}_t, \boldsymbol{\lambda}_t)] | \mathcal{H}_{t-1}\right] &= \mathbb{E}_{\boldsymbol{f}_t \sim P_t}\left[\mathbb{E}[M_t(\boldsymbol{f}_t, \boldsymbol{\lambda}_t) | \mathcal{H}_{t-1}]\right] \\
&= \mathbb{E}_{\boldsymbol{f}_t \sim P_t}\left[M_{t-1}(\boldsymbol{f}_{t-1}, \boldsymbol{\lambda}_{t-1})\right] \\
&= \mathbb{E}_{\boldsymbol{f}_{t-1} \sim P_{t-1}}\left[M_{t-1}(\boldsymbol{f}_{t-1}, \boldsymbol{\lambda}_{t-1})\right].
\end{aligned}
$$

Using the tower rule one more time, we have for any $t \geq 2$

$$\mathbb{E}[\mathbb{E}_{\boldsymbol{f}_t \sim P_t}[M_t(\boldsymbol{f}_t, \boldsymbol{\lambda}_t)]] = \mathbb{E}[\mathbb{E}[\mathbb{E}_{\boldsymbol{f}_t \sim P_t}[M_t(\boldsymbol{f}_t, \boldsymbol{\lambda}_t)] | \mathcal{H}_{t-1}]] = \mathbb{E}[\mathbb{E}_{\boldsymbol{f}_{t-1} \sim P_{t-1}}[M_{t-1}(\boldsymbol{f}_{t-1}, \boldsymbol{\lambda}_{t-1})]].$$

Therefore, we have

$$\mathbb{E}[\mathbb{E}_{\boldsymbol{f}_t \sim P_t}[M_t(\boldsymbol{f}_t, \boldsymbol{\lambda}_t)]] = \mathbb{E}[\mathbb{E}_{\boldsymbol{f}_1 \sim P_1}[M_1(\boldsymbol{f}_1, \boldsymbol{\lambda}_1)]] = 1.$$

$\square$

## A.2 Proof of Theorem 5.1

To prove Thm. 5.1, we use Ville's inequality for non-negative supermartingales (Ville, 1939), which can be thought of as a time-uniform version of Markov's inequality. Instead of the martingale property $\mathbb{E}[M_t | \mathcal{H}_{t-1}] = M_{t-1}$, a supermartingale satisfies $\mathbb{E}[M_t | \mathcal{H}_{t-1}] \leq M_{t-1}$, which means that any martingale is also a supermartingale. Therefore, Ville's inequality for non-negative supermartingales also holds for the non-negative martingale in Lemma A.2.

**Lemma A.3** (Ville's inequality for non-negative supermartingales (Ville, 1939)). *Let* $(M_t | t \in \mathbb{N})$ *be a non-negative supermartingale with respect to the filtration* $(\mathcal{H}_t | t \in \mathbb{N})$, *which satisfies* $M_0 = 1$. *For any* $\delta \in (0, 1]$, *it holds with probability at least* $1 - \delta$:

$$\forall t \geq 1 : \quad M_t \leq 1/\delta.$$

*Proof of Thm. 5.1.* We choose an arbitrary $\delta \in (0, 1]$. From Lemma A.2, for any adaptive sequence of mixture distributions $(P_t | t \in \mathbb{N})$, $(\mathbb{E}_{\boldsymbol{f}_t \sim P_t}[M_t(\boldsymbol{f}_t, \boldsymbol{\lambda}_t)] | t \in \mathbb{N})$ is a martingale and $\mathbb{E}[\mathbb{E}_{\boldsymbol{f}_t \sim P_t}[M_t(\boldsymbol{f}_t, \boldsymbol{\lambda}_t)]] = 1$. In addition, $(\mathbb{E}_{\boldsymbol{f}_t \sim P_t}[M_t(\boldsymbol{f}_t, \boldsymbol{\lambda}_t)] | t \in \mathbb{N})$ is clearly non-negative. Therefore, using Lemma A.3, with probability at least $1 - \delta$

$$\forall t \geq 1, \quad \mathbb{E}_{\boldsymbol{f}_t \sim P_t}[M_t(\boldsymbol{f}_t, \boldsymbol{\lambda}_t)] \leq 1/\delta.$$

Taking the logarithm of both sides yields the statement of Thm. 5.1. $\square$

# B    Closed-Form Gaussian Integration

Here, we calculate the integral in the inequality (see beginning of Sec. 5.2):

$$\mathop{\mathbb{E}}_{\boldsymbol{f}_t \sim \mathcal{N}(\boldsymbol{\mu}_t, \boldsymbol{T}_t)} \left[ \exp\left\{ \sum_{k=1}^{t} \lambda_k (f_k - \phi(a_k)^\top \boldsymbol{\theta}^*)(r_k - \phi(a_k)^\top \boldsymbol{\theta}^*) - \frac{\sigma^2}{2} \lambda_k^2 (f_k - \phi(a_k)^\top \boldsymbol{\theta}^*)^2 \right\} \right] \leq \frac{1}{\delta}. \tag{12}$$

First, we rearrange the integrand into a more convenient form. For every $k$, using $r_k = \phi(a_k)^\top \boldsymbol{\theta}^* + \epsilon_k$, we have

$$
\begin{aligned}
(\phi(a_k)^\top \boldsymbol{\theta}^* - r_k)^2 - (f_k - r_k)^2 &= \epsilon_k^2 - (f_k - \phi(a_k)^\top \boldsymbol{\theta}^* - \epsilon_k)^2 \\
&= \epsilon_k^2 - (f_k - \phi(a_k)^\top \boldsymbol{\theta}^*)^2 + 2(f_k - \phi(a_k)^\top \boldsymbol{\theta}^*)\epsilon_k - \epsilon_k^2 \\
&= 2(f_k - \phi(a_k)^\top \boldsymbol{\theta}^*)(r_k - \phi(a_k)^\top \boldsymbol{\theta}^*) - (f_k - \phi(a_k)^\top \boldsymbol{\theta}^*)^2.
\end{aligned}
$$

Therefore, we have that

$$
\begin{aligned}
&\lambda_k (f_k - \phi(a_k)^\top \boldsymbol{\theta}^*)(r_k - \phi(a_k)^\top \boldsymbol{\theta}^*) - \frac{\sigma^2}{2} \lambda_k^2 (f_k - \phi(a_k)^\top \boldsymbol{\theta}^*)^2 \\
&= \frac{\lambda_k}{2}(\phi(a_k)^\top \boldsymbol{\theta}^* - r_k)^2 - \frac{\lambda_k}{2}(f_k - r_k)^2 + \frac{1}{2}(\lambda_k - \sigma^2 \lambda_k^2)(f_k - \phi(a_k)^\top \boldsymbol{\theta}^*)^2.
\end{aligned}
$$

Equation (12) can now be re-written as

$$\mathop{\mathbb{E}}_{\boldsymbol{f}_t \sim \mathcal{N}(\boldsymbol{\mu}_t, \boldsymbol{T}_t)} \left[ \exp\left\{ \sum_{k=1}^{t} \frac{\lambda_k}{2}(\phi(a_k)^\top \boldsymbol{\theta}^* - r_k)^2 - \frac{\lambda_k}{2}(f_k - r_k)^2 + \frac{1}{2}(\lambda_k - \sigma^2 \lambda_k^2)(f_k - \phi(a_k)^\top \boldsymbol{\theta}^*)^2 \right\} \right] \leq \frac{1}{\delta}. \tag{13}$$

In the special case where $\lambda_t \equiv 1/\sigma^2$, we have $\lambda_k - \sigma^2 \lambda_k^2 = 0$, which means that $\frac{1}{2}(\lambda_k - \sigma^2 \lambda_k^2)(f_k - \phi(a_k)^\top \boldsymbol{\theta}^*)^2$ disappears. In addition, $(\lambda_k/2)(\phi(a_k)^\top \boldsymbol{\theta}^* - r_k)^2$ does not depend on $f_k$, so it can be moved outside the integral.

## B.1    General $\lambda_t$

Let $\boldsymbol{\Lambda}_t$ be the $t \times t$ diagonal matrix with diagonal elements $\lambda_1, \lambda_2, \ldots, \lambda_t$. Starting from (13), taking the logarithm of both sides, rearranging terms and then writing everything in matrix notation, we arrive at

$$(\Phi_t \boldsymbol{\theta}^* - \boldsymbol{r}_t) \boldsymbol{\Lambda}_t (\Phi_t \boldsymbol{\theta}^* - \boldsymbol{r}_t) \leq 2 \ln(1/\delta) \tag{14}$$

$$- 2 \ln \left( \mathop{\mathbb{E}}_{\boldsymbol{f}_t \sim \mathcal{N}(\boldsymbol{\mu}_t, \boldsymbol{T}_t)} \left[ \exp\left( -\frac{1}{2}(\boldsymbol{f}_t - \boldsymbol{r}_t)^\top \boldsymbol{\Lambda}_t (\boldsymbol{f}_t - \boldsymbol{r}_t) + \frac{1}{2}(\boldsymbol{f}_t - \Phi_t \boldsymbol{\theta}^*)^\top \left( \boldsymbol{\Lambda}_t - \sigma^2 \boldsymbol{\Lambda}_t^2 \right)(\boldsymbol{f}_t - \Phi_t \boldsymbol{\theta}^*) \right) \right] \right)$$

The expected value inside the logarithm can be re-written as

$$\frac{1}{\sqrt{(2\pi)^t \det(\boldsymbol{T}_t)}} \int \exp\left( -\frac{1}{2}(\boldsymbol{f}_t - \boldsymbol{\mu}_t)^\top \boldsymbol{T}_t^{-1}(\boldsymbol{f}_t - \boldsymbol{\mu}_t) - \frac{1}{2}(\boldsymbol{f}_t - \boldsymbol{r}_t)^\top \boldsymbol{\Lambda}_t (\boldsymbol{f}_t - \boldsymbol{r}_t) \right. \tag{15}$$

$$\left. + \frac{1}{2}(\boldsymbol{f}_t - \Phi_t \boldsymbol{\theta}^*)^\top \left( \boldsymbol{\Lambda}_t - \sigma^2 \boldsymbol{\Lambda}_t^2 \right)(\boldsymbol{f}_t - \Phi_t \boldsymbol{\theta}^*) \right) \mathrm{d}\boldsymbol{f}_t.$$

We will calculate the integral by "completing the square", i.e. rewriting the exponent in the form $-\frac{1}{2}(\boldsymbol{f}_t - \boldsymbol{b})^\top \boldsymbol{A}(\boldsymbol{f}_t - \boldsymbol{b}) + c$, to recover the integral of a Gaussian density function. For a symmetric matrix $\boldsymbol{A}$, we have

$$-\frac{1}{2}(\boldsymbol{f}_t - \boldsymbol{b})^\top \boldsymbol{A}(\boldsymbol{f}_t - \boldsymbol{b}) + c = -\frac{1}{2}\boldsymbol{f}_t^\top \boldsymbol{A}\boldsymbol{f}_t + \boldsymbol{b}^\top \boldsymbol{A}\boldsymbol{f}_t - \frac{1}{2}\boldsymbol{b}^\top \boldsymbol{A}\boldsymbol{b} + c.$$

We also have

$$-\frac{1}{2}(\boldsymbol{f}_t - \boldsymbol{\mu}_t)^\top \boldsymbol{T}_t^{-1}(\boldsymbol{f}_t - \boldsymbol{\mu}_t) = -\frac{1}{2}\boldsymbol{f}_t^\top \boldsymbol{T}_t^{-1}\boldsymbol{f}_t + \boldsymbol{\mu}_t^\top \boldsymbol{T}_t^{-1}\boldsymbol{f}_t - \frac{1}{2}\boldsymbol{\mu}_t^\top \boldsymbol{T}_t^{-1}\boldsymbol{\mu}_t$$

$$-\frac{1}{2}(\boldsymbol{f}_t - \boldsymbol{r}_t)^\top \boldsymbol{\Lambda}_t(\boldsymbol{f}_t - \boldsymbol{r}_t) = -\frac{1}{2}\boldsymbol{f}_t^\top \boldsymbol{\Lambda}_t \boldsymbol{f}_t + \boldsymbol{r}_t^\top \boldsymbol{\Lambda}_t \boldsymbol{f}_t - \frac{1}{2}\boldsymbol{r}_t^\top \boldsymbol{\Lambda}_t \boldsymbol{r}_t$$

$$\frac{1}{2}(\boldsymbol{f}_t - \Phi_t \boldsymbol{\theta}^*)^\top \left(\boldsymbol{\Lambda}_t - \sigma^2 \boldsymbol{\Lambda}_t^2\right)(\boldsymbol{f}_t - \Phi_t \boldsymbol{\theta}^*) = \frac{1}{2}\boldsymbol{f}_t^\top \left(\boldsymbol{\Lambda}_t - \sigma^2 \boldsymbol{\Lambda}_t^2\right)\boldsymbol{f}_t - \boldsymbol{\theta}^{*\top}\Phi_t^\top \left(\boldsymbol{\Lambda}_t - \sigma^2 \boldsymbol{\Lambda}_t^2\right)\boldsymbol{f}_t$$
$$+ \frac{1}{2}\boldsymbol{\theta}^{*\top}\Phi_t^\top \left(\boldsymbol{\Lambda}_t - \sigma^2 \boldsymbol{\Lambda}_t^2\right)\Phi_t \boldsymbol{\theta}^*.$$

We now equate coefficients to find $\boldsymbol{A}$, $\boldsymbol{b}$ and $c$. We find that $\boldsymbol{A}$ is

$$\boldsymbol{A} = \boldsymbol{T}_t^{-1} + \sigma^2 \boldsymbol{\Lambda}_t^2.$$

Note that $\boldsymbol{A}$ is symmetric. We find that $\boldsymbol{b}$ is

$$\boldsymbol{b}^\top \boldsymbol{A} = \boldsymbol{\mu}_t^\top \boldsymbol{T}_t^{-1} + \boldsymbol{r}_t^\top \boldsymbol{\Lambda}_t - \boldsymbol{\theta}^{*\top}\Phi_t^\top \left(\boldsymbol{\Lambda}_t - \sigma^2 \boldsymbol{\Lambda}_t^2\right)$$
$$\implies \boldsymbol{Ab} = \boldsymbol{T}_t^{-1}\boldsymbol{\mu}_t + \boldsymbol{\Lambda}_t \boldsymbol{r}_t - \left(\boldsymbol{\Lambda}_t - \sigma^2 \boldsymbol{\Lambda}_t^2\right)\Phi_t \boldsymbol{\theta}^*$$
$$\implies \boldsymbol{b} = \left(\boldsymbol{T}_t^{-1} + \sigma^2 \boldsymbol{\Lambda}_t^2\right)^{-1}\left(\boldsymbol{T}_t^{-1}\boldsymbol{\mu}_t + \boldsymbol{\Lambda}_t \boldsymbol{r}_t - \left(\boldsymbol{\Lambda}_t - \sigma^2 \boldsymbol{\Lambda}_t^2\right)\Phi_t \boldsymbol{\theta}^*\right).$$

Finally, we find that $c$ is

$$c = \frac{1}{2}\boldsymbol{b}^\top \boldsymbol{Ab} - \frac{1}{2}\boldsymbol{\mu}_t^\top \boldsymbol{T}_t^{-1}\boldsymbol{\mu}_t - \frac{1}{2}\boldsymbol{r}_t^\top \boldsymbol{\Lambda}_t \boldsymbol{r}_t + \frac{1}{2}\boldsymbol{\theta}^{*\top}\Phi_t^\top \left(\boldsymbol{\Lambda}_t - \sigma^2 \boldsymbol{\Lambda}_t^2\right)\Phi_t \boldsymbol{\theta}^*$$

$$= \frac{1}{2}\left(\boldsymbol{T}_t^{-1}\boldsymbol{\mu}_t + \boldsymbol{\Lambda}_t \boldsymbol{r}_t - \left(\boldsymbol{\Lambda}_t - \sigma^2 \boldsymbol{\Lambda}_t^2\right)\Phi_t \boldsymbol{\theta}^*\right)^\top \left(\boldsymbol{T}_t^{-1} + \sigma^2 \boldsymbol{\Lambda}_t^2\right)^{-1}\left(\boldsymbol{T}_t^{-1}\boldsymbol{\mu}_t + \boldsymbol{\Lambda}_t \boldsymbol{r}_t - \left(\boldsymbol{\Lambda}_t - \sigma^2 \boldsymbol{\Lambda}_t^2\right)\Phi_t \boldsymbol{\theta}^*\right)$$

$$- \frac{1}{2}\boldsymbol{\mu}_t^\top \boldsymbol{T}_t^{-1}\boldsymbol{\mu}_t - \frac{1}{2}\boldsymbol{r}_t^\top \boldsymbol{\Lambda}_t \boldsymbol{r}_t + \frac{1}{2}\boldsymbol{\theta}^{*\top}\Phi_t^\top \left(\boldsymbol{\Lambda}_t - \sigma^2 \boldsymbol{\Lambda}_t^2\right)\Phi_t \boldsymbol{\theta}^*$$

Now, we can rewrite and calculate the integral in (15) as

$$\frac{\exp(c)}{\sqrt{(2\pi)^t \det(\boldsymbol{T}_t)}} \int \exp\left(-\frac{1}{2}(\boldsymbol{f}_t - \boldsymbol{b})^\top \boldsymbol{A}(\boldsymbol{f}_t - \boldsymbol{b})\right)\mathrm{d}\boldsymbol{f}_t = \frac{\exp(c)\sqrt{(2\pi)^t \det(\boldsymbol{A}^{-1})}}{\sqrt{(2\pi)^t \det(\boldsymbol{T}_t)}}$$

$$= \exp(c)\sqrt{\frac{\det(\boldsymbol{A}^{-1})}{\det(\boldsymbol{T}_t)}}$$

Substituting this into (14), we obtain the constraint

$$(\Phi_t \boldsymbol{\theta}^* - \boldsymbol{r}_t)^\top \boldsymbol{\Lambda}_t(\Phi_t \boldsymbol{\theta}^* - \boldsymbol{r}_t) \leq -2\ln\left(\exp(c)\sqrt{\frac{\det(\boldsymbol{A}^{-1})}{\det(\boldsymbol{T}_t)}}\right) + 2\ln(1/\delta)$$

$$= -2c + \ln\left(\det(\boldsymbol{AT}_t)\right) + 2\ln(1/\delta)$$

$$= -\left(\boldsymbol{T}_t^{-1}\boldsymbol{\mu}_t + \boldsymbol{\Lambda}_t \boldsymbol{r}_t - \left(\boldsymbol{\Lambda}_t - \sigma^2 \boldsymbol{\Lambda}_t^2\right)\Phi_t \boldsymbol{\theta}^*\right)^\top \left(\boldsymbol{T}_t^{-1} + \sigma^2 \boldsymbol{\Lambda}_t^2\right)^{-1}\left(\boldsymbol{T}_t^{-1}\boldsymbol{\mu}_t + \boldsymbol{\Lambda}_t \boldsymbol{r}_t - \left(\boldsymbol{\Lambda}_t - \sigma^2 \boldsymbol{\Lambda}_t^2\right)\Phi_t \boldsymbol{\theta}^*\right)$$

$$+ \boldsymbol{\mu}_t^\top \boldsymbol{T}_t^{-1}\boldsymbol{\mu}_t + \boldsymbol{r}_t^\top \boldsymbol{\Lambda}_t \boldsymbol{r}_t - \boldsymbol{\theta}^{*\top}\Phi_t^\top \left(\boldsymbol{\Lambda}_t - \sigma^2 \boldsymbol{\Lambda}_t^2\right)\Phi_t \boldsymbol{\theta}^* + \ln\left(\det(\mathbb{1} + \sigma^2 \boldsymbol{\Lambda}_t^2 \boldsymbol{T}_t)\right) + 2\ln(1/\delta)$$

Note that $\boldsymbol{\theta}^*$ appears on both the left-hand-side and right-hand-side of this inequality. However, when $\boldsymbol{\Lambda}_t - \sigma^2 \boldsymbol{\Lambda}_t^2$ is the zero matrix (e.g. when $\lambda_t \equiv 1/\sigma^2$), all the $\boldsymbol{\theta}^*$-dependent terms on the right-hand-side disappear.

## B.2 The Special Case $\lambda_t \equiv 1/\sigma^2$

Starting from (13), choosing $\lambda_t \equiv 1/\sigma^2$, taking the logarithm of both sides and then rearranging terms, we arrive at

$$\|\Phi_t \boldsymbol{\theta}^* - \boldsymbol{r}_t\|_2^2 \leq -2\sigma^2 \ln\left(\mathop{\mathbb{E}}_{\boldsymbol{f}_t \sim \mathcal{N}(\boldsymbol{\mu}_t, \boldsymbol{T}_t)}\left[\exp\left(-\frac{1}{2\sigma^2}(\boldsymbol{f}_t - \boldsymbol{r}_t)^\top(\boldsymbol{f}_t - \boldsymbol{r}_t)\right)\right]\right) + 2\sigma^2 \ln(1/\delta).$$
$$(16)$$

For any $t \times t$ covariance matrix $\boldsymbol{T}$, let $Z(\boldsymbol{T})$ denote the normalising constant of a Gaussian distribution with covariance $\boldsymbol{T}$, so

$$Z(\boldsymbol{T}) = \sqrt{(2\pi)^t \det(\boldsymbol{T})}.$$

For any $t$-dimensional vectors $\boldsymbol{x}$ and $\boldsymbol{\mu}$, and any $t \times t$ covariance matrix $\boldsymbol{T}$, let $p(\boldsymbol{x}|\boldsymbol{\mu}, \boldsymbol{T})$ denote the density function of a Gaussian distribution with mean $\boldsymbol{\mu}$ and covariance $\boldsymbol{T}$, evaluated at $\boldsymbol{x}$. This means that

$$p(\boldsymbol{x}|\boldsymbol{\mu}, \boldsymbol{T}) = \frac{1}{Z(\boldsymbol{T})} \exp\left( -\frac{1}{2}(\boldsymbol{x} - \boldsymbol{\mu})^\top \boldsymbol{T}^{-1}(\boldsymbol{x} - \boldsymbol{\mu}) \right).$$

We will use the product of Gaussians trick from (Petersen et al., 2008) (Section 8.1.8, Equation 371), which states

$$p(\boldsymbol{x}|\boldsymbol{\mu}_1, \boldsymbol{\Sigma}_1)p(\boldsymbol{x}|\boldsymbol{\mu}_2, \boldsymbol{\Sigma}_2) = p(\boldsymbol{\mu}_1|\boldsymbol{\mu}_2, \boldsymbol{\Sigma}_1 + \boldsymbol{\Sigma}_2)p(\boldsymbol{x}|\boldsymbol{\mu}_c, \boldsymbol{\Sigma}_c), \tag{17}$$

where

$$\boldsymbol{\mu}_c = \left(\boldsymbol{\Sigma}_1^{-1} + \boldsymbol{\Sigma}_2^{-1}\right)^{-1}\left(\boldsymbol{\Sigma}_1^{-1}\boldsymbol{\mu}_1 + \boldsymbol{\Sigma}_2^{-1}\boldsymbol{\mu}_2\right), \qquad \boldsymbol{\Sigma}_c = \left(\boldsymbol{\Sigma}_1^{-1} + \boldsymbol{\Sigma}_2^{-1}\right)^{-1}.$$

We have that

$$\mathbb{E}_{\boldsymbol{f}_t \sim \mathcal{N}(\boldsymbol{\mu}_t, \boldsymbol{T}_t)} \left[ \exp\left( -\frac{1}{2\sigma^2}(\boldsymbol{f}_t - \boldsymbol{r}_t)^\top(\boldsymbol{f}_t - \boldsymbol{r}_t) \right) \right] = \mathbb{E}_{\boldsymbol{f}_t \sim \mathcal{N}(\boldsymbol{\mu}_t, \boldsymbol{T}_t)} \left[ Z(\sigma^2 \mathbb{1})p(\boldsymbol{f}_t|\boldsymbol{r}_t, \sigma^2 \mathbb{1}) \right]$$

$$= Z(\sigma^2 \mathbb{1}) \int_{\mathbb{R}^t} p(\boldsymbol{f}_t|\boldsymbol{\mu}_t, \boldsymbol{T}_t)p(\boldsymbol{f}_t|\boldsymbol{r}_t, \sigma^2 \mathbb{1})\mathrm{d}\boldsymbol{f}_t$$

$$= Z(\sigma^2 \mathbb{1}) \int_{\mathbb{R}^t} p(\boldsymbol{\mu}_t|\boldsymbol{r}_t, \boldsymbol{T}_t + \sigma^2 \mathbb{1})p(\boldsymbol{f}_t|\boldsymbol{\mu}_c, \boldsymbol{\Sigma}_c)\mathrm{d}\boldsymbol{f}_t$$

$$= Z(\sigma^2 \mathbb{1})p(\boldsymbol{\mu}_t|\boldsymbol{r}_t, \boldsymbol{T}_t + \sigma^2 \mathbb{1})$$

$$= \sqrt{\frac{\det(\sigma^2 \mathbb{1})}{\det(\boldsymbol{T}_t + \sigma^2 \mathbb{1})}} \exp\left( -\frac{1}{2}(\boldsymbol{\mu}_t - \boldsymbol{r}_t)^\top(\boldsymbol{T}_t + \sigma^2 \mathbb{1})^{-1}(\boldsymbol{\mu}_t - \boldsymbol{r}_t) \right)$$

Substituting this into (16), the constraint becomes

$$\|\Phi_t \boldsymbol{\theta}^* - \boldsymbol{r}_t\|_2^2 \leq \sigma^2(\boldsymbol{\mu}_t - \boldsymbol{r}_t)^\top(\boldsymbol{T}_t + \sigma^2 \mathbb{1})^{-1}(\boldsymbol{\mu}_t - \boldsymbol{r}_t) - 2\sigma^2 \ln\left( \sqrt{\frac{\det(\sigma^2 \mathbb{1})}{\det(\boldsymbol{T}_t + \sigma^2 \mathbb{1})}} \right) + 2\sigma^2 \ln\left( \frac{1}{\delta} \right).$$

$$= (\boldsymbol{\mu}_t - \boldsymbol{r}_t)^\top \left( \mathbb{1} + \frac{\boldsymbol{T}_t}{\sigma^2} \right)^{-1} (\boldsymbol{\mu}_t - \boldsymbol{r}_t) + \sigma^2 \ln\det\left( \mathbb{1} + \frac{\boldsymbol{T}_t}{\sigma^2} \right) + 2\sigma^2 \ln\left( \frac{1}{\delta} \right).$$

## C   Computing Upper Confidence Bounds

First, we state and prove some useful lemmas.

**Lemma C.1.** *For any $\alpha > 0$*

$$(\Phi_t \boldsymbol{\theta} - \boldsymbol{r}_t)^\top(\Phi_t \boldsymbol{\theta} - \boldsymbol{r}_t) + \alpha \boldsymbol{\theta}^\top \boldsymbol{\theta} - R_{\mathrm{MM},t}^2 - \alpha B^2 = (\boldsymbol{\theta} - \widehat{\boldsymbol{\theta}}_{\alpha,t})^\top \left( \Phi_t^\top \Phi_t + \alpha \mathbb{1} \right) (\boldsymbol{\theta} - \widehat{\boldsymbol{\theta}}_{\alpha,t}) - R_{\mathrm{AMM},t}^2,$$

*where $R_{\mathrm{MM},t}^2$ is the squared radius quantity from Cor. 5.2 and*

$$\widehat{\boldsymbol{\theta}}_{\alpha,t} = \left( \Phi_t^\top \Phi_t + \alpha \mathbb{1} \right)^{-1} \Phi_t^\top \boldsymbol{r}_t,$$

$$R_{\mathrm{AMM},t}^2 = R_{\mathrm{MM},t}^2 + \alpha B^2 - \boldsymbol{r}_t^\top \boldsymbol{r}_t + \boldsymbol{r}_t^\top \Phi_t \left( \Phi_t^\top \Phi_t + \alpha \mathbb{1} \right)^{-1} \Phi_t^\top \boldsymbol{r}_t.$$

*Proof.* For any symmetric matrix $\boldsymbol{A}$, we have

$$(\boldsymbol{\theta} - \boldsymbol{b})^\top \boldsymbol{A}(\boldsymbol{\theta} - \boldsymbol{b}) + c = \boldsymbol{\theta}^\top \boldsymbol{A}\boldsymbol{\theta} - 2\boldsymbol{b}^\top \boldsymbol{A}\boldsymbol{\theta} + \boldsymbol{b}^\top \boldsymbol{A}\boldsymbol{b} + c.$$

We also have

$$(\Phi_t \boldsymbol{\theta} - \boldsymbol{r}_t)^\top(\Phi_t \boldsymbol{\theta} - \boldsymbol{r}_t) + \alpha \boldsymbol{\theta}^\top \boldsymbol{\theta} - R_{\mathrm{MM},t}^2 - \alpha B^2 = \boldsymbol{\theta}^\top \left( \Phi_t^\top \Phi_t + \alpha \mathbb{1} \right) \boldsymbol{\theta} - 2\boldsymbol{r}_t^\top \Phi_t \boldsymbol{\theta} + \boldsymbol{r}_t^\top \boldsymbol{r}_t - R_{\mathrm{MM},t}^2 - \alpha B^2.$$

We can now find $\boldsymbol{A}$, $\boldsymbol{b}$ and $c$ by equating coefficients. We find that

$$\boldsymbol{A} = \Phi_t^\top \Phi_t + \alpha \mathbb{1},$$

which is a symmetric matrix. We have

$$\boldsymbol{b}^\top \boldsymbol{A} = \boldsymbol{r}_t^\top \Phi_t$$
$$\implies \boldsymbol{A}\boldsymbol{b} = \Phi_t^\top \boldsymbol{r}_t$$
$$\implies \boldsymbol{b} = \left(\Phi_t^\top \Phi_t + \alpha \mathbb{1}\right)^{-1} \Phi_t^\top \boldsymbol{r}_t = \widehat{\boldsymbol{\theta}}_{\alpha,t}.$$

Finally, we have

$$c = -R_{\mathrm{MM},t}^2 - \alpha B^2 + \boldsymbol{r}_t^\top \boldsymbol{r}_t - \boldsymbol{b}^\top \boldsymbol{A}\boldsymbol{b}$$
$$= -R_{\mathrm{MM},t}^2 - \alpha B^2 + \boldsymbol{r}_t^\top \boldsymbol{r}_t - \boldsymbol{r}_t^\top \Phi_t \left(\Phi_t^\top \Phi_t + \alpha \mathbb{1}\right)^{-1} \Phi_t^\top \boldsymbol{r}_t$$
$$= -R_{\mathrm{AMM},t}^2.$$

Therefore, we have shown that

$$(\Phi_t \boldsymbol{\theta} - \boldsymbol{r}_t)^\top (\Phi_t \boldsymbol{\theta} - \boldsymbol{r}_t) + \alpha \boldsymbol{\theta}^\top \boldsymbol{\theta} - R_{\mathrm{MM},t}^2 - \alpha B^2 = (\boldsymbol{\theta} - \widehat{\boldsymbol{\theta}}_{\alpha,t})^\top \left(\Phi_t^\top \Phi_t + \alpha \mathbb{1}\right) (\boldsymbol{\theta} - \widehat{\boldsymbol{\theta}}_{\alpha,t}) - R_{\mathrm{AMM},t}^2.$$

$\square$

**Lemma C.2.** *For any symmetric, positive-definite matrix $\boldsymbol{A} \in \mathbb{R}^{d \times d}$, any vectors $\boldsymbol{a}, \boldsymbol{b} \in \mathbb{R}^d$, any $R > 0$, and any $\eta < 0$,*

$$\max_{\boldsymbol{\theta} \in \mathbb{R}^d} \left\{ \boldsymbol{a}^\top \boldsymbol{\theta} + \eta \left( (\boldsymbol{\theta} - \boldsymbol{b})^\top \boldsymbol{A} (\boldsymbol{\theta} - \boldsymbol{b}) - R^2 \right) \right\} = \boldsymbol{a}^\top \boldsymbol{b} - \frac{1}{4\eta} \boldsymbol{a}^\top \boldsymbol{A}^{-1} \boldsymbol{a} - \eta R^2.$$

*Proof.* Let

$$f(\boldsymbol{\theta}) = \boldsymbol{a}^\top \boldsymbol{\theta} + \eta \left( (\boldsymbol{\theta} - \boldsymbol{b})^\top \boldsymbol{A} (\boldsymbol{\theta} - \boldsymbol{b}) - R^2 \right)$$

The gradient and Hessian of $f$ are

$$\frac{\partial}{\partial \boldsymbol{\theta}} f(\boldsymbol{\theta}) = \boldsymbol{a} + 2\eta \boldsymbol{A}(\boldsymbol{\theta} - \boldsymbol{b}), \qquad \frac{\partial^2}{\partial \boldsymbol{\theta}^2} f(\boldsymbol{\theta}) = 2\eta \boldsymbol{A}.$$

Since $\boldsymbol{A}$ is positive-definite and $\eta < 0$, $\frac{\partial^2}{\partial \boldsymbol{\theta}^2} f(\boldsymbol{\theta})$ is negative-definite for all $\boldsymbol{\theta} \in \mathbb{R}^d$. Therefore, any solution $\boldsymbol{\theta}^*$ of $\frac{\partial}{\partial \boldsymbol{\theta}} f(\boldsymbol{\theta}) = 0$ must be a maximiser of $f(\boldsymbol{\theta})$. There is a unique solution, which is

$$\boldsymbol{\theta}^* = \boldsymbol{b} - \frac{1}{2\eta} \boldsymbol{A}^{-1} \boldsymbol{a}.$$

The maximum is

$$f(\boldsymbol{\theta}^*) = \boldsymbol{a}^\top \boldsymbol{b} - \frac{1}{4\eta} \boldsymbol{a}^\top \boldsymbol{A}^{-1} \boldsymbol{a} - \eta R^2.$$

$\square$

**Lemma C.3.** *For any symmetric, positive-definite matrix $\boldsymbol{A} \in \mathbb{R}^{d \times d}$, any vectors $\boldsymbol{a}, \boldsymbol{b} \in \mathbb{R}^d$, and any $R > 0$,*

$$\min_{\eta < 0} \left\{ \boldsymbol{a}^\top \boldsymbol{b} - \frac{1}{4\eta} \boldsymbol{a}^\top \boldsymbol{A}^{-1} \boldsymbol{a} - \eta R^2 \right\} = \boldsymbol{a}^\top \boldsymbol{b} + R\sqrt{\boldsymbol{a}^\top \boldsymbol{A}^{-1} \boldsymbol{a}}.$$

*Proof.* Let

$$g(\eta) = \boldsymbol{a}^\top \boldsymbol{b} - \frac{1}{4\eta} \boldsymbol{a}^\top \boldsymbol{A}^{-1} \boldsymbol{a} - \eta R^2.$$

The first and second derivatives of $g$ are

$$\frac{\mathrm{d}}{\mathrm{d}\eta}g(\eta) = \frac{1}{4\eta^2}\boldsymbol{a}^\top \boldsymbol{A}^{-1}\boldsymbol{a} - R^2, \qquad \frac{\mathrm{d}^2}{\mathrm{d}\eta^2}g(\eta) = -\frac{1}{2\eta^3}\boldsymbol{a}^\top \boldsymbol{A}^{-1}\boldsymbol{a}.$$

Since $\boldsymbol{A}$ is positive-definite, $\frac{\mathrm{d}^2}{\mathrm{d}\eta^2}g(\eta)$ is positive for all $\eta < 0$. Therefore, any negative solution $\eta^*$ of $\frac{\mathrm{d}}{\mathrm{d}\eta}g(\eta) = 0$ must be a minimiser of $g(\eta)$. There is a unique (negative) solution, which is

$$\eta^* = -\frac{1}{2R}\sqrt{\boldsymbol{a}^\top \boldsymbol{A}^{-1}\boldsymbol{a}}.$$

The minimum is

$$g(\eta^*) = \boldsymbol{a}^\top \boldsymbol{b} + R\sqrt{\boldsymbol{a}^\top \boldsymbol{A}^{-1}\boldsymbol{a}}.$$

$\square$

### C.1   Analytic UCBs

Here, we prove Theorem 6.1, which states that for all $\alpha > 0$:

$$\max_{\boldsymbol{\theta}\in\Theta_t}\left\{\phi(a)^\top \boldsymbol{\theta}\right\} \leq \phi(a)^\top \widehat{\boldsymbol{\theta}}_{\alpha,t} + R_{\mathrm{AMM},t}\sqrt{\phi(a)^\top \left(\Phi_t^\top \Phi_t + \alpha\mathbb{1}\right)^{-1}\phi(a)}, \tag{18}$$

$$\text{where} \qquad \widehat{\boldsymbol{\theta}}_{\alpha,t} = \left(\Phi_t^\top \Phi_t + \alpha\mathbb{1}\right)^{-1}\Phi_t^\top \boldsymbol{r}_t,$$

$$R_{\mathrm{AMM},t}^2 = R_{\mathrm{MM},t}^2 + \alpha B^2 - \boldsymbol{r}_t^\top \boldsymbol{r}_t + \boldsymbol{r}_t^\top \Phi_t \left(\Phi_t^\top \Phi_t + \alpha\mathbb{1}\right)^{-1}\Phi_t^\top \boldsymbol{r}_t.$$

$\Theta_t$ is the confidence set at time $t$ in our confidence sequence from Cor. 5.2 and $R_{\mathrm{MM},t}$ is the radius from Cor. 5.2 and Eq. (5). As well as proving this statement, we will also show that if $\Theta_t$ has an interior point, then when the right-hand-side of (18) is optimised with respect to $\alpha > 0$, the inequality in (18) becomes an equality, i.e.

$$\max_{\boldsymbol{\theta}\in\Theta_t}\left\{\phi(a)^\top \boldsymbol{\theta}\right\} = \min_{\alpha>0}\left\{\phi(a)^\top \widehat{\boldsymbol{\theta}}_{\alpha,t} + R_{\mathrm{AMM},t}\sqrt{\phi(a)^\top \left(\Phi_t^\top \Phi_t + \alpha\mathbb{1}\right)^{-1}\phi(a)}\right\}. \tag{19}$$

*Proof of Thm. 6.1.* We use weak Lagrangian duality to prove the upper bound and strong Lagrangian duality to prove the second part. The convex optimisation problem $\max_{\boldsymbol{\theta}\in\Theta_t}\left\{\phi(a)^\top \boldsymbol{\theta}\right\}$ can be stated as

$$\max_{\boldsymbol{\theta}\in\mathbb{R}^d}\ \phi(a)^\top \boldsymbol{\theta} \quad \text{s.t. } (\Phi_t\boldsymbol{\theta} - \boldsymbol{r}_t)^\top (\Phi_t\boldsymbol{\theta} - \boldsymbol{r}_t) \leq R_{\mathrm{MM},t}^2 \quad \text{and} \quad \boldsymbol{\theta}^\top \boldsymbol{\theta} \leq B^2. \tag{20}$$

Rewriting both constraints in the form $f(\boldsymbol{\theta}) \leq 0$, we can see that the Lagrangian for this problem is

$$L(\boldsymbol{\theta}, \eta_1, \eta_2) = \phi(a)^\top \boldsymbol{\theta} + \eta_1\left((\Phi_t\boldsymbol{\theta} - \boldsymbol{r}_t)^\top (\Phi_t\boldsymbol{\theta} - \boldsymbol{r}_t) - R_{\mathrm{MM},t}^2\right) + \eta_2\left(\boldsymbol{\theta}^\top \boldsymbol{\theta} - B^2\right).$$

$\eta_1$ and $\eta_2$ are called the Lagrange multipliers. The Lagrange dual function (or just dual function) is

$$g(\eta_1, \eta_2) = \max_{\boldsymbol{\theta}\in\mathbb{R}^d}\left\{L(\boldsymbol{\theta}, \eta_1, \eta_2\right\}.$$

By weak duality, for any $\eta_1, \eta_2 \leq 0$, the dual function is an upper bound on the solution of the primal problem in (20), i.e. for any $\eta_1, \eta_2 \leq 0$

$$\max_{\boldsymbol{\theta}\in\Theta_t}\left\{\phi(a)^\top \boldsymbol{\theta}\right\} \leq g(\eta_1, \eta_2). \tag{21}$$

Alternatively, (21) can be verified by starting from the inequality $\phi(a)^\top \boldsymbol{\theta} \leq L(\boldsymbol{\theta}, \eta_1, \eta_2)$ for all $\boldsymbol{\theta} \in \Theta_t, \eta_1 \leq 0$, and $\eta_2 \leq 0$. The challenge is to set the Lagrange multipliers such that the dual function has a closed-form expression while being as close as possible to its minimum value $\min_{\eta_1,\eta_2\leq 0}\left\{g(\eta_1, \eta_2)\right\}$. We will now show that for any $\alpha > 0$, $\min_{\eta\leq 0}\left\{g(\eta, \alpha\eta)\right\}$ has a closed-form solution, which is the right-hand-side of (18). The Lagrangian, evaluated with the Lagrange multipliers $\eta$ and $\alpha\eta$, is

$$L(\boldsymbol{\theta}, \eta, \alpha\eta) = \phi(a)^\top \boldsymbol{\theta} + \eta\left((\Phi_t\boldsymbol{\theta} - \boldsymbol{r}_t)^\top (\Phi_t\boldsymbol{\theta} - \boldsymbol{r}_t) + \alpha\boldsymbol{\theta}^\top \boldsymbol{\theta} - R_{\mathrm{MM},t}^2 - \alpha B^2\right).$$

Using Lemma C.1, the Lagrangian can be rewritten as

$$L(\boldsymbol{\theta}, \eta, \alpha\eta) = \phi(a)^\top \boldsymbol{\theta} + \eta \left( (\boldsymbol{\theta} - \widehat{\boldsymbol{\theta}}_{\alpha,t})^\top (\Phi_t^\top \Phi_t + \alpha\mathbb{1})(\boldsymbol{\theta} - \widehat{\boldsymbol{\theta}}_{\alpha,t}) - R_{\mathrm{AMM},t}^2 \right).$$

Using Lemma C.2, the dual function evaluated at $\eta$ and $\alpha\eta$ is

$$g(\eta, \alpha\eta) = \max_{\boldsymbol{\theta} \in \mathbb{R}^d} \left\{ \phi(a)^\top \boldsymbol{\theta} + \eta \left( (\boldsymbol{\theta} - \widehat{\boldsymbol{\theta}}_{\alpha,t})^\top (\Phi_t^\top \Phi_t + \alpha\mathbb{1})(\boldsymbol{\theta} - \widehat{\boldsymbol{\theta}}_{\alpha,t}) - R_{\mathrm{AMM},t}^2 \right) \right\}$$

$$= \phi(a)^\top \widehat{\boldsymbol{\theta}}_{\alpha,t} - \frac{1}{4\eta} \phi(a)^\top (\Phi_t^\top \Phi_t + \alpha\mathbb{1})^{-1} \phi(a) - \eta R_{\mathrm{AMM},t}^2.$$

Using Lemma C.3, we have

$$\min_{\eta \leq 0} \{ g(\eta, \alpha\eta) \} = \min_{\eta \leq 0} \left\{ \phi(a)^\top \widehat{\boldsymbol{\theta}}_{\alpha,t} - \frac{1}{4\eta} \phi(a)^\top (\Phi_t^\top \Phi_t + \alpha\mathbb{1})^{-1} \phi(a) - \eta R_{\mathrm{AMM},t}^2 \right\}$$

$$= \phi(a)^\top \widehat{\boldsymbol{\theta}}_{\alpha,t} + R_{\mathrm{AMM},t} \sqrt{\phi(a)^\top (\Phi_t^\top \Phi_t + \alpha\mathbb{1})^{-1} \phi(a)}.$$

This concludes the proof of Theorem 6.1. To prove (19), we use strong duality. Clearly $\min_{\alpha>0} \min_{\eta \leq 0} \{ g(\eta, \alpha\eta) \} = \min_{\eta_1, \eta_2 \leq 0} \{ g(\eta_1, \eta_2) \}$, so if we optimise the upper bound in (18) with respect to $\alpha$, then we will recover the minimum of the dual function. If strong duality holds, then the minimum of the dual function is equal to $\max_{\boldsymbol{\theta} \in \Theta_t} \{ \phi(a)^\top \boldsymbol{\theta} \}$. Since $\max_{\boldsymbol{\theta} \in \Theta_t} \{ \phi(a)^\top \boldsymbol{\theta} \}$ is a convex optimisation problem, we can use Slater's condition to obtain a sufficient condition for strong duality to hold. In particular, if $\Theta_t$ has an interior point, then strong duality holds, which means

$$\max_{\boldsymbol{\theta} \in \Theta_t} \{ \phi(a)^\top \boldsymbol{\theta} \} = \min_{\alpha>0} \min_{\eta \leq 0} \{ g(\eta, \alpha\eta) \}$$

$$= \min_{\alpha>0} \left\{ \phi(a)^\top \widehat{\boldsymbol{\theta}}_{\alpha,t} + R_{\mathrm{AMM},t} \sqrt{\phi(a)^\top (\Phi_t^\top \Phi_t + \alpha\mathbb{1})^{-1} \phi(a)} \right\}.$$

$\square$

One can follow the same steps, with a few minor modifications, to prove a similar statement for lower confidence bounds. For all $\alpha > 0$

$$\min_{\boldsymbol{\theta} \in \Theta_t} \{ \phi(a)^\top \boldsymbol{\theta} \} \geq \phi(a)^\top \widehat{\boldsymbol{\theta}}_{\alpha,t} - R_{\mathrm{AMM},t} \sqrt{\phi(a)^\top (\Phi_t^\top \Phi_t + \alpha\mathbb{1})^{-1} \phi(a)}.$$

If $\Theta_t$ has an interior point, then

$$\min_{\boldsymbol{\theta} \in \Theta_t} \{ \phi(a)^\top \boldsymbol{\theta} \} = \max_{\alpha>0} \left\{ \phi(a)^\top \widehat{\boldsymbol{\theta}}_{\alpha,t} - R_{\mathrm{AMM},t} \sqrt{\phi(a)^\top (\Phi_t^\top \Phi_t + \alpha\mathbb{1})^{-1} \phi(a)} \right\}.$$

## C.2   OFUL vs AMM-UCB (and CMM-UCB)

We will now show that for any value of the parameter $\alpha$, we can choose a sequence of Gaussian mixture distributions, such that the confidence bounds of AMM-UCB (and therefore also CMM-UCB) are always tighter than the confidence bounds of OFUL (Abbasi-Yadkori et al., 2011).

To do this, we will use the following lemma.

**Lemma C.4.** *For any $\gamma > 0$, $\boldsymbol{v} \in \mathbb{R}^t$ and $\boldsymbol{M} \in \mathbb{R}^{t \times d}$, we have*

$$\boldsymbol{v}^\top \boldsymbol{v} - \boldsymbol{v}^\top \boldsymbol{M} \left( \boldsymbol{M}^\top \boldsymbol{M} + \gamma\mathbb{1} \right)^{-1} \boldsymbol{M}^\top \boldsymbol{v} = \boldsymbol{v}^\top \left( \frac{1}{\gamma} \boldsymbol{M}\boldsymbol{M}^\top + \mathbb{1} \right)^{-1} \boldsymbol{v}.$$

*Proof.* We start with the identity

$$\boldsymbol{M} \left( \boldsymbol{M}^\top \boldsymbol{M} + \gamma\mathbb{1} \right) = \left( \boldsymbol{M}\boldsymbol{M}^\top + \gamma\mathbb{1} \right) \boldsymbol{M}.$$

By post-multiplying both sides with $\left(M^\top M + \gamma \mathbb{1}\right)^{-1}$ and pre-multiplying both sides with $\left(MM^\top + \gamma \mathbb{1}\right)^{-1}$, we obtain

$$\left(MM^\top + \gamma \mathbb{1}\right)^{-1} M = M \left(M^\top M + \gamma \mathbb{1}\right)^{-1}. \tag{22}$$

Now, using (22), we have

$$
\begin{aligned}
v^\top v - v^\top M \left(M^\top M + \gamma \mathbb{1}\right)^{-1} M^\top v &= v^\top v - v^\top \left(MM^\top + \gamma \mathbb{1}\right)^{-1} MM^\top v \\
&= v^\top v - v^\top \left(MM^\top + \gamma \mathbb{1}\right)^{-1} \left(MM^\top + \gamma \mathbb{1} - \gamma \mathbb{1}\right) v \\
&= v^\top v - v^\top v + \gamma v^\top \left(MM^\top + \gamma \mathbb{1}\right)^{-1} v \\
&= v^\top \left(\frac{1}{\gamma} MM^\top + \mathbb{1}\right)^{-1} v.
\end{aligned}
$$

$\square$

With $v = r_t$ and $M = \Phi_t$, we obtain

$$r_t^\top r_t - r_t^\top \Phi_t \left(\Phi_t^\top \Phi_t + \gamma \mathbb{1}\right)^{-1} \Phi_t^\top r_t = r_t^\top \left(\frac{1}{\gamma} \Phi_t \Phi_t^\top + \mathbb{1}\right)^{-1} r_t.$$

We will also use the fact that, due to the Weinstein–Aronszajn identity, for any $\gamma > 0$

$$\det(\gamma \Phi_t^\top \Phi_t + \mathbb{1}) = \det(\gamma \Phi_t \Phi_t^\top + \mathbb{1}). \tag{23}$$

For any $\alpha > 0$ (in (Abbasi-Yadkori et al., 2011), what we call $\alpha$ is called $\lambda$), the OFUL UCB states that

$$\phi(a)^\top \boldsymbol{\theta}^* \leq \phi(a)^\top \widehat{\boldsymbol{\theta}}_{\alpha,t} + R_{\mathrm{OFUL},t} \sqrt{\phi(a)^\top \left(\Phi_t^\top \Phi_t + \alpha \mathbb{1}\right)^{-1} \phi(a)},$$

$$\text{where} \quad R_{\mathrm{OFUL},t} = \sigma \sqrt{\ln \left( \det \left(\frac{1}{\alpha} \Phi_t^\top \Phi_t + \mathbb{1}\right)\right) + 2\ln(1/\delta)} + \sqrt{\alpha} B.$$

For any $\alpha > 0$ and any $\delta \in (0,1]$, this statement holds with probability at least $1 - \delta$ for all $t \geq 0$ and all $a \in \mathcal{A}$. By comparison, our AMM-UCB holds uniformly over all $t \geq 0$, all $a \in \mathcal{A}$ *and* all $\alpha > 0$ (i.e. we could optimise the AMM-UCB with respect to $\alpha$ in a data-dependent manner, which would yield our CMM-UCB).

Notice that for any history $a_1, r_1, a_2, r_2, \ldots$ and any $\alpha > 0$, the OFUL UCB is the same as our AMM-UCB, except that our AMM radius quantity $R_{\mathrm{AMM},t}$ is replaced with $R_{\mathrm{OFUL},t}$. The same is true for the LCBs of OFUL and AMM-UCB (with the same $R_{\mathrm{OFUL},t}$), so we will only focus on the UCBs. We will now show that for any history $a_1, r_1, a_2, r_2, \ldots$ and any $\alpha > 0$, we can chose a sequence of Gaussian mixture distributions such that $R_{\mathrm{AMM},t} \leq R_{\mathrm{OFUL},t}$. This means that the UCBs of our CMM-UCB and AMM-UCB algorithms are never worse than the OFUL UCB.

Without loss of generality, suppose we choose $\alpha = \sigma^2/c$, for some $c > 0$. With this choice, the OFUL radius is

$$R_{\mathrm{OFUL},t} = \sigma \left( \sqrt{\ln \left( \det \left(\frac{c}{\sigma^2} \Phi_t^\top \Phi_t + \mathbb{1}\right)\right) + 2\ln(1/\delta)} + \frac{B}{\sqrt{c}} \right).$$

For any $\alpha > 0$ and a Gaussian mixture distribution $P_t = \mathcal{N}(\boldsymbol{\mu}_t, \boldsymbol{T}_t)$, the squared AMM-UCB radius is

$$
\begin{aligned}
R_{\mathrm{AMM},t}^2 &= R_{\mathrm{MM},t}^2 + \alpha B^2 - r_t^\top r_t + r_t^\top \Phi_t \left(\Phi_t^\top \Phi_t + \alpha \mathbb{1}\right)^{-1} \Phi_t^\top r_t \\
&= (\boldsymbol{\mu}_t - r_t)^\top \left(\mathbb{1} + \frac{\boldsymbol{T}_t}{\sigma^2}\right)^{-1} (\boldsymbol{\mu}_t - r_t) + \sigma^2 \ln \left( \det \left(\mathbb{1} + \frac{\boldsymbol{T}_t}{\sigma^2}\right)\right) + 2\sigma^2 \ln \left(\frac{1}{\delta}\right) \\
&\quad + \alpha B^2 - r_t^\top r_t + r_t^\top \Phi_t \left(\Phi_t^\top \Phi_t + \alpha \mathbb{1}\right)^{-1} \Phi_t^\top r_t.
\end{aligned}
$$

For AMM-UCB, we will use $\alpha = \sigma^2/c$ and the scaled standard mixture distributions $P_t = \mathcal{N}(\mathbf{0}, c\Phi_t\Phi_t^\top)$ for each $t$. With these choices, and using Lemma C.4 and (23), the squared AMM-UCB radius is

$$R_{\text{AMM},t}^2 = \boldsymbol{r}_t^\top \left(\frac{c}{\sigma^2}\Phi_t\Phi_t^\top + \mathbb{1}\right)^{-1} \boldsymbol{r}_t - \boldsymbol{r}_t^\top \boldsymbol{r}_t + \boldsymbol{r}_t^\top \Phi_t \left(\Phi_t^\top \Phi_t + \frac{\sigma^2}{c}\mathbb{1}\right)^{-1} \Phi_t^\top \boldsymbol{r}_t \quad (24)$$

$$+ \sigma^2 \ln\left(\det\left(\frac{c}{\sigma^2}\Phi_t\Phi_t^\top + \mathbb{1}\right)\right) + 2\sigma^2 \ln\left(\frac{1}{\delta}\right) + \frac{\sigma^2 B^2}{c}$$

$$= \sigma^2 \left(\ln\left(\det\left(\frac{c}{\sigma^2}\Phi_t^\top \Phi_t + \mathbb{1}\right)\right) + 2\ln\left(\frac{1}{\delta}\right) + \frac{B^2}{c}\right).$$

Using the basic inequality $\sqrt{a+b} \leq \sqrt{a} + \sqrt{b}$ for $a, b \geq 0$, we have

$$R_{\text{AMM},t} = \sigma\sqrt{\ln\left(\det\left(\frac{c}{\sigma^2}\Phi_t^\top \Phi_t + \mathbb{1}\right)\right) + 2\ln\left(\frac{1}{\delta}\right) + \frac{B^2}{c}}$$

$$\leq \sigma\left(\sqrt{\ln\left(\det\left(\frac{c}{\sigma^2}\Phi_t^\top \Phi_t + \mathbb{1}\right)\right) + 2\ln\left(\frac{1}{\delta}\right)} + \frac{B}{\sqrt{c}}\right)$$

$$= R_{\text{OFUL},t}.$$

Therefore, the confidence bounds of AMM-UCB, with $\alpha = \sigma^2/c$ and $P_t = \mathcal{N}(\mathbf{0}, c\Phi_t\Phi_t^\top)$, are never looser than the confidence bounds of OFUL with an arbitrary $\alpha = \sigma^2/c$. Since $\ln\left(\det\left(\frac{c}{\sigma^2}\Phi_t^\top \Phi_t + \mathbb{1}\right)\right) + 2\ln\left(\frac{1}{\delta}\right)$ and $B^2/c$ are strictly positive, there is actually a strict inequality. This means that the AMM-UCB (and CMM-UCB) confidence bounds are always strictly tighter than the OFUL confidence bounds.

Note that $P_t = \mathcal{N}(\mathbf{0}, c\Phi_t\Phi_t^\top)$ is not necessarily the best choice for the mixture distribution. With a better choice of the mixture distribution, e.g. a mixture distribution that is chosen using some prior knowledge about the expected reward function and/or refined using previously observed rewards, $R_{\text{AMM},t}$ will be smaller and the gap between AMM-UCB and OFUL will be greater.

## D   Cumulative Regret Bounds

In this section, we prove the cumulative regret bounds stated in Section 7. We prove the data-dependent regret bound in Thm. 7.5. We also prove the data-independent regret bound in Thm. 7.6 and another data-independent regret bound, which holds for more general choices of the mixture distributions and the $\alpha$ parameter.

For convenience, we use some more compact notation in this section. For a symmetric positive semi-definite matrix $\boldsymbol{A}$ and vector $\boldsymbol{x}$, let

$$\|\boldsymbol{x}\|_{\boldsymbol{A}} := \sqrt{\boldsymbol{x}^\top \boldsymbol{A}\boldsymbol{x}}.$$

Before presenting the proof of the main results, we state some useful lemmas.

**Lemma D.1** (Donsker-Varadhan Change of Measure (Donsker & Varadhan, 1976)). *For any set $\mathcal{X}$, any measurable function $h : \mathcal{X} \to \mathbb{R}$ and any probability distribution $P \in \mathcal{P}(\mathcal{X})$ (i.e. any distribution on $\mathcal{X}$), such that $\mathbb{E}_{x \sim P}[e^{h(x)}] < \infty$, we have*

$$\sup_{Q \in \mathcal{P}(\mathcal{X})} \left\{\mathbb{E}_{x \sim Q}[h(x)] - D_{\text{KL}}(Q\|P)\right\} = \ln\left(\mathbb{E}_{x \sim P}\left[e^{h(x)}\right]\right). \quad (25)$$

By rearranging (25), we have

$$\inf_{Q \in \mathcal{P}(\mathcal{X})} \left\{\mathbb{E}_{x \sim Q}[h(x)] + D_{\text{KL}}(Q\|P)\right\} = -\ln\left(\mathbb{E}_{x \sim P}\left[e^{-h(x)}\right]\right). \quad (26)$$

**Lemma D.2** (Determinant-Trace Inequality (Abbasi-Yadkori et al., 2011)). *If assumption 7.3 holds (i.e. $\|\phi(a)\|_2 \leq L$), then for any $\gamma > 0$*

$$\ln\left(\det\left(\gamma \Phi_t^\top \Phi_t + \mathbb{1}\right)\right) \leq d\ln\left(1 + \gamma t L^2/d\right). \tag{27}$$

The Determinant-Trace Inequality in Lemma 10 of (Abbasi-Yadkori et al., 2011) is stated in the form

$$\det\left(\Phi_t^\top \Phi_t + \frac{1}{\gamma}\mathbb{1}\right) \leq (1/\gamma + tL^2/d)^d. \tag{28}$$

Since $\det\left(\gamma \Phi_t^\top \Phi_t + \mathbb{1}\right) = \det\left(\Phi_t^\top \Phi_t + (1/\gamma)\mathbb{1}\right) / \det\left((1/\gamma)\mathbb{1}\right)$, the statement in (27) follows from (28).

**Lemma D.3.** *For any $\sigma > 0$ and any $\sigma_0 > 0$, define $\mathbf{\Sigma}_t = (\frac{1}{\sigma^2}\Phi_t^\top \Phi_t + \frac{1}{\sigma_0^2}\mathbb{1})^{-1}$. We have*

$$\mathrm{tr}(\Phi_t^\top \Phi_t \mathbf{\Sigma}_t) = \sigma^2 d - \frac{\sigma^2}{\sigma_0^2}\mathrm{tr}(\mathbf{\Sigma}_t) \leq \sigma^2 d.$$

*Proof.* Since $\mathbf{\Sigma}_t$ is positive-definite, its trace is positive. Now

$$
\begin{aligned}
\mathrm{tr}(\Phi_t^\top \Phi_t \mathbf{\Sigma}_t) &= \sigma^2 \mathrm{tr}\left(\frac{1}{\sigma^2}\Phi_t^\top \Phi_t \left(\frac{1}{\sigma^2}\Phi_t^\top \Phi_t + \frac{1}{\sigma_0^2}\mathbb{1}\right)^{-1}\right) \\
&= \sigma^2 \mathrm{tr}\left(\left(\frac{1}{\sigma^2}\Phi_t^\top \Phi_t + \frac{1}{\sigma_0^2}\mathbb{1}\right)\left(\frac{1}{\sigma^2}\Phi_t^\top \Phi_t + \frac{1}{\sigma_0^2}\mathbb{1}\right)^{-1} - \frac{1}{\sigma_0^2}\left(\frac{1}{\sigma^2}\Phi_t^\top \Phi_t + \frac{1}{\sigma_0^2}\mathbb{1}\right)^{-1}\right) \\
&= \sigma^2 d - \frac{\sigma^2}{\sigma_0^2}\mathrm{tr}(\mathbf{\Sigma}_t) \\
&\leq \sigma^2 d.
\end{aligned}
$$

$\square$

**Lemma D.4.** *For any $\sigma > 0$ and any $\sigma_0 > 0$, the matrix $\mathbf{\Sigma}_t = (\frac{1}{\sigma^2}\Phi_t^\top \Phi_t + \frac{1}{\sigma_0^2}\mathbb{1})^{-1}$ satisfies*

$$\mathrm{tr}(\mathbf{\Sigma}_t) \leq \frac{d}{\sigma_0^2}.$$

*Proof.* Let $\{\gamma_i\}_{i=1}^d$ denote the eigenvalues of $\Phi_t^\top \Phi_t$. Since $\Phi_t^\top \Phi_t$ is positive semi-definite, its eigenvalues are real and non-negative. From the definition of eigenvalues, one can verify that the eigenvalues of $\mathbf{\Sigma}_t$ are $\{\frac{\sigma^2}{\gamma_i + \sigma^2/\sigma_0^2}\}_{i=1}^d$. Using this, we have

$$\mathrm{tr}(\mathbf{\Sigma}_t) = \sum_{i=1}^d \frac{\sigma^2}{\gamma_i + \sigma^2/\sigma_0^2} \leq \sum_{i=1}^d \frac{\sigma^2}{\sigma^2/\sigma_0^2} = \frac{d}{\sigma_0^2}.$$

$\square$

**Lemma D.5.** *Let $\boldsymbol{\epsilon}_t$ denote the vector containing the first $t$ noise variables (so $\boldsymbol{r}_t = \Phi_t \boldsymbol{\theta}^* + \boldsymbol{\epsilon}_t$). For any $\alpha > 0$, we have*

$$
\begin{aligned}
(\Phi_t \boldsymbol{\theta}^* - \boldsymbol{r}_t)^\top (\Phi_t \boldsymbol{\theta}^* - \boldsymbol{r}_t) - \boldsymbol{r}_t^\top \boldsymbol{r}_t + \boldsymbol{r}_t^\top \Phi_t (\Phi_t^\top \Phi_t + \alpha\mathbb{1})^{-1}\Phi_t^\top \boldsymbol{r}_t &\leq \left\|\Phi_t^\top \boldsymbol{\epsilon}_t\right\|_{(\Phi_t^\top \Phi_t + \alpha\mathbb{1})^{-1}}^2 \\
&\quad + 2\alpha \left\|\boldsymbol{\theta}^*\right\|_{(\Phi_t^\top \Phi_t + \alpha\mathbb{1})^{-1}} \left\|\Phi_t^\top \boldsymbol{\epsilon}_t\right\|_{(\Phi_t^\top \Phi_t + \alpha\mathbb{1})^{-1}}.
\end{aligned}
$$

*Proof.* Using $\boldsymbol{r}_t = \Phi_t \boldsymbol{\theta}^* + \boldsymbol{\epsilon}_t$, we have

$$(\Phi_t \boldsymbol{\theta}^* - \boldsymbol{r}_t)^\top (\Phi_t \boldsymbol{\theta}^* - \boldsymbol{r}_t) = \boldsymbol{\epsilon}_t^\top \boldsymbol{\epsilon}_t,$$

and

$$
\begin{aligned}
-\boldsymbol{r}_t^\top \boldsymbol{r}_t + \boldsymbol{r}_t^\top \Phi_t (\Phi_t^\top \Phi_t + \alpha \mathbb{1})^{-1} \Phi_t^\top \boldsymbol{r}_t = {} & -(\Phi_t \boldsymbol{\theta}^* + \boldsymbol{\epsilon}_t)^\top (\Phi_t \boldsymbol{\theta}^* + \boldsymbol{\epsilon}_t) \\
& + (\Phi_t \boldsymbol{\theta}^* + \boldsymbol{\epsilon}_t)^\top \Phi_t (\Phi_t^\top \Phi_t + \alpha \mathbb{1})^{-1} \Phi_t^\top (\Phi_t \boldsymbol{\theta}^* + \boldsymbol{\epsilon}_t) \\
= {} & -\boldsymbol{\epsilon}_t^\top \boldsymbol{\epsilon}_t - 2\boldsymbol{\theta}^{*\top} \Phi_t^\top \boldsymbol{\epsilon}_t - \boldsymbol{\theta}^{*\top} \Phi_t^\top \Phi_t \boldsymbol{\theta}^* + \boldsymbol{\theta}^{*\top} \Phi_t^\top \Phi_t (\Phi_t^\top \Phi_t + \alpha \mathbb{1})^{-1} \Phi_t^\top \Phi_t \boldsymbol{\theta}^* \\
& + 2\boldsymbol{\theta}^{*\top} \Phi_t^\top \Phi_t (\Phi_t^\top \Phi_t + \alpha \mathbb{1})^{-1} \Phi_t^\top \boldsymbol{\epsilon}_t + \boldsymbol{\epsilon}_t^\top \Phi_t (\Phi_t^\top \Phi_t + \alpha \mathbb{1})^{-1} \Phi_t^\top \boldsymbol{\epsilon}_t \\
\leq {} & -\boldsymbol{\epsilon}_t^\top \boldsymbol{\epsilon}_t - 2\boldsymbol{\theta}^{*\top} \Phi_t^\top \boldsymbol{\epsilon}_t + 2\boldsymbol{\theta}^{*\top} \Phi_t^\top \Phi_t (\Phi_t^\top \Phi_t + \alpha \mathbb{1})^{-1} \Phi_t^\top \boldsymbol{\epsilon}_t \\
& + \boldsymbol{\epsilon}_t^\top \Phi_t (\Phi_t^\top \Phi_t + \alpha \mathbb{1})^{-1} \Phi_t^\top \boldsymbol{\epsilon}_t \\
= {} & -\boldsymbol{\epsilon}_t^\top \boldsymbol{\epsilon}_t - 2\alpha \boldsymbol{\theta}^{*\top} (\Phi_t^\top \Phi_t + \alpha \mathbb{1})^{-1} \Phi_t^\top \boldsymbol{\epsilon}_t + \boldsymbol{\epsilon}_t^\top \Phi_t (\Phi_t^\top \Phi_t + \alpha \mathbb{1})^{-1} \Phi_t^\top \boldsymbol{\epsilon}_t.
\end{aligned}
$$

Using the Cauchy-Schwarz inequality, we have

$$\left| \boldsymbol{\theta}^{*\top} (\Phi_t^\top \Phi_t + \alpha \mathbb{1})^{-1} \Phi_t^\top \boldsymbol{\epsilon}_t \right| \leq \left\| \boldsymbol{\theta}^* \right\|_{(\Phi_t^\top \Phi_t + \alpha \mathbb{1})^{-1}} \left\| \Phi_t^\top \boldsymbol{\epsilon}_t \right\|_{(\Phi_t^\top \Phi_t + \alpha \mathbb{1})^{-1}}.$$

Therefore

$$-2\alpha \boldsymbol{\theta}^{*\top} (\Phi_t^\top \Phi_t + \alpha \mathbb{1})^{-1} \Phi_t^\top \boldsymbol{\epsilon}_t \leq 2\alpha \left\| \boldsymbol{\theta}^* \right\|_{(\Phi_t^\top \Phi_t + \alpha \mathbb{1})^{-1}} \left\| \Phi_t^\top \boldsymbol{\epsilon}_t \right\|_{(\Phi_t^\top \Phi_t + \alpha \mathbb{1})^{-1}},$$

and

$$
\begin{aligned}
(\Phi_t \boldsymbol{\theta}^* - \boldsymbol{r}_t)^\top (\Phi_t \boldsymbol{\theta}^* - \boldsymbol{r}_t) - \boldsymbol{r}_t^\top \boldsymbol{r}_t + \boldsymbol{r}_t^\top \Phi_t (\Phi_t^\top \Phi_t + \alpha \mathbb{1})^{-1} \Phi_t^\top \boldsymbol{r}_t \leq {} & \boldsymbol{\epsilon}_t^\top \boldsymbol{\epsilon}_t - \boldsymbol{\epsilon}_t^\top \boldsymbol{\epsilon}_t + \left\| \Phi_t^\top \boldsymbol{\epsilon}_t \right\|_{(\Phi_t^\top \Phi_t + \alpha \mathbb{1})^{-1}}^2 \\
& + 2\alpha \left\| \boldsymbol{\theta}^* \right\|_{(\Phi_t^\top \Phi_t + \alpha \mathbb{1})^{-1}} \left\| \Phi_t^\top \boldsymbol{\epsilon}_t \right\|_{(\Phi_t^\top \Phi_t + \alpha \mathbb{1})^{-1}} \\
= {} & \left\| \Phi_t^\top \boldsymbol{\epsilon}_t \right\|_{(\Phi_t^\top \Phi_t + \alpha \mathbb{1})^{-1}}^2 + 2\alpha \left\| \boldsymbol{\theta}^* \right\|_{(\Phi_t^\top \Phi_t + \alpha \mathbb{1})^{-1}} \left\| \Phi_t^\top \boldsymbol{\epsilon}_t \right\|_{(\Phi_t^\top \Phi_t + \alpha \mathbb{1})^{-1}}.
\end{aligned}
$$

$\square$

**Theorem D.6** (Self-Normalised Bound for Vector-Valued Martingales (Theorem 1 of (Abbasi-Yadkori et al., 2011))). *Let $(\mathcal{H}_t | t \geq 0)$ be a filtration. Let $(\epsilon_t | t \geq 1)$ be a real-valued stochastic process such that $\epsilon_t$ is $\mathcal{H}_t$-measurable and $\epsilon_t$ is conditionally $\sigma$-sub-Gaussian for some $\sigma > 0$. Let $(\phi(a_t) | t \geq 1)$ be an $\mathbb{R}^d$-valued stochastic process such that $\phi(a_t)$ is $\mathcal{H}_{t-1}$-measurable. For any $\delta \in (0, 1]$ and any $\alpha > 0$, with probability at least $1 - \delta$*

$$\forall t \geq 0, \qquad \left\| \Phi_t^\top \boldsymbol{\epsilon}_t \right\|_{(\Phi_t^\top \Phi_t + \alpha \mathbb{1})^{-1}}^2 \leq \sigma^2 \ln \left( \det \left( \frac{1}{\alpha} \Phi_t^\top \Phi_t + \mathbb{1} \right) \right) + 2\sigma^2 \ln(1/\delta).$$

**Lemma D.7.** *For any symmetric positive semi-definite matrix $\boldsymbol{A}$ with largest eigenvalue $\gamma_{\max}$, we have*

$$\|\boldsymbol{x}\|_{\boldsymbol{A}}^2 \leq \gamma_{\max} \|\boldsymbol{x}\|_2^2.$$

*Proof.* Let $\{\gamma_i\}_{i=1}^d$ and $\{\boldsymbol{v}_i\}_{i=1}^d$ be the eigenvalues and eigenvectors of $\boldsymbol{A}$. Since $\{\boldsymbol{v}_i\}_{i=1}^d$ form a basis, there are constants $\{c_i\}_{i=1}^d$ such that $\boldsymbol{x} = \sum_{i=1}^d c_i \boldsymbol{v}_i$. We have

$$\|\boldsymbol{x}\|_{\boldsymbol{A}}^2 = \sum_{i=1, j=1}^d c_i c_j \boldsymbol{v}_i^\top \boldsymbol{A} \boldsymbol{v}_j = \sum_{i=1, j=1}^d \gamma_j c_i c_j \boldsymbol{v}_i^\top \boldsymbol{v}_j \leq \gamma_{\max} \sum_{i=1, j=1}^d c_i c_j \boldsymbol{v}_i^\top \boldsymbol{v}_j = \gamma_{\max} \|\boldsymbol{x}\|_2^2.$$

$\square$

**Lemma D.8.** *For all $x \geq 0$,*

$$\min(1, x) \leq \frac{1}{\ln(2)} \ln(1 + x).$$

*Proof.* Since $\ln(1 + x)/\ln(2)$ is monotonically increasing in $x$, we only need to prove that $x \leq \ln(1 + x)/\ln(2)$ for all $x \in [0, 1]$. For any positive constant $a$, the function $a\ln(1 + x)$ is concave on the domain $[0, 1]$. Therefore, if $x \leq a\ln(1 + x)$ at the end points $x = 0$ and $x = 1$, then $x \leq a\ln(1 + x)$ for every $x \in [0, 1]$. At $x = 0$, we have $a\ln(1 + x) = 0$ for any $a$, which means we can choose the smallest $a$ such that $1 \leq a\ln(1 + 1)$. By rearranging this inequality, we obtain $a \geq 1/\ln(2)$. □

## D.1 Data-Dependent Regret Bound

First, we show that the cumulative regret of both of our algorithms can be upper bounded by the sum of the widths of the UCB/LCBs that they use. Let

$$\mathrm{UCB}_{\Theta_t}(a) = \max_{\boldsymbol{\theta} \in \Theta_t} \left\{ \phi(a)^\top \boldsymbol{\theta} \right\}, \quad \text{and} \quad \mathrm{LCB}_{\Theta_t}(a) = \min_{\boldsymbol{\theta} \in \Theta_t} \left\{ \phi(a)^\top \boldsymbol{\theta} \right\}.$$

In words, $\mathrm{UCB}_{\Theta_t}(a)$ and $\mathrm{LCB}_{\Theta_t}(a)$ are the upper and lower confidence bounds used by CMM-UCB (evaluated at $a$). Similarly, let

$$\mathrm{AUCB}_{\Theta_t}(a) = \phi(a)^\top \widehat{\boldsymbol{\theta}}_{\alpha,t} + R_{\mathrm{AMM},t} \|\phi(a)\|_{(\Phi_t^\top \Phi_t + \alpha \mathbb{1})^{-1}},$$

$$\mathrm{ALCB}_{\Theta_t}(a) = \phi(a)^\top \widehat{\boldsymbol{\theta}}_{\alpha,t} - R_{\mathrm{AMM},t} \|\phi(a)\|_{(\Phi_t^\top \Phi_t + \alpha \mathbb{1})^{-1}}.$$

$\mathrm{AUCB}_{\Theta_t}(a)$ and $\mathrm{ALCB}_{\Theta_t}(a)$ are the analytic upper and lower confidence bounds used by AMM-UCB. Lemma D.9 shows that the cumulative regret of CMM-UCB and AMM-UCB can be upper bounded by the sum of the widths (UCB minus LCB) of the confidence bounds that they use.

**Lemma D.9.** *Suppose the actions $a_1, a_2, \ldots$ are selected by the CMM-UCB algorithm. For any adaptive sequence of mixture distributions $P_t = \mathcal{N}(\boldsymbol{\mu}_t, \boldsymbol{T}_t)$ and any $\delta \in (0, 1]$, with probability at least $1 - \delta$*

$$\forall T \geq 1, \qquad \sum_{t=1}^T \Delta(a_t) \leq \sum_{t=1}^T \mathrm{UCB}_{\Theta_{t-1}}(a_t) - \mathrm{LCB}_{\Theta_{t-1}}(a_t). \tag{29}$$

*Suppose the actions $a_1, a_2, \ldots$ are selected by the AMM-UCB algorithm. For any adaptive sequence of mixture distributions $P_t = \mathcal{N}(\boldsymbol{\mu}_t, \boldsymbol{T}_t)$ and any $\delta \in (0, 1]$, with probability at least $1 - \delta$*

$$\forall \alpha > 0, T \geq 1, \qquad \sum_{t=1}^T \Delta(a_t) \leq \sum_{t=1}^T \mathrm{AUCB}_{\Theta_{t-1}}(a_t) - \mathrm{ALCB}_{\Theta_{t-1}}(a_t). \tag{30}$$

*Proof.* Using Cor. 5.2 (i.e. the fact that $\Theta_1, \Theta_2, \ldots$ is a confidence sequence), for any adaptive sequence of mixture distributions $P_t = \mathcal{N}(\boldsymbol{\mu}_t, \boldsymbol{T}_t)$ and any $\delta \in (0, 1]$, with probability at least $1 - \delta$

$$\forall a \in \mathcal{A}, t \geq 1, \qquad \mathrm{LCB}_{\Theta_{t-1}}(a) \leq \phi(a)^\top \boldsymbol{\theta}^* \leq \mathrm{UCB}_{\Theta_{t-1}}(a).$$

Using Thm. 6.1, this implies

$$\forall \alpha > 0, a \in \mathcal{A}, t \geq 1, \qquad \mathrm{ALCB}_{\Theta_{t-1}}(a) \leq \phi(a)^\top \boldsymbol{\theta}^* \leq \mathrm{AUCB}_{\Theta_{t-1}}(a).$$

Let $a_1, a_2, \ldots$ be the actions selected by CMM-UCB, i.e. $a_t = \mathrm{argmax}_{a \in \mathcal{A}_t} \left\{ \mathrm{UCB}_{\Theta_{t-1}}(a) \right\}$. Then, with probability at least $1 - \delta$, we have

$$\sum_{t=1}^T \Delta(a_t) = \sum_{t=1}^T \phi(a_t^*)^\top \boldsymbol{\theta}^* - \phi(a_t)^\top \boldsymbol{\theta}^*$$

$$\leq \sum_{t=1}^T \mathrm{UCB}_{\Theta_{t-1}}(a_t^*) - \mathrm{LCB}_{\Theta_{t-1}}(a_t)$$

$$\leq \sum_{t=1}^T \mathrm{UCB}_{\Theta_{t-1}}(a_t) - \mathrm{LCB}_{\Theta_{t-1}}(a_t).$$

Now, let $a_1, a_2, \ldots$ be the actions selected by AMM-UCB, i.e. $a_t = \text{argmax}_{a \in \mathcal{A}_t} \{ \text{AUCB}_{\Theta_{t-1}}(a) \}$. Then, with probability at least $1 - \delta$, we have

$$
\sum_{t=1}^{T} \Delta(a_t) = \sum_{t=1}^{T} \phi(a_t^*)^\top \boldsymbol{\theta}^* - \phi(a_t)^\top \boldsymbol{\theta}^*
$$

$$
\leq \sum_{t=1}^{T} \text{AUCB}_{\Theta_{t-1}}(a_t^*) - \text{ALCB}_{\Theta_{t-1}}(a_t)
$$

$$
\leq \sum_{t=1}^{T} \text{AUCB}_{\Theta_{t-1}}(a_t) - \text{ALCB}_{\Theta_{t-1}}(a_t).
$$

$\square$

Since $\forall \alpha > 0, a \in \mathcal{A}$ and $t \geq 1$, $\text{AUCB}_{\Theta_{t-1}}(a) \geq \text{UCB}_{\Theta_{t-1}}(a)$ and $\text{ALCB}_{\Theta_{t-1}}(a) \leq \text{LCB}_{\Theta_{t-1}}(a)$, (29) implies that (30) also holds when $a_1, a_2, \ldots$ are the actions selected by CMM-UCB.

*Proof of Theorem 7.5.* We start by using Lemma D.9. Suppose $a_1, a_2, \ldots$ are the actions selected by CMM-UCB or AMM-UCB. For any adaptive sequence of mixture distributions $P_t = \mathcal{N}(\boldsymbol{\mu}_t, \boldsymbol{T}_t)$ and any $\delta \in (0, 1]$, with probability at least $1 - \delta$

$$
\forall \alpha > 0, T \geq 1, \qquad \sum_{t=1}^{T} \Delta(a_t) \leq \sum_{t=1}^{T} \text{AUCB}_{\Theta_{t-1}}(a_t) - \text{ALCB}_{\Theta_{t-1}}(a_t).
$$

Using the definitions of $\text{AUCB}_{\Theta_{t-1}}(a_t)$ and $\text{ALCB}_{\Theta_{t-1}}(a_t)$, we have

$$
\forall \alpha > 0, T \geq 1, \qquad \sum_{t=1}^{T} \Delta(a_t) \leq \sum_{t=1}^{T} 2 R_{\text{AMM},t-1} \| \phi(a_t) \|_{(\Phi_{t-1}^\top \Phi_{t-1} + \alpha \mathbb{1})^{-1}}.
$$

$\square$

## D.2 Data-Independent Regret Bound

To establish data-independent regret bounds, we first prove data-independent upper bounds on the radius $R_{\text{AMM},t}$ and the norms $\| \phi(a_t) \|_{(\Phi_{t-1}^\top \Phi_{t-1} + \alpha \mathbb{1})^{-1}}$. Then, we take the data-dependent regret bound in Lemma D.9 and substitute in these bounds on the radius and the norms.

### D.2.1 Bounding the Radius

**Lemma D.10.** *If, for any $c > 0$, the sequence of mixture distributions is $P_t = \mathcal{N}(\mathbf{0}, c \Phi_t \Phi_t^\top)$ and $\alpha = \sigma^2 / c$, then*

$$
R_{\text{AMM},t}^2 \leq \sigma^2 d \ln \left( 1 + \frac{ct L^2}{\sigma^2 d} \right) + \frac{\sigma^2 B^2}{c} + 2\sigma^2 \ln(1/\delta). \tag{31}
$$

*Proof.* In Equation (24), we already saw that for this choice of $\alpha$ and the mixture distributions, we have

$$
R_{\text{AMM},t}^2 = \sigma^2 \ln \left( \det \left( \frac{c}{\sigma^2} \Phi_t^\top \Phi_t + \mathbb{1} \right) \right) + \frac{\sigma^2 B^2}{c} + 2\sigma^2 \ln(1/\delta).
$$

To obtain a data-independent upper bound on the radius, all that remains is to upper bound $\ln \left( \det \left( \frac{c}{\sigma^2} \Phi_t^\top \Phi_t + \mathbb{1} \right) \right)$ by a data-independent quantity. Using Lemma D.2, we have

$$
\ln \left( \det \left( \frac{c}{\sigma^2} \Phi_t^\top \Phi_t + \mathbb{1} \right) \right) \leq d \ln \left( 1 + \frac{ct L^2}{\sigma^2 d} \right).
$$

Therefore

$$
R_{\text{AMM},t}^2 \leq \sigma^2 d \ln \left( 1 + \frac{ct L^2}{\sigma^2 d} \right) + \frac{\sigma^2 B^2}{c} + 2\sigma^2 \ln(1/\delta).
$$

$\square$

**Lemma D.11.** *If, for any* $\boldsymbol{\theta}_0 \in \mathbb{R}^d$ *and any* $\sigma_0 > 0$, *the sequence of mixture distributions is* $P_t = \mathcal{N}(\Phi_t \boldsymbol{\theta}_0, \sigma_0^2 \Phi_t \Phi_t^\top)$, *then for any* $\delta \in (0, 1]$ *and any* $\alpha > 0$, *with probability at least* $1 - \delta$, *for all* $t \geq 1$

$$R^2_{\text{AMM},t} \leq \sigma^2 d + \frac{\sigma^2}{\sigma_0^2} \|\boldsymbol{\theta}^* - \boldsymbol{\theta}_0\|_2^2 + \sigma^2 d \ln\left(1 + \frac{t\sigma_0^2 L^2}{\sigma^2 d}\right) + \alpha B^2 + 4\sigma^2 \ln(1/\delta)$$

$$+ \sigma^2 d \ln\left(1 + tL^2/(\alpha d)\right) + 2\sqrt{\alpha} \|\boldsymbol{\theta}^*\|_2 \sqrt{\sigma^2 d \ln\left(1 + tL^2/(\alpha d)\right) + 2\sigma^2 \ln(1/\delta)}.$$

*Proof.* In App. B.2 (see Equation 16), we saw that the squared radius $R^2_{\text{MM},t}$ can be written as

$$R^2_{\text{MM},t} = -2\sigma^2 \ln\left(\mathop{\mathbb{E}}_{\boldsymbol{f}_t \sim \mathcal{N}(\Phi_t \boldsymbol{\theta}_0, \sigma_0^2 \Phi_t \Phi_t^\top)}\left[\exp\left(-\frac{1}{2\sigma^2}(\boldsymbol{f}_t - \boldsymbol{r}_t)^\top(\boldsymbol{f}_t - \boldsymbol{r}_t)\right)\right]\right) + 2\sigma^2 \ln(1/\delta). \tag{32}$$

Using the substitution $\Phi_t \boldsymbol{\theta} = \boldsymbol{f}_t$, (32) is equivalent to

$$R^2_{\text{MM},t} = -2\sigma^2 \ln\left(\mathop{\mathbb{E}}_{\boldsymbol{\theta} \sim \mathcal{N}(\boldsymbol{\theta}_0, \sigma_0^2 \mathbb{1})}\left[\exp\left(-\frac{1}{2\sigma^2}(\Phi_t \boldsymbol{\theta} - \boldsymbol{r}_t)^\top(\Phi_t \boldsymbol{\theta} - \boldsymbol{r}_t)\right)\right]\right) + 2\sigma^2 \ln(1/\delta). \tag{33}$$

Using the Donsker-Varadhan change of measure inequality (specifically (26)), the first term on the right-hand-side of (33) is equal to

$$\inf_{Q \in \mathcal{P}(\mathbb{R}^d)} \left\{\mathop{\mathbb{E}}_{\boldsymbol{\theta} \sim Q}\left[(\Phi_t \boldsymbol{\theta} - \boldsymbol{r}_t)^\top(\Phi_t \boldsymbol{\theta} - \boldsymbol{r}_t)\right] + 2\sigma^2 D_{\text{KL}}(Q \| \mathcal{N}(\boldsymbol{\theta}_0, \sigma_0^2 \mathbb{1}))\right\}.$$

If we evaluate this at any specific distribution $Q$, we obtain an upper bound on the infimum over $Q$. We choose $Q = \mathcal{N}(\boldsymbol{\theta}^*, \boldsymbol{\Sigma}_t)$, where $\boldsymbol{\Sigma}_t = (\frac{1}{\sigma^2}\Phi_t^\top \Phi_t + \frac{1}{\sigma_0^2}\mathbb{1})^{-1}$. Combining everything so far, we have

$$R^2_{\text{MM},t} \leq \mathop{\mathbb{E}}_{\boldsymbol{\theta} \sim \mathcal{N}(\boldsymbol{\theta}^*, \boldsymbol{\Sigma}_t)}\left[(\Phi_t \boldsymbol{\theta} - \boldsymbol{r}_t)^\top(\Phi_t \boldsymbol{\theta} - \boldsymbol{r}_t)\right] + 2\sigma^2 D_{\text{KL}}(\mathcal{N}(\boldsymbol{\theta}^*, \boldsymbol{\Sigma}_t) \| \mathcal{N}(\boldsymbol{\theta}_0, \sigma_0^2 \mathbb{1})) + 2\sigma^2 \ln(1/\delta)$$

$$= (\Phi_t \boldsymbol{\theta}^* - \boldsymbol{r}_t)^\top(\Phi_t \boldsymbol{\theta}^* - \boldsymbol{r}_t) + \text{tr}(\Phi_t^\top \Phi_t \boldsymbol{\Sigma}_t) + \frac{\sigma^2}{\sigma_0^2}\text{tr}(\boldsymbol{\Sigma}_t)$$

$$- \sigma^2 d + \frac{\sigma^2}{\sigma_0^2}\|\boldsymbol{\theta}^* - \boldsymbol{\theta}_0\|_2^2 + \sigma^2 \ln\left(\frac{\det(\boldsymbol{\Sigma}_t^{-1})}{\det((1/\sigma_0^2)\mathbb{1})}\right) + 2\sigma^2 \ln(1/\delta). \tag{34}$$

Using Lemma D.3, we have

$$\text{tr}(\Phi_t^\top \Phi_t \boldsymbol{\Sigma}_t) \leq \sigma^2 d.$$

Using Lemma D.4, we have

$$\frac{\sigma^2}{\sigma_0^2}\text{tr}(\boldsymbol{\Sigma}_t) \leq \sigma^2 d.$$

Using Lemma D.2, we have

$$\ln\left(\frac{\det(\boldsymbol{\Sigma}_t^{-1})}{\det((1/\sigma_0^2)\mathbb{1})}\right) = \ln\left(\det\left(\frac{\sigma_0^2}{\sigma^2}\Phi_t^\top \Phi_t + \mathbb{1}\right)\right) \leq d \ln\left(1 + \frac{\sigma_0^2 t L^2}{\sigma^2 d}\right).$$

The bound on $R^2_{\text{MM},t}$ in (34) becomes

$$R^2_{\text{MM},t} \leq (\Phi_t \boldsymbol{\theta}^* - \boldsymbol{r}_t)^\top(\Phi_t \boldsymbol{\theta}^* - \boldsymbol{r}_t) + \frac{\sigma^2}{\sigma_0^2}\|\boldsymbol{\theta}^* - \boldsymbol{\theta}_0\|_2^2 + \sigma^2 d + \sigma^2 d \ln\left(1 + \frac{\sigma_0^2 t L^2}{\sigma^2 d}\right) + 2\sigma^2 \ln(1/\delta).$$

This means that

$$R^2_{\text{AMM},t} \leq (\Phi_t \boldsymbol{\theta}^* - \boldsymbol{r}_t)^\top(\Phi_t \boldsymbol{\theta}^* - \boldsymbol{r}_t) + \frac{\sigma^2}{\sigma_0^2}\|\boldsymbol{\theta}^* - \boldsymbol{\theta}_0\|_2^2 + \sigma^2 d + \sigma^2 d \ln\left(1 + \frac{\sigma_0^2 t L^2}{\sigma^2 d}\right) + 2\sigma^2 \ln(1/\delta)$$

$$+ \alpha B^2 - \boldsymbol{r}_t^\top \boldsymbol{r}_t + \boldsymbol{r}_t^\top \Phi_t \left(\Phi_t^\top \Phi_t + \alpha \mathbb{1}\right)^{-1} \Phi_t^\top \boldsymbol{r}_t. \tag{35}$$

Finally, using Lemma D.5, then Theorem D.6 and Lemma D.7, and then Lemma D.2, for any $\delta \in (0,1]$ and any $\alpha > 0$, with probability at least $1 - \delta$, for all $t \geq 0$ simultaneously

$$(\Phi_t \boldsymbol{\theta}^* - \boldsymbol{r}_t)^\top (\Phi_t \boldsymbol{\theta}^* - \boldsymbol{r}_t) - \boldsymbol{r}_t^\top \boldsymbol{r}_t + \boldsymbol{r}_t^\top \Phi_t \left(\Phi_t^\top \Phi_t + \alpha \mathbb{1}\right)^{-1} \Phi_t^\top \boldsymbol{r}_t$$

$$\leq \left\|\Phi_t^\top \boldsymbol{\epsilon}_t\right\|_{(\Phi_t^\top \Phi_t + \alpha \mathbb{1})^{-1}}^2 + 2\alpha \left\|\boldsymbol{\theta}^*\right\|_{(\Phi_t^\top \Phi_t + \alpha \mathbb{1})^{-1}} \left\|\Phi_t^\top \boldsymbol{\epsilon}_t\right\|_{(\Phi_t^\top \Phi_t + \alpha \mathbb{1})^{-1}}$$

$$\leq \sigma^2 \ln\left(\det\left(\frac{1}{\alpha}\Phi_t^\top \Phi_t + \mathbb{1}\right)\right) + 2\sigma^2 \ln(1/\delta) + 2\sqrt{\alpha}\left\|\boldsymbol{\theta}^*\right\|_2 \sigma \sqrt{\ln\left(\det\left(\frac{1}{\alpha}\Phi_t^\top \Phi_t + \mathbb{1}\right)\right) + 2\ln(1/\delta)}$$

$$\leq \sigma^2 d \ln\left(\det\left(1 + \frac{tL^2}{\alpha d}\right)\right) + 2\sigma^2 \ln(1/\delta) + 2\sqrt{\alpha}\left\|\boldsymbol{\theta}^*\right\|_2 \sigma \sqrt{d\ln\left(\det\left(1 + \frac{tL^2}{\alpha d}\right)\right) + 2\ln(1/\delta)}.$$

Substituting this into (35), we have

$$R_{\mathrm{AMM},t}^2 \leq \sigma^2 d + \frac{\sigma^2}{\sigma_0^2}\left\|\boldsymbol{\theta}^* - \boldsymbol{\theta}_0\right\|_2^2 + \sigma^2 d\ln\left(1 + \frac{t\sigma_0^2 L^2}{\sigma^2 d}\right) + \alpha B^2 + 4\sigma^2 \ln(1/\delta)$$

$$+ \sigma^2 d\ln\left(1 + tL^2/(\alpha d)\right) + 2\sqrt{\alpha}\left\|\boldsymbol{\theta}^*\right\|_2 \sqrt{\sigma^2 d\ln\left(1 + tL^2/(\alpha d)\right) + 2\sigma^2 \ln(1/\delta)}.$$

$\square$

### D.2.2 Bounding the Sum of Norms

We use the following upper bound on the sum of the squared norms.

**Lemma D.12** (Lemma 11 of (Abbasi-Yadkori et al., 2011)). *For any $\alpha > 0$, we have*

$$\sum_{t=1}^T \min\left(1, \|\phi(a_t)\|_{(\Phi_{t-1}^\top \Phi_{t-1} + \alpha \mathbb{1})^{-1}}^2\right) \leq \frac{1}{\ln(2)} d\ln\left(1 + \frac{TL^2}{\alpha d}\right).$$

In Lemma 11 of (Abbasi-Yadkori et al., 2011), $1/\ln(2) \approx 1.44$ is replaced with 2. We achieve an improved constant by using Lemma D.8 instead of the looser bound $\min(1, x) \leq 2\ln(1 + x)$, for $x \geq 0$.

### D.2.3 Regret Bounds

We are now ready to prove our data-independent regret bounds.

*Proof of Theorem 7.6.* Following the same steps as in the proof of Lemma D.9, we can also obtain the following data-dependent bound on the per-round regret for actions selected by CMM-UCB or AMM-UCB. For the mixture distributions $P_t = \mathcal{N}(\boldsymbol{0}_t, c\Phi_t \Phi_t^\top)$, $\alpha = \sigma^2/c$ and any $\delta \in (0,1]$, with probability at least $1 - \delta$

$$\forall t \geq 1, \qquad \Delta(a_t) \leq 2R_{\mathrm{AMM},t-1}\|\phi(a_t)\|_{(\Phi_{t-1}^\top \Phi_{t-1} + \frac{\sigma^2}{c}\mathbb{1})^{-1}}. \tag{36}$$

From Assumption 7.4 ($\phi(a)^\top \boldsymbol{\theta}^* \in [-C, C]$), we have another bound on the per-round regret

$$\Delta(a_t) \leq 2C. \tag{37}$$

The combination of (36) and (37) yields

$$\Delta(a_t) \leq \min(2C, 2R_{\mathrm{AMM},t-1}\|\phi(a_t)\|_{(\Phi_{t-1}^\top \Phi_{t-1} + \frac{\sigma^2}{c}\mathbb{1})^{-1}})$$

$$\leq 2\max(C, R_{\mathrm{AMM},t-1})\min(1, \|\phi(a_t)\|_{(\Phi_{t-1}^\top \Phi_{t-1} + \frac{\sigma^2}{c}\mathbb{1})^{-1}}).$$

Starting with the Cauchy-Schwarz inequality, we have

$$\sum_{t=1}^T \Delta(a_t) \leq \sqrt{T\sum_{t=1}^T \Delta(a_t)^2} \tag{38}$$

$$\leq \sqrt{T\sum_{t=1}^T 4\max\left(C^2, R_{\mathrm{AMM},t-1}^2\right)\min\left(1, \|\phi(a_t)\|_{(\Phi_{t-1}^\top \Phi_{t-1} + \frac{\sigma^2}{c}\mathbb{1})^{-1}}^2\right)}.$$

We will now use the upper bound on $R^2_{\text{AMM},t-1}$ from Lemma D.10. Let $U^2_{\text{AMM},t-1}$ denote this upper bound (i.e. the right-hand-side of (31)). We have

$$\sum_{t=1}^{T} \Delta(a_t) \leq \sqrt{T \sum_{t=1}^{T} 4\max\left(C^2, U^2_{\text{AMM},t-1}\right) \min\left(1, \|\phi(a_t)\|^2_{(\Phi_{t-1}^\top \Phi_{t-1} + \frac{\sigma^2}{c}\mathbb{1})^{-1}}\right)}$$

$$\leq 2\max\left(C, U_{\text{AMM},T-1}\right) \sqrt{T \sum_{t=1}^{T} \min\left(1, \|\phi(a_t)\|^2_{(\Phi_{t-1}^\top \Phi_{t-1} + \frac{\sigma^2}{c}\mathbb{1})^{-1}}\right)}$$

Finally, using the bound on the sum of norms in Lemma D.12, we have

$$\sum_{t=1}^{T} \Delta(a_t) \leq \frac{2}{\sqrt{\ln(2)}} \max\left(C, \sigma\sqrt{d\ln\left(1 + \frac{c(T-1)L^2}{\sigma^2 d}\right) + \frac{B^2}{c} + 2\ln\left(\frac{1}{\delta}\right)}\right) \sqrt{dT\ln\left(1 + \frac{cTL^2}{\sigma^2 d}\right)}.$$

$\square$

Now, we state and prove a cumulative regret bound that holds for more general choices of the mixture distributions and the parameter $\alpha$.

**Theorem D.13.** *Suppose that assumptions 7.1-7.4 hold. If, for any $\boldsymbol{\theta}_0 \in \mathbb{R}^d$ and any $\sigma_0 > 0$, the sequence of mixture distributions is $P_t = \mathcal{N}(\Phi_t\boldsymbol{\theta}_0, \sigma_0^2\Phi_t\Phi_t^\top)$, then for any $\delta \in (0, 1/2]$ and any $\alpha > 0$, with probability at least $1 - 2\delta$, for all $T \geq 1$ simultaneously, the cumulative regret of CMM-UCB and AMM-UCB is bounded by*

$$\Delta_{1:T} \leq \frac{2}{\sqrt{\ln 2}} \max\{C, U_{\text{AMM},T-1}\} \sqrt{dT\ln\left(1 + \frac{L^2 T}{\alpha d}\right)} = \mathcal{O}(d\sqrt{T}\ln(T)),$$

*where*

$$U^2_{\text{AMM},T-1} \leq \sigma^2 d + \frac{\sigma^2}{\sigma_0^2} \|\boldsymbol{\theta}^* - \boldsymbol{\theta}_0\|_2^2 + \sigma^2 d\ln\left(1 + \frac{(T-1)\sigma_0^2 L^2}{\sigma^2 d}\right) + \alpha B^2 + 4\sigma^2\ln(1/\delta) \quad (39)$$

$$+ \sigma^2 d\ln\left(1 + \frac{(T-1)L^2}{\alpha d}\right) + 2\sqrt{\alpha} \|\boldsymbol{\theta}^*\|_2 \sqrt{\sigma^2 d\ln\left(1 + \frac{(T-1)L^2}{\alpha d}\right) + 2\sigma^2\ln(1/\delta)}.$$

*Proof.* Following the proof of Theorem 7.6, we obtain (with high probability)

$$\sum_{t=1}^{T} \Delta(a_t) \leq \sqrt{T \sum_{t=1}^{T} 4\max\left(C^2, R^2_{\text{AMM},t-1}\right) \min\left(1, \|\phi(a_t)\|^2_{(\Phi_{t-1}^\top \Phi_{t-1} + \alpha\mathbb{1})^{-1}}\right)}.$$

This time, we use the bound on the radius from Lemma D.11. Let $U_{\text{AMM},T-1}$ denote this bound on the radius (i.e. the square root of the right-hand-side of (39)). Also, note that this bound on the radius holds with probability at $1 - \delta$. Since $U_{\text{AMM},T-1}$ is monotonically increasing with $T$, we have

$$\sum_{t=1}^{T} \Delta(a_t) \leq 2\max\left(C, U_{\text{AMM},T-1}\right) \sqrt{T \sum_{t=1}^{T} \min\left(1, \|\phi(a_t)\|^2_{(\Phi_{t-1}^\top \Phi_{t-1} + \alpha\mathbb{1})^{-1}}\right)}.$$

Finally, we use Lemma D.12 to obtain

$$\sum_{t=1}^{T} \Delta(a_t) \leq \frac{2}{\sqrt{\ln 2}} \max\{C, U_{\text{AMM},T-1}\} \sqrt{dT\ln\left(1 + \frac{L^2 T}{\alpha d}\right)}.$$

We used two inequalities that each hold with probability at least $1 - \delta$. By a union bound argument, the cumulative regret bound holds with probability at least $1 - 2\delta$. $\square$

# E  Additional Experiments

In this section, we present the results of some additional experiments in which we investigate the effect of the mixture distributions (or priors) on our upper and lower confidence bounds.

## E.1  The Effect of The Mixture Distributions

We investigate how our CMM-UCB upper and lower confidence bounds behave when we provide an uninformative but well-specified mixture distribution/prior, an informative and well-specified mixture distribution/prior, and a misspecified mixture distribution/prior. Here misspecification refers to prior misspecification in a Bayesian sense. For reference, we compare the behaviour of our upper and lower confidence bounds with the upper and lower limits of a Bayesian credible interval.

For a fair comparison with CMM-UCB, we attempt to construct a Bayesian credible interval that holds with high probability for all rounds $t \geq 0$. However, whilst the CMM-UCB confidence set holds with high probability over the random draw of the data $a_1, r_1, a_2, r_2, \ldots$, the Bayesian credible interval holds with high probability over the random draw of $\boldsymbol{\theta}^*$ from a prior (for fixed data $a_1, r_1, a_2, r_2, \ldots$).

For the Bayesian credible interval, we use a Gaussian prior and assume $\boldsymbol{\theta}^* \sim \mathcal{N}(\boldsymbol{\mu}_0, \boldsymbol{\Sigma}_0)$. We assume a Gaussian likelihood function, i.e. rewards are of the form $r_t = \phi(a_t)^\top \boldsymbol{\theta} + \epsilon_t$, where $\epsilon_t \sim \mathcal{N}(0, \sigma^2)$. The Bayesian posterior for $\boldsymbol{\theta}^*$ is another Gaussian $\mathcal{N}(\boldsymbol{\mu}_t, \boldsymbol{\Sigma}_t)$, where

$$\boldsymbol{\mu}_t = \boldsymbol{\Sigma}_t \left( \boldsymbol{\Sigma}_0^{-1} \boldsymbol{\mu}_0 + \frac{1}{\sigma^2} \Phi_t^\top \boldsymbol{r}_t \right), \qquad \boldsymbol{\Sigma}_t = \left( \frac{1}{\sigma^2} \Phi_t^\top \Phi_t + \boldsymbol{\Sigma}_0^{-1} \right)^{-1}.$$

Using Bayes' rule, at any round $t$, we have $\boldsymbol{\theta}^* \sim \mathcal{N}(\boldsymbol{\mu}_t, \boldsymbol{\Sigma}_t)$. Therefore

$$(\boldsymbol{\mu}_t - \boldsymbol{\theta}^*)^\top \boldsymbol{\Sigma}_t^{-1} (\boldsymbol{\mu}_t - \boldsymbol{\theta}^*) \sim \chi^2(d),$$

where $\chi^2(d)$ is a chi-squared distribution with $d$ degrees of freedom. Let $Q_d(\cdot)$ be the quantile function of the chi-squared distribution with $d$ degrees of freedom. With probability at least $1 - \delta_t$ (over the random draw of $\boldsymbol{\theta}^*$ from $\mathcal{N}(\boldsymbol{\mu}_t, \boldsymbol{\Sigma}_t)$)

$$(\boldsymbol{\mu}_t - \boldsymbol{\theta}^*)^\top \boldsymbol{\Sigma}_t^{-1} (\boldsymbol{\mu}_t - \boldsymbol{\theta}^*) \leq Q_d(1 - \delta_t). \tag{40}$$

Using a union bound argument, if $\delta_t = \frac{6\delta}{(t+1)^2 \pi^2}$, then (40) holds with probability at least $1 - \delta_t$ for all $t \geq 0$. Therefore, if $\boldsymbol{\theta}^* \sim \mathcal{N}(\boldsymbol{\mu}_0, \boldsymbol{\Sigma}_0)$, then with high probability the following credible sets contain $\boldsymbol{\theta}^*$ for all $t \geq 0$ simultaneously:

$$\Theta_t = \left\{ \boldsymbol{\theta} \in \mathbb{R}^d \,\middle|\, (\boldsymbol{\mu}_t - \boldsymbol{\theta}^*)^\top \boldsymbol{\Sigma}_t^{-1} (\boldsymbol{\mu}_t - \boldsymbol{\theta}^*) \leq Q_d \left( 1 - \frac{6\delta}{(t+1)^2 \pi^2} \right) \right\}.$$

The upper limit of the credible interval for this credible set (and the one we use in Figure 4) is

$$\sup_{\boldsymbol{\theta} \in \Theta_t} \left\{ \phi(a)^T \boldsymbol{\theta} \right\} = \phi(a)^T \boldsymbol{\mu}_t + \sqrt{Q_d \left( 1 - \frac{6\delta}{(t+1)^2 \pi^2} \right)} \sqrt{\phi(a)^\top \boldsymbol{\Sigma}_t \phi(a)}.$$

We compute confidence bounds/credible intervals for a randomly generated linear function of the form $f(x) = \phi(x)^\top \boldsymbol{\theta}^*$, with inputs $x \in \mathbb{R}$ and $\boldsymbol{\theta}^* \in \mathbb{R}^{20}$, the latter drawn from a standard Gaussian distribution and if necessary scaled down to $\|\boldsymbol{\theta}^*\|_2 \leq 10 =: B$. For the feature map $\phi$, we use Random Fourier Features. We generate random data $\{(x_k, y_k)\}_{k=1}^t$, where $y_k = \phi(x_k)^\top \boldsymbol{\theta}^* + \eta_k$, $\eta_k \sim \mathcal{N}(0, \sigma^2)$ and $\sigma = 0.1$ (so the Gaussian likelihood is well-specified).

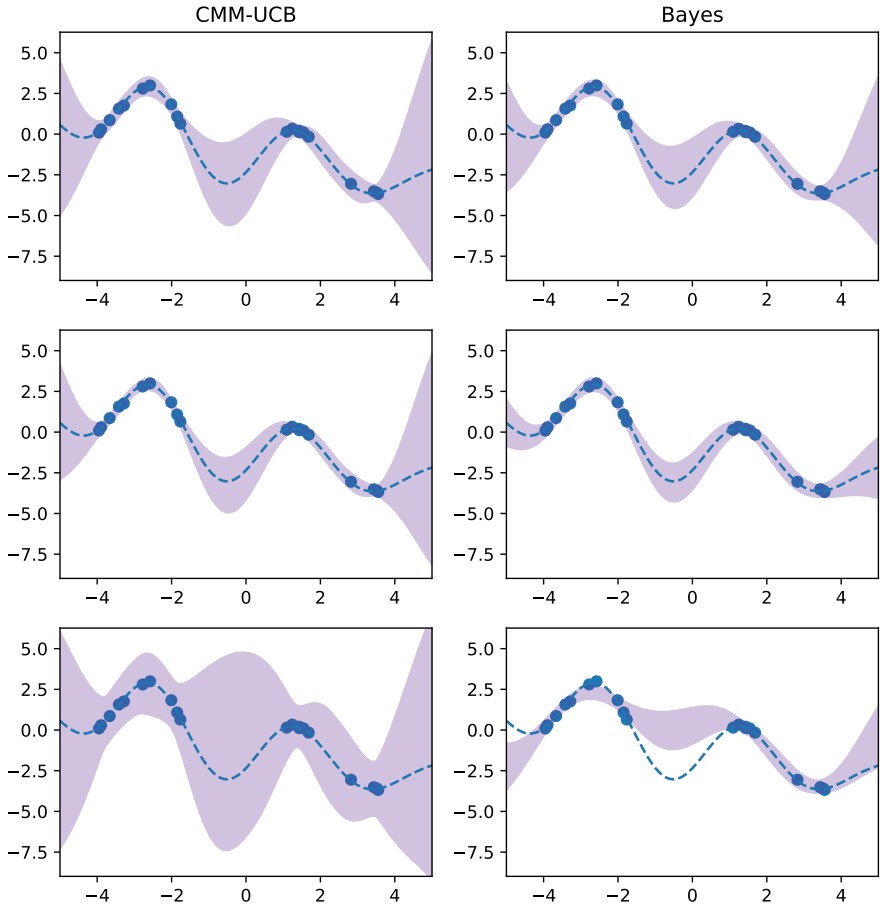

Figure 4: The upper and lower confidence bounds of our CMM-UCB method (left) and Bayesian posterior credible intervals (right) with different choices of the prior. The top row uses the prior $\boldsymbol{f}_t \sim \mathcal{N}(\boldsymbol{0}, \Phi_t \Phi_t^\top)$ for CMM-UCB and $\boldsymbol{\theta}^* \sim \mathcal{N}(\boldsymbol{0}, \mathbb{1})$) for Bayes. The middle row uses an informative prior: $\boldsymbol{f}_t \sim \mathcal{N}(\Phi_t \boldsymbol{\theta}^*, 0.1\Phi_t \Phi_t^\top)$ for CMM-UCB and $\boldsymbol{\theta}^* \sim \mathcal{N}(\boldsymbol{\theta}^*, 0.1\mathbb{1})$) for Bayes. The bottom row uses a misspecified prior: $\boldsymbol{f}_t \sim \mathcal{N}(-\Phi_t \boldsymbol{\theta}^*, 0.1\Phi_t \Phi_t^\top)$ for CMM-UCB and $\boldsymbol{\theta}^* \sim \mathcal{N}(-\boldsymbol{\theta}^*, 0.1\mathbb{1})$) for Bayes.

Figure 4 shows the CMM-UCB upper and lower confidence bounds (left) and the Bayesian credible intervals (right) with different choices of the prior. We use roughly equivalent priors for both methods. If the Bayesian credible interval uses the prior $\boldsymbol{\theta}^* \sim \mathcal{N}(\boldsymbol{\mu}_0, \boldsymbol{\Sigma}_0)$, then CMM-UCB uses the induced distribution over the function values $\Phi_t \boldsymbol{\theta}^*$, i.e. $\boldsymbol{f}_t \sim \mathcal{N}(\Phi_t \boldsymbol{\mu}_0, \Phi_t \boldsymbol{\Sigma}_0 \Phi_t^\top)$.

In the top and middle rows of Figure 4, where the prior is well-specified, we observe that the CMM-UCB upper and lower confidence bounds are slightly looser than the Bayesian credible intervals. In the bottom row of Figure 4, when the prior is misspecified, the CMM-UCB interval gets looser whereas the Bayesian credible interval becomes wrong. In summary, the Bayesian credible interval appears to be slightly tighter when the prior is well-specified and at least somewhat informative, but CMM-UCB is robust to "misspecified" mixture distributions.

## E.2 Benefits of Adaptive Mixture Distributions

In this section, we investigate a method for refining $\boldsymbol{\mu}_t$ and $\boldsymbol{T}_t$ based on previously observed actions *and* rewards. Recall that $\boldsymbol{\mu}_t$ and $\boldsymbol{T}_t$ must be chosen such that: (a) $\boldsymbol{\mu}_t$ and $\boldsymbol{T}_t$ can only depend on $a_1, \ldots, a_t$ and $r_1, \ldots, r_{t-1}$; (b) the first $t-1$ elements of $\boldsymbol{\mu}_t$ must be equal to $\boldsymbol{\mu}_{t-1}$; (c) the upper left $t-1 \times t-1$ block of $\boldsymbol{T}_t$ must be $\boldsymbol{T}_{t-1}$; (d) $\boldsymbol{T}_t$ must be positive (semi-)definite.

As in Sec. 6.4, $m$ and $k$ are any fixed mean and kernel functions. Each new row and column of $\boldsymbol{T}_t$ is set using an adaptive kernel function $k_{t-1}$. For $\beta > 0$, define

$$k_t(a, a') := k(a, a') - \boldsymbol{k}_t(a)^\top \left(\boldsymbol{K}_t + \beta \mathbb{1}\right)^{-1} \boldsymbol{k}_t(a'),$$

where $\boldsymbol{k}_t(a) = [k(a, a_1), \ldots, k(a, a_t)]^\top$ and $\boldsymbol{K}_t$ is the kernel matrix whose $(i, j)^{\text{th}}$ element is $k(a_i, a_j)$. For $\boldsymbol{T}_t$, we choose

$$\boldsymbol{T}_t = \begin{bmatrix} k_0(a_1, a_1) & k_1(a_1, a_2) & \cdots & k_{t-1}(a_1, a_t) \\ k_1(a_2, a_1) & k_1(a_2, a_2) & \cdots & k_{t-1}(a_2, a_t) \\ \vdots & \vdots & \ddots & \vdots \\ k_{t-1}(a_t, a_1) & k_{t-1}(a_t, a_2) & \cdots & k_{t-1}(a_t, a_t) \end{bmatrix}. \tag{41}$$

The $i^{\text{th}}$ column and $i^{\text{th}}$ row of this matrix depend only on only $a_1, r_1, \ldots, a_i$. Our motivation for this kernel function is: (a) generalising the usual Bayesian Gaussian process (GP) posterior covariance, one can show that if the kernel function $k$ is positive definite, then $\boldsymbol{T}_t$ is positive semi-definite; (b) $k_t$ is the Bayesian GP posterior covariance function (with a Gaussian likelihood with variance $\beta$). Each new element of $\boldsymbol{\mu}_t$ is set by evaluating an adaptive mean function $m_{t-1}$ at the latest action $a_t$. Define

$$m_t(a) := m(a) - \boldsymbol{k}_t(a)^\top \left(\boldsymbol{K}_t + \beta \mathbb{1}\right)^{-1} \left(\boldsymbol{m}_t - \boldsymbol{r}_t\right).$$

For $\boldsymbol{\mu}_t$, we choose

$$\boldsymbol{\mu}_t = [m_0(a_1), m_1(a_2), \ldots, m_{t-1}(a_t)]^\top. \tag{42}$$

The $i^{\text{th}}$ element, $m_{i-1}(a_i)$, depends on only $a_1, r_1, \ldots, a_i$, so this is a valid choice for $\boldsymbol{\mu}_t$. Note that $m_t$ is the Bayesian GP posterior mean function (again with a Gaussian likelihood with variance $\beta$).

We now compare the standard "non-adaptive" mixture distributions (with $\boldsymbol{\mu}_t$ and $\boldsymbol{T}_t$ as in (9)) and an adaptive sequence of Gaussian mixture distributions (with $\boldsymbol{\mu}_t$ as in (42) and $\boldsymbol{T}_t$ as in (41)). With both sequences of mixture distributions, we use $m(a) = 0$ and $k(a, a') = \phi(a)^\top \phi(a')$. For the adaptive sequence of mixture distributions, we set $\beta = 4\sigma^2$.

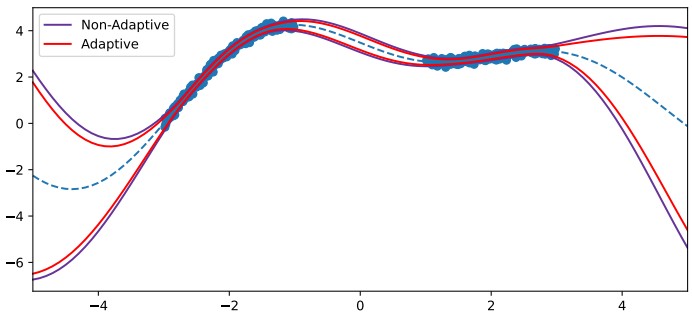

Figure 5: The upper and lower confidence bounds of CMM-UCB with the standard (non-adaptive) sequence of Gaussian mixture distributions (purple) and the adaptive sequence of Gaussian mixture distributions (red).

Figure 5 shows upper and lower confidence bounds for a randomly generated linear function $f(x) = \phi(x)^\top \boldsymbol{\theta}^*$, where $x \in \mathbb{R}$, $\boldsymbol{\theta}^* \in \mathbb{R}^{20}$, and $\phi$ is a random Fourier feature map. In this example, the adaptive sequence of mixture distributions leads to tighter upper and lower confidence bounds.

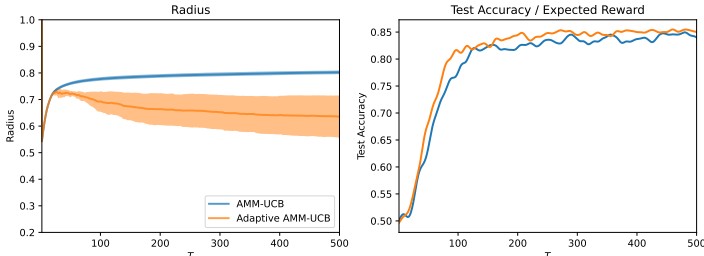

Figure 6: The radius $R_{\mathrm{AMM},t}$ (left) and test accuracy (right) with the standard sequence of Gaussian mixture distributions (blue) and the adaptive sequence of Gaussian mixture distributions (orange). On the left, we plot the mean value of the radius $R_{\mathrm{AMM},t}$ over 10 runs and show the mean $\pm$ one standard deviation in the shaded regions. On the right, we plot the mean reward over 10 runs and after Gaussian kernel smoothing.

Figure 6 shows the radius $R_{\mathrm{AMM},t}$ (left) and test accuracy (right) for AMM-UCB in the SVM hyperparameter tuning problem with the Raisin data set (described in Sec. 8.2). We observe that AMM-UCB with the adaptive mixture distributions achieves slightly higher test accuracy.

In Sec. D.2.1, we saw that for a standard sequence of mixture distributions, $R_{\mathrm{AMM},T}$ could be upper bounded by a data-independent quantity of order $\mathcal{O}(\sqrt{d\ln(T)})$. In Figure 6, $R_{\mathrm{AMM},T}$ does appear to grow roughly logarithmically with $T$ when the mixture distributions are the standard choice. However, $R_{\mathrm{AMM},T}$ appears to be bounded by a constant when the mixture distributions are adaptive. If, when using adaptive mixture distributions, we could prove a data-independent bound on $R_{\mathrm{AMM},T}$ of order $\mathcal{O}(\sqrt{d})$, then we would be able to improve our data-independent cumulative regret bounds to $\mathcal{O}(d\sqrt{T\ln(T)})$ (rather than $\mathcal{O}(d\sqrt{T}\ln(T))$). This would be within a $\sqrt{\ln(T)}$ factor of the lower bound $\Omega(d\sqrt{T})$.

## F   Efficient Radius Computation

If we compute the squared radius $R^2_{\mathrm{MM},t}$ using the expression in (5), then we have to compute the inverse and determinant of the $t \times t$ matrix $\mathbb{1} + \boldsymbol{T}_t/\sigma^2$. We will now show that for any mixture distribution of the form $P_t = \mathcal{N}(\boldsymbol{\mu}_t, \Phi_t \boldsymbol{\Sigma}_0 \Phi_t^\top)$, where $\boldsymbol{\Sigma}_0$ is symmetric and positive-definite, we can re-write the expression for $R^2_{\mathrm{MM},t}$ such that we instead need to compute the inverse and determinant of a $d \times d$ matrix. When $P_t = \mathcal{N}(\boldsymbol{\mu}_t, \Phi_t \boldsymbol{\Sigma}_0 \Phi_t^\top)$, the squared radius $R^2_{\mathrm{MM},t}$ is equal to

$$R^2_{\mathrm{MM},t} = (\boldsymbol{\mu}_t - \boldsymbol{r}_t)^\top \left( \mathbb{1} + \frac{\Phi_t \boldsymbol{\Sigma}_0 \Phi_t^\top}{\sigma^2} \right)^{-1} (\boldsymbol{\mu}_t - \boldsymbol{r}_t) + \sigma^2 \ln \left( \det \left( \mathbb{1} + \frac{\Phi_t \boldsymbol{\Sigma}_0 \Phi_t^\top}{\sigma^2} \right) \right) + 2\sigma^2 \ln(1/\delta).$$

By using the Weinstein–Aronszajn identity, and then doing some algebra, we have

$$\det \left( \mathbb{1} + \frac{\Phi_t \boldsymbol{\Sigma}_0 \Phi_t^\top}{\sigma^2} \right) = \det \left( \mathbb{1} + \frac{\boldsymbol{\Sigma}_0^{1/2} \Phi_t^\top \Phi_t \boldsymbol{\Sigma}_0^{1/2}}{\sigma^2} \right) = \det(\boldsymbol{\Sigma}_0/\sigma^2) \det \left( \Phi_t^\top \Phi_t + \sigma^2 \boldsymbol{\Sigma}_0^{-1} \right).$$

Using Lemma C.4 with $\gamma = \sigma^2$, $\boldsymbol{v} = \boldsymbol{\mu}_t - \boldsymbol{r}_t$ and $\boldsymbol{M} = \Phi_t \boldsymbol{\Sigma}_0^{1/2}$, we have

$$(\boldsymbol{\mu}_t - \boldsymbol{r}_t)^\top \left( \mathbb{1} + \frac{1}{\sigma^2} \Phi_t \boldsymbol{\Sigma}_0 \Phi_t^\top \right)^{-1} (\boldsymbol{\mu}_t - \boldsymbol{r}_t) = (\boldsymbol{\mu}_t - \boldsymbol{r}_t)^\top (\boldsymbol{\mu}_t - \boldsymbol{r}_t)$$

$$- (\boldsymbol{\mu}_t - \boldsymbol{r}_t)^\top \Phi_t \boldsymbol{\Sigma}_0^{1/2} \left( \boldsymbol{\Sigma}_0^{1/2} \Phi_t^\top \Phi_t \boldsymbol{\Sigma}_0^{1/2} + \sigma^2 \mathbb{1} \right)^{-1} \boldsymbol{\Sigma}_0^{1/2} \Phi_t^\top (\boldsymbol{\mu}_t - \boldsymbol{r}_t)$$

$$= (\boldsymbol{\mu}_t - \boldsymbol{r}_t)^\top (\boldsymbol{\mu}_t - \boldsymbol{r}_t) - (\boldsymbol{\mu}_t - \boldsymbol{r}_t)^\top \Phi_t \boldsymbol{\Sigma}_0^{1/2} \left( \boldsymbol{\Sigma}_0^{1/2} \left( \Phi_t^\top \Phi_t + \sigma^2 \boldsymbol{\Sigma}_0^{-1} \right) \boldsymbol{\Sigma}_0^{1/2} \right)^{-1} \boldsymbol{\Sigma}_0^{1/2} \Phi_t^\top (\boldsymbol{\mu}_t - \boldsymbol{r}_t)$$

$$= (\boldsymbol{\mu}_t - \boldsymbol{r}_t)^\top (\boldsymbol{\mu}_t - \boldsymbol{r}_t) - (\boldsymbol{\mu}_t - \boldsymbol{r}_t)^\top \Phi_t \left( \Phi_t^\top \Phi_t + \sigma^2 \boldsymbol{\Sigma}_0^{-1} \right)^{-1} \Phi_t^\top (\boldsymbol{\mu}_t - \boldsymbol{r}_t).$$

The resulting expression for $R^2_{\text{MM},t}$ is rather cumbersome, but the upshot is that we now (only) need to compute the inverse and determinant of the $d \times d$ matrix $\Phi_t^\top \Phi_t + \sigma^2 \mathbf{\Sigma}_0^{-1}$. Since, we compute $R^2_{\text{MM},t}$ at each round $t$, we can update the inverse of $\Phi_t^\top \Phi_t + \sigma^2 \mathbf{\Sigma}_0^{-1}$ incrementally using the Sherman-Morrison formula (Sherman & Morrison, 1950). The determinant of $\Phi_t^\top \Phi_t + \sigma^2 \mathbf{\Sigma}_0^{-1}$ can be updated incrementally using the relation

$$\det\left(\Phi_t^\top \Phi_t + \sigma^2 \mathbf{\Sigma}_0^{-1}\right) = \det(\Phi_{t-1}^\top \Phi_{t-1} + \sigma^2 \mathbf{\Sigma}_0^{-1})(1 + \phi(a_t)^\top (\Phi_{t-1}^\top \Phi_{t-1} + \sigma^2 \mathbf{\Sigma}_0^{-1})^{-1} \phi(a_t)),$$

which can be found in Eq. (6) in Lemma 11 of (Abbasi-Yadkori et al., 2011). Additionally, recall that (see Eq. (24)) when $P_t = \mathcal{N}(\mathbf{0}, c\Phi_t\Phi_t^\top)$ and $\alpha = \sigma^2/c$ for any $c > 0$, the radius $R_{\text{AMM},t}$ of our analytic confidence bounds simplifies to

$$R^2_{\text{AMM},t} = \sigma^2 \ln\left(\det\left(\frac{c}{\sigma^2}\Phi_t^\top \Phi_t + \mathbb{1}\right)\right) + \frac{\sigma^2 B^2}{c} + 2\sigma^2 \ln(1/\delta).$$

Note that the inverse $(\Phi_t^\top \Phi_t + (\sigma^2/c)\mathbb{1})^{-1}$ still appears in the expression for $\text{AUCB}_{\Theta_t}(a)$ and in the incremental determinant update rule.

