# OpenReview forum: "Improved Algorithms for Stochastic Linear Bandits Using Tail Bounds for Martingale Mixtures"
_NeurIPS.cc/2023/Conference — NeurIPS 2023 oral_

### Official Review · Reviewer_LyTZ · 2023-06-23

**Soundness:** 3 good
**Presentation:** 3 good
**Contribution:** 3 good
**Rating:** 7
**Confidence:** 3

**Summary:**

This paper studies the stochastic linear bandits as introduced in Abbasi-Yadkori et al., 2011. Here, at each time step $t$, an action set $\mathcal{A}_t$ is given. The learner then selects an action $a_t \in \mathcal{A}_t$, which maps to a feature vector $\phi(a_t)$, and subsequently receives a reward $\phi(a_t)^{\mathsf{T}}\pmb{\theta}^*+\epsilon_t$. The objective is to maximize the cumulative rewards over a designated time horizon $T$.

The main contributions of this paper can be summarized as:

1. The authors propose a general approach based on the notion of Martingale Mixtures to create confidence sets. These are subsequently utilized in the meta LinUCB algorithm to derive the bandit algorithms. It is further demonstrated that such algorithms can be efficiently computed via convex optimization.

2. The paper shows that such bandit algorithm derived from a suitable selection of mixture distributions $P_t$ attains the same worst-case regret as found in Abbasi-Yadkori et al., 2011.

3. Evidence is presented to show that the algorithm derived in this paper outperforms the approach proposed in Abbasi-Yadkori et al., 2011 when applied to several real-world datasets.

**Strengths:**

While I'm not thoroughly acquainted with the most recent literature on stochastic linear bandits, this paper appears to present compelling results. The idea of using mixture martingales to derive linear bandit algorithms is innovative and provides a general methodology for generating new algorithms. The authors supplement their theoretical contributions with empirical evidence showing that their algorithm outperforms those presented in previous literature, further strengthening their findings.

**Weaknesses:**

The downside of the paper lies in its inability to improve upon the theoretical worst-case bound as shown in Abbasi-Yadkori et al., 2011. The strength of this paper would be significantly enhanced if the authors could demonstrate that employing the methodology from their current work could theoretically offer improved bounds compared to previous results (or new results in novel settings).

Typos:
- Line 216,  $\sum_{t=1}^T$ is missing
- Line 696,  AUCB appeared twice

**Questions:**

I'm curious, how would the present work compare to the general approach as in "The Statistical Complexity of Interactive Decision Making" by D. J. Foster, S. M. Kakade, J. Qian and A. Rakhlin.?

**Limitations:**

No issue with negative societal impact.

---

> ### Author Rebuttal · Authors · 2023-08-07
>
> We thank the reviewer for their comments and questions.
>
> We respond to the reviewer's points:
>
> * *The downside of the paper lies in its inability to improve upon the theoretical worst-case bound as shown in Abbasi-Yadkori et al., 2011. The strength of this paper would be significantly enhanced if the authors could demonstrate that employing the methodology from their current work could theoretically offer improved bounds compared to previous results (or new results in novel settings).*
>
>    While our worst-case regret bound in Thm. 7.6 only matches the equivalent regret bound for the OFUL algorithm by Abassi-Yadkori et al., we can prove that our UCBs are tighter than the OFUL UCBs. In App. C.2, we show that for any value of the OFUL regularization parameter (similar to our $\alpha$ parameter), there are valid (and simple) choices of $\alpha$, $\mu_t$ and $T_t$ (which are not necessarily the optimal choices) such that our analytic UCBs (and therefore also our numerical UCBs) are always strictly tighter than the OFUL UCBs.
>
>    We believe that there is potential to obtain worst-case regret bounds with improved dependence on $T$ using our methodology. In App. E.2, we investigate “more adaptive” choices for the mixture distributions, which depend on previously observed rewards. We observe that the radius quantity grows at a slower rate (in $T$) when using these more adaptive mixture distributions. This means that the data-dependent regret bound (in Thm. 7.5) also grows at a slower rate. In future work, we would like to search for improved data-independent bounds on the radius (which would give improved worst-case regret bounds).
>
> * *I'm curious, how would the present work compare to the general approach as in "The Statistical Complexity of Interactive Decision Making" by D. J. Foster, S. M. Kakade, J. Qian and A. Rakhlin.?*
>
>    The present work gives a new and improved way to *construct confidence sets* for bandits, which are turned into improved bandit algorithms with guarantees *via the UCB/LinUCB meta-algorithm* (Sec. 4). The cited work by Foster et al. turns *any online predictor* (with guarantees) into a bandit/RL algorithm (with guarantees) *via the E2D meta-algorithm*. While Foster et al. aim to establish a complexity measure for general interactive decision making, we focus on providing as tight as possible confidence statements for a given bandit task. There are many further differences between both works in the scope and in the techniques.
>
> Thank you for pointing out the typos.
>
> We hope that this addresses the reviewer's points. We are open to discussion.

---

> > ### Comment · Reviewer_LyTZ · 2023-08-13
> >
> > I appreciate the authors addressing my concerns. Based on my understanding, while your meta-algorithm may have tighter UCBs than OFUL through appropriate parameter selection, this doesn't necessarily demonstrate that the current approach achieves tighter (asymptotic) worst-case regrets than OFUL (e.g., removing the  $\ln T$ factor from the regret). However, I concur that the "adaptive" strategy outlined in Appendix E appears to be a promising route for attaining a tight dependency on $T$.
> >
> > I agree that the current work is significant enough to justify acceptance, and I'm happy to recommend it for acceptance.

---

> > > ### Author Response · Authors · 2023-08-16
> > >
> > > Thank you for responding to our rebuttal. Your response is a good summary of the theoretical results in the paper.
> > >
> > > Thank you for raising your score and recommending acceptance.

---

### Official Review · Reviewer_uqhQ · 2023-06-30

**Soundness:** 4 excellent
**Presentation:** 4 excellent
**Contribution:** 3 good
**Rating:** 8
**Confidence:** 4

**Summary:**

A very well written paper introducing a slightly different way of constructing confidence sets for $\theta^\star$ in the adaptive regression setting. The two main differences are that the paper bounds the norm of the observation noises $\epsilon_t$ directly, rather than the projection of the noises that features in the standard bound. This is done by showing that the method of mixtures can be used with a suitably adapted sequence of mixing measures, and choosing that mixture appropriately.

**Strengths:**

Paper is _very_ well written. Method introduced is neat and the perspective taken in deriving it will be of interest to other researchers in adaptive regression/design.

**Weaknesses:**

I don't believe the paper has any significant weaknesses. One could argue that it has, perhaps, limited scope: but bandits and adaptive regression/design are very popular topics nowadays, and this is an interesting read for many people interested in those.

One issue I feel strongly about, but that can be easily addressed: you advertise that your confidence intervals are robust to misspecification (abstract, line 12). I was very disappointed to see, when I got to section 8.1 and particularly lines 282-289, that you mean misspecification to a prior in a Bayesian setting. The general setting of your work is frequentist, and in the frequentist setting, misspecification has a well understood meaning which, of course, does not coincide with that which you show. It is unclear that your method is any better in this respect than the standard concentration inequality used in OFUL (Yasin's original work); and indeed, that's a generous interpretation: it is generally accepted that bounds derived under frequentist assumptions work well in a misspecified Bayesian setting. Please remove this claim from your abstract.

Another issue is that your plots are unreadable when printed in grayscale (all lines look the same). I'm reviewing a grayscale printed version of this paper, so cannot asses empirical performance from plots. Fortunately for you, I also happen to care little for empirical results.

I have some minor feedback, solely for the purpose of improving the manuscript:

-High level: please reiterate in the introduction, e.g. on line 44, that your UCBs are tighter in an empirical sense; that you have not (to my understanding) shown them to be tighter in a theoretical sense. Your results are neat: by risking the perception that you might be overclaiming/misleading you'd be doing yourself a disservice.

-High level: your method is highly related to the rather excellent paper

-Line 65: the result you cite Chowdhury & Gopalan 2017 for is implied directly by theorem 4.1 in Yasin Abbasi-Yadkori's PhD thesis (2012); that the result of Chowdhury & Gopalan was novel is an error in the literature that ought not be propagated.

-Lines 97-98, you state that $\mathcal{H}_t$-measurability equates to 'can be calculated using the data available just after reward $r_t$ is revealed'. From the rest of the paper, I know that you know that this isn't true. It may seem like a nice simplifying explanation, but its misleading, and often inexperienced authors make a mess of things because they take that to be the definition. Indeed, in your setting, the event $\{ \theta^\star \in \Theta_t\}$ is $\mathcal{H}_t$ measurable (assuming $\Theta_t$ is a closed set, see next comment), since $\theta^\star$ is a constant and $\Theta_t$ is a measurable random set. But $\theta^\star$ is explicitly unknown at the end of the $t$th iteration, and so the indicator of $\theta^\star \in \Theta_t$  _cannot_ be computed with the information available at that point. Please remove that statement, and if you feel the reader might need a primer on measurability, please include a reference to any standard measure/probability textbook.

-Lines 95 to 100: you define what are effectively random sets with a certain property. One has to be careful around the definitions of random sets to ensure they behave in a way that one would expect. For example, we'd usually like that for a random set $A'$ subset of $A$, for any $a \in A$, the event $\{a \in A'\}$ is measurable. Your sets are closed and you work on a Polish space, so this is true; but I would point out that this is so (and indeed, you might run into trouble if the confidence sets were open). See Molchanov, Ilya: Theory of Random Sets. Springer London. 2017, 2nd edition. Proposition 1.1.2; that should be all you need.

-Eq (5), I would point out that this is just the 2-norm of $\epsilon_t$; this makes it much clearer, for example, where your naive bound of line 158 comes from.

-On your assumptions 7.1-7.4: It seems to me that 7.2+7.3 together imply a bound of the form asked for in 7.4; is there a good reason you have a separate assumption 7.4?

PS: I have not read the appendix. I am confident from the sketches in the main text that the result claimed goes through.

**Questions:**

I included some minor questions in the weaknesses section. I have no major questions for the authors.

**Limitations:**

Some aspects of the writing could be thought to overclaim, specifically when it comes to robustness to misspecification. I feel strongly that this ought to be addressed. But this is also easily fixable; I hope the authors do so.

---

> ### Author Rebuttal · Authors · 2023-08-07
>
> We thank the reviewer very much for their careful and helpful comments!
>
> We address the reviewer's points under "Weaknesses":
>
> * *One issue I feel strongly about, but that can be easily addressed: you advertise that your confidence intervals are robust to misspecification (abstract, line 12). ...*
>
>    We understand the reviewer's point, thank you for pointing this out. We will remove the claim about misspecification from the abstract, and in the main text of the paper we will clarify this term by writing "Bayesian prior misspecification". We hope that this makes clear that we make no claim about the frequentist notion of misspecification.
>
> * *Another issue is that your plots are unreadable when printed in grayscale (all lines look the same). ...*
>
>    We will use a more grayscale-friendly color scheme for the revised version of the paper.
>
> * *High level: please reiterate in the introduction, e.g. on line 44, that your UCBs are tighter in an empirical sense; that you have not (to my understanding) shown them to be tighter in a theoretical sense. ...*
>
>    Our UCBs are tighter in a theoretical sense. In App. C.2, we show that for any value of the OFUL regularization parameter (similar to our $\alpha$ parameter), there are valid (and simple) choices of $\alpha$, $\mu_t$ and $T_t$ (which are not necessarily the optimal choices) such that our analytic UCBs (and therefore also our numerical UCBs) are always strictly tighter than the OFUL UCBs. We will add a pointer to App. C.2 in the paragraph on line 44.
>
> * *High level: your method is highly related to the rather excellent paper*
>
>    We would be curious to know which excellent paper the reviewer is referring to here, especially if we haven't cited it in our paper yet.
>
> * *Lines 97-98, ...*
>
>    We will replace our previous statement about $\mathcal{H}_t$-measurability with: "each $\Theta_t$ can be calculated using the data $a_1, r_1, \dots, a_t, r_t$."
>
> * *Lines 95 to 100: ...*
>
>    After this paragraph we will add the sentence: "We remark that the confidence sets $\Theta_t$ in this paper are random closed sets in the sense of [Molchanov, Def. 1.1.1], which implies that the event $\theta\in\Theta_t$ is actually measurable for any $\theta\in\mathbb{R}^d$."
>
> * *On your assumptions 7.1-7.4: It seems to me that 7.2+7.3 together imply a bound of the form asked for in 7.4; is there a good reason you have a separate assumption 7.4?*
>
>    We agree that 7.2 + 7.3 imply assumption 7.4 with $C = LB$. Our reasons for stating a separate assumption are: (a) this is in line with the conventions of other linear bandit analyses (e.g. [Lattimore and Szepesvári, Section 19.3]); (b) this leaves open the possibility that a better (than $LB$) value for $C$ is known.
>
> The reviewer's other minor points under "Weaknesses" which we did not address above, we will directly fix in the paper as suggested.
>
> [Lattimore and Szepesvári] Lattimore, T. and Szepesvári, C., *Bandit algorithms.* Cambridge University Press. 2020.

---

> > ### Comment · Reviewer_uqhQ · 2023-08-10
> >
> > Oh, I seem to have failed to paste the paper I had in mind in, and now I had no idea what it was. My apologies.
> >
> > I've increased my score to 8, I believe this paper should absolutely be accepted.

---

> > > ### Comment · Reviewer_Cd9x · 2023-08-11
> > > **...**
> > >
> > > please share the link to this "excellent paper" once you remember it, all of us would be interested :)))

---

> > > ### Author Response · Authors · 2023-08-16
> > >
> > > Thank you for responding to our rebuttal, raising your score and recommending acceptance.

---

### Official Review · Reviewer_Cd9x · 2023-07-09

**Soundness:** 4 excellent
**Presentation:** 4 excellent
**Contribution:** 3 good
**Rating:** 7
**Confidence:** 4

**Summary:**

The paper considers the problem of stochastic linear contextual bandits, and proposes an improvement of the classic LinUCB / OFUL algorithmic template via a more sophisticated construction of the confidence sets for the hidden reward vector. The improvement comes from replacing the confidence ellipsoid used since the classic work of Abbasi-Yadkori et al. (based on the method of mixtures) with a tighter confidence set based on what the authors call "adaptive martingale mixtures". The authors eventually derive a confidence ellipsoid resembling the ones used by Russo and Van Roy, which enjoys the useful property of having a potentially data-dependent radius, and can also be turned into a confidence sequence very easily via an application of Ville's inequality. Using two different methods (an exact convex solver and an approximation of the optimal width), the authors then turn these confidence sets for theta into confidence bounds for the rewards of each action, and use the resulting bounds in a UCB scheme. The algorithms are then shown to outperform standard LinUCB in some simple experiments.

**Strengths:**

The paper is superbly written and presents a very interesting technique, improving one of the main building blocks of UCB algorithms that have been used for over a decade without any substantial changes. The proposed techniques make use of techniques that have recently gained popularity for mean estimation and proving PAC-Bayesian generalization bounds. The derivations up until the end of Section 5 are very neat and satisfying, and I believe that everyone interested in confidence ellipsoids and linear bandits should find them interesting.


**Weaknesses:**

The concrete approaches proposed in Section 6 are also nice but leave something to be desired. Perhaps it is my fault, but I have missed how one should choose the predictions mu_t and T_t. From what I understand, both approaches in Sections 6.1 and 6.2 should give valid results irrespective of the choice of these parameters, but I still wonder how the choice will impact the quality of the guarantees. The authors only suggest that choosing mu and T that "are good predictors of the (stochastic) reward r" will yield good results, but do not elaborate further. For instance, is setting mu_t as the least squares estimator and T_t as the Grammian a good idea? It feels somewhat unsatisfying to introduce all the possibilities for adaptivity and then simply set a constant lambda_t and go with the standard choice of mu and T... Also, the confidence set proposed in Section 6.2 doesn't seem all that different from the standard tail bound popularized by Abbasi-Yadkori et al. Accordingly, the regret bounds of Theorem 7.5 and 7.6 also take the same for as previous bounds, which is not unexpected given how the theorems are stated for rather generic choices of mu and T. I can see that the new bounds *could* be tighter, but at the moment the theory does not reflect this, which is somewhat disappointing.

One additional thing that I would like to comment on is the computational complexity of the resulting method. I understand that the UCB's obtained from the method are always convex in the feature representation of the actions (given how they are a maximum of linear functions). Thus, calculating the action with maximal UCB is a convex maximization problem, which is NP-hard in general. This is generally true for all UCB-like methods I can think of, so perhaps it is a bit odd to avoid discussing this question altogether in the paper and suggest that the gradient-based optimization scheme for finding the optimistic actions is a theoretically well-justified idea. (It is not, but I understand that it often still works in practice.) I think it would be nice to add a comment on this in order to not mislead more casual readers who may not be familiar with this computational difficulty.

Despite all my criticism above, I am happy to support acceptance of this paper to the NeurIPS program. I am looking forward to future literature addressing the current limitations of the approach proposed in this otherwise very nice paper.

**Questions:**

See above.

---

> ### Author Rebuttal · Authors · 2023-08-07
>
> We thank the reviewer for their comments. We are very happy to receive such an enthusiastic review!
>
> We now address the reviewer's points under "Weaknesses":
>
> * *how one should choose the predictions mu_t and T_t ... I still wonder how the choice will impact the quality of the guarantees ... For instance, is setting mu_t as the least squares estimator and T_t as the Grammian a good idea? It feels somewhat unsatisfying to introduce all the possibilities for adaptivity and then simply set a constant lambda_t and go with the standard choice of mu and T ...*
>
>    The effects of $\mu_t$ and $T_t$ on the tightness of the UCBs and the regret guarantees are determined by their effects on the squared radius $R_{\mathrm{MM}, t}^2$, which is defined in Eq. (5). Based on Eq. (5), we can treat the mean vector $\mu_t$ as a prediction of the reward vector $r_t$. The covariance matrix $T_t$ can be thought of as the uncertainty associated with this prediction. If the distance between $\mu_t$ and $r_t$ is close to 0 (i.e. $\mu_t$ is a good predictor of $r_t$), then the quadratic "prediction error" term in (5) will be close to 0, and we can afford to choose $T_t$ close to zero to minimize the log determinant term.
>
>    Unfortunately, we cannot simply choose $\mu_t = r_t$, because then the mixture distributions would not satisfy the conditions in lines 812-816 (i.e. each component of $\mu_t$ can only depend on the *preceding* rewards). Hence, we can think of the $k$th component of $\mu_t$ a prediction/guess/bet for the $k$th reward.
>
>    We agree that it is exciting to consider more adaptive choices of $\lambda_t$, $\mu_t$ and $T_t$. In App. E.2, we investigate a generic method for setting $\mu_t$ and $T_t$ based on previously observed actions and rewards; this results in somewhat lower regret in an experiment (Fig. 6 in App. E.2) and possibly better regret bounds. In App. B.1, we derive the radius $R_{\mathrm{MM}, t}$ for more general choices of $\lambda_t$. However, we don't analyze these choices of $\lambda_t$, $\mu_t$ and $T_t$ in the main paper because: (a) these choices make the resulting bandit algorithms very difficult to analyze theoretically; (b) we already struggled to fit everything into the 9 page limit. We view the analysis of more adaptive versions of our method as an exciting challenge to address in future work.
>
> * *Also, the confidence set proposed in Section 6.2 doesn't seem all that different from the standard tail bound popularized by Abbasi-Yadkori et al. Accordingly, the regret bounds of Theorem 7.5 and 7.6 also take the same for as previous bounds, which is not unexpected given how the theorems are stated for rather generic choices of mu and T. I can see that the new bounds could be tighter, but at the moment the theory does not reflect this, which is somewhat disappointing.*
>
>    In App. C.2, we compare our analytic UCBs (from Thm. 6.1) to the OFUL UCBs. We prove that for any value of the OFUL regularization parameter (similar to our $\alpha$ parameter), there are valid (and simple) choices of $\alpha$, $\mu_t$ and $T_t$ (which are not necessarily the optimal choices) such that our analytic UCBs (and therefore also our convex-program UCBs from Eq. (6)) are always strictly tighter than the OFUL UCBs. Furthermore, there is hope that more the adaptive choices for $\mu_t$ and $T_t$ described in App. E.2 could lead to regret bounds with an improved growth rate in $T$. See Fig. 6 in App. E.2 and the discussion below it for more.
>
> * *One additional thing that I would like to comment on is the computational complexity of the resulting method. ... This is generally true for all UCB-like methods I can think of, so perhaps it is a bit odd to avoid discussing this question altogether ... I think it would be nice to add a comment on this in order to not mislead more casual readers who may not be familiar with this computational difficulty.*
>
>    We fully agree with the reviewer on this point. We will add a comment on this in the revised paper.

---

> > ### Comment · Reviewer_Cd9x · 2023-08-11
> > **thank you**
> >
> > Thank you for the response! All comments make perfect sense. I am keeping my score and will continue to support acceptance of the paper.

---

> > > ### Author Response · Authors · 2023-08-16
> > >
> > > Thank you for responding to our rebuttal and recommending acceptance.

---

### Official Review · Reviewer_qEDN · 2023-07-27

**Soundness:** 4 excellent
**Presentation:** 4 excellent
**Contribution:** 4 excellent
**Rating:** 7
**Confidence:** 4

**Summary:**

This paper studies the stochastic linear bandits problem and proposes an improved algorithm with sub-linear regret guarantees. The improvement is achieved using a novel tail bound for adaptive martingale mixtures to construct tighter upper confidence bounds, which leads to a smaller regret than existing algorithms for linear bandits. The authors also verify the performance of the proposed algorithm via experiments on hyperparameter tuning tasks.

**Strengths:**

#### **The following are the strengths of the paper:**
1. The performance of any upper confidence bound (UCB) based bandit algorithm depends on the tightness of the confidence bounds. This paper proposes a novel way (using the tail bound for adaptive martingale mixtures) to improve the upper confidence bound, leading to an improved UCB-based bandit algorithm with a smaller regret.

2. The authors propose two novel methods for computing the confidence bounds: Convex Martingale Mixture UCB (CMM-UCB) and Analytic Martingale Mixture UCB (AMM-UCB). CMM-UCB uses a convex solver for the UCB maximization (gradient differentiable convex optimization), whereas AMM-UCB uses a weak Lagrangian duality to obtain an analytic UCB (gradient can be computed in closed-form or via standard automatic differentiation procedures).

3. When the mixture distribution is a Gaussian distribution, the authors showed the sub-linear data dependent/ independent cumulative regret for CMM_UCB and AMM-UCB. The author also empirically validated the performance gain of the proposed methods over existing linear bandit algorithms.

**Weaknesses:**

#### **The following are the weaknesses of the paper:**
1. Assumption of mixture distribution having Gaussian distributions: The regret bounds stated in the paper hold only when the mixture distribution is Gaussian. It is unclear from the paper how practical this assumption is and what the consequences are (especially in analysis) if this assumption does not hold.

2. Linearity assumption: Assuming a linear relationship between the reward and action's features (in high dimensional space using a known feature map) restricts the applications of proposed methods. Even though authors claim their tail bounds can be used to derive confidence sequences for non-linear reward functions, it is unclear what are the challenges of extending their work to non-linear reward functions (or kernelized bandits).


3. Unexplained notations: Many notations are not properly defined in the paper. For examples: \
i. Line 145: What is $Z_t(f_t)$ connection with existing regret analysis (e.g., OFUL)? \
ii. Line 149: How are the parameters ($\boldsymbol{\mu}_t, \boldsymbol{T}_t$) of Gaussian distribution computed? \
iii. In Theorem 7.5: what is the regret upper bound in terms of $T$? \
iv. Line 234: How to set the value of $c$? \
v: Line 248: What is $\sigma_0$ and how to set its value?

**Questions:**

Please address the weakness raised in ***Weaknesses**.

Minor comment:
1. Line  303: FTS: Freq-TS

I can change my score based on the authors' responses.

**Limitations:**

I have raised a few limitations of the paper in my response to the ***Weaknesses**. Since the paper is a theoretical contribution to linear bandits literature, I do not find any potential negative societal impact of this work.

---

> ### Author Rebuttal · Authors · 2023-08-07
>
> We thank the reviewer for their kind and constructive comments.
>
> We now address the reviewer's points under "Weaknesses":
>
> 1. The reason for choosing Gaussian mixture distributions is that the expected value just above Eq. (5) can be calculated analytically when $P_t$ is a Gaussian, which is convenient for running the algorithm and for proving regret bounds. Restricting the mixture distributions to be Gaussian does not impose any assumptions on the ground truth reward function or the bandit problem. The mixture distributions can simply be thought of as hyperparameters of our bandit algorithm.
>
> 2. Our bandit algorithm applies to kernel bandits with minor modifications. The main challenge is in obtaining data-independent regret bounds analogous to the one in Thm. 7.6, since the feature dimension is $d=\infty$ for interesting kernels.
>
>    In more detail, our confidence sequences in Corollary 5.2 can be immediately extended to non-linear reward functions $f^*$, simply by replacing $\phi(a_t)^\top\theta^*$ with $f^*(a_t)$ in the definition of $Z_t(f_t)$. The result is a confidence set, as in Corollary 5.2, with a squared error constraint for non-linear functions $f$ and a suitable boundedness constraint $\Vert f\Vert\leq B$. For kernel bandits, the corresponding UCB is still the solution of a convex program, so we can run our CMM-UCB and AMM-UCB algorithms in kernel bandit problems. While the data-dependent regret bound in Thm. 7.5 remains basically unaltered, the (derivation of the) data-independent regret bound in Thm. 7.6 must be modified. The main challenge is that the regret bound must now depend on quantities like the effective dimension or the maximum information gain of the kernel instead of the dimension $d$ of the feature vectors (which is $d=\infty$ for interesting kernels).
>
>    We will explain this in more detail in the revised version of the paper.
>
> 3. i. The random variables $Z_t(f_t)$ in our submission play a similar role to the random variables $Z_t$ in [Russo and Van Roy, App. B.1].
>
>    ii. A standard choice is $\mu_t\equiv 0$ and $T_t = \Phi_t\Phi_t^\top$, which can be motivated by choosing $\theta\sim{\mathcal N}(0,I)$ and then considering the distribution of the function values $\Phi_t\theta$. In general, $\boldsymbol{\mu}_t$ and $\boldsymbol{T}_t$ can be freely chosen as long as the sequence of mixture distributions $(\mathcal{N}(\boldsymbol{\mu}_t, \boldsymbol{T}_t) |t \in \mathbb{N})$ satisfies the requirements for being a sequence of adaptive mixture distributions; see lines 126-130 for general mixture distributions and lines 812-816 for Gaussian mixture distributions. A particular adaptive choice of $\mu_t$ and $T_t$ is examined in App. E.2 (i.e. such that the entries of $\mu_t$ and $T_t$ depend on the previously observed rewards, namely they are predictors of the reward at the newly selected action), and shown to yield good results.
>
>    iii. The growth rate in $T$ of the regret bound in Thm. 7.5 is determined by the growth rate of the radius and the sum of norms. If we upper bound each of these terms by quantities with explicit dependence on $T$, then we arrive at the data-independent bound in Thm. 7.6. We can therefore say that the dependence on $T$ of the regret bound in Thm. 7.5 is no worse than that of the regret bound in Thm. 7.6 (i.e. no worse than $\mathcal{O}(\sqrt{T}\ln(T))$).
>
>    iv. There are at least two good choices of $c$. In all of our experiments, we used $c = 1$, which is a simple choice that appears to work well. Alternatively, one can choose $c = B$. With this choice, the data-independent regret bound in Thm. 7.6. has improved dependence on the norm bound $B$ (roughly $\mathcal{O}(\sqrt{B})$ instead of $\mathcal{O}(B)$).
>
>    v. One can think of the real number $\sigma_0$ as a guess for the distance between the observed reward vector $\boldsymbol{r}_t$ and the predictions $\Phi_t\boldsymbol{\theta}_0$. Consider Eq. (5) with $\boldsymbol{\mu}_t = \Phi_t\boldsymbol{\theta}_0$ and $\boldsymbol{T}_t = \sigma_0^2\Phi_t\Phi_t^{\top}$. If the distance between $\Phi_t\boldsymbol{\theta}_0$ and $\boldsymbol{r}_t$ is close to 0, then we should choose $\sigma_0$ to be close to 0, since both the quadratic and log determinant terms in Eq. (5) will then be close to 0. Alternatively, if the distance between $\Phi_t\boldsymbol{\theta}_0$ and $\boldsymbol{r}_t$ is large, we should choose a larger $\sigma_0$ so that the quadratic term in (5) is not too large. We call $\sigma_0$ a guess because it has to be chosen *before* observing $\Phi_t\boldsymbol{\theta}_0$ and $\boldsymbol{r}_t$.
>
> Thank you for pointing out the typo on line 303.
>
> We hope that the reviewer can now recommend acceptance of the paper.
>
> [Russo and Van Roy] Russo, D. and Van Roy, B., Eluder dimension and the sample complexity of optimistic exploration. *Advances in Neural Information Processing Systems*, 26. 2013.

---

> ### Comment · Reviewer_qEDN · 2023-08-16
>
> Thanks for the clarifications. As the authors have clearly addressed my concerns, I am increasing my score.Thank you for the clarifications. As the authors have clearly addressed my concerns, I have increased my score.

---

> > ### Author Response · Authors · 2023-08-16
> >
> > Thank you for responding to our rebuttal and raising your score.

---

### Decision · Program_Chairs · 2023-09-21

**Decision:**

Accept (oral)

**Comment:**

This paper proposes novel confidence bounds form martingale sequences motivated by UCB-type algorithms for contextual bandits, in the spirit of the seminal work of Abbasi-Yadkori et al. While the method also leverages mixing distributions to obtain any-time bounds via Ville’s inequality, these mixing distributions are potentially data-driven, opening the door to empirical Bayes type estimators. Even without these improved predictors, they demonstrate data-dependent and independent regret bounds using their new bounds, and demonstrate empirically that choosing data-dependent predictors can improve performance (in E.2). While regret for contextual bandits is the main motivation, the tools introduced in Abbasi-Yadkori et al have found very broad applications and this work is sure to expand this scope even further.